# A THEORY OF NON-LINEAR FEATURE LEARNING WITH ONE GRADIENT STEP IN TWO-LAYER NEURAL NETWORKS

## ABSTRACT

Feature learning is thought to be one of the fundamental reasons for the success of deep neural networks. It is rigorously known that in two-layer fully-connected neural networks under certain conditions, one step of gradient descent on the first layer followed by ridge regression on the second layer can lead to feature learning; characterized by the appearance of a separated rank-one component—spike—in the spectrum of the feature matrix. However, with a constant gradient descent step size, this spike only carries information from the linear component of the target function and therefore learning non-linear components is impossible. We show that with a learning rate that grows with the sample size, such training in fact introduces multiple rank-one components, each corresponding to a specific polynomial feature. We further prove that the limiting large-dimensional and large sample training and test errors of the updated neural networks are fully characterized by these spikes. By precisely analyzing the improvement in the loss, we demonstrate that these non-linear features can enhance learning.

## 1 INTRODUCTION

Learning non-linear features—or representations—from data is thought to be one of the fundamental reasons for the success of deep neural networks (e.g., Bengio et al., 2013; Donahue et al., 2016; Yang & Hu, 2021; Shi et al., 2022; Radhakrishnan et al., 2022, etc.). This has been observed in a wide range of domains, including computer vision and natural language processing. At the same time, the current theoretical understanding of feature learning is incomplete. In particular, among many theoretical approaches to study neural nets, much work has focused on two-layer fully-connected neural networks with a randomly generated, untrained first layer and a trained second layer—or *random features models* (Rahimi & Recht, 2007). Despite their simplicity, random features models can capture various empirical properties of deep neural networks, and have been used to study generalization, overparametrization and "double descent", adversarial robustness, transfer learning, estimation of out-of-distribution performance, and uncertainty quantification (see e.g., Mei & Montanari (2022); Hassani & Javanmard (2022); Tripuraneni et al. (2021); Lee et al. (2023); Bombari & Mondelli (2023); Clarté et al. (2023); Lin & Dobriban (2021); Adlam et al. (2022), etc.).

Nevertheless, feature learning is absent in random features models, because the first layer weights are assumed to be randomly generated, and then fixed. Although these models can represent non-linear functions of the data, in the commonly studied setting where the sample size, dimension, and hidden layer size are proportional, under certain reasonable conditions they can only learn the *linear* component of the true model—or, teacher function—and other components of the teacher function effectively behave as Gaussian noise. Thus, in this setting, learning in a random features model is equivalent to learning in a *noisy linear model* with Gaussian features and Gaussian noise. This property is known as the *Gaussian equivalence property* (see e.g., Adlam et al. (2022); Adlam & Pennington (2020a); Hu & Lu (2023); Mei & Montanari (2022); Montanari & Saeed (2022)). While other models such as the neural tangent kernel (Jacot et al., 2018; Du et al., 2019) can be more expressive, they also lack feature learning.

To bridge the gap between random features models and feature learning, several recent approaches have shown provable feature learning for neural nets under certain conditions; see Section 1.1 for

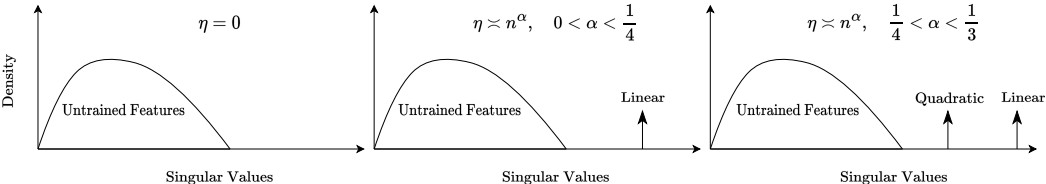

Figure 1: Spectrum of the updated feature matrix for different regimes of the gradient step size $\eta$. Spikes corresponding to monomial features are added to the spectrum of the initial matrix. The number of spikes depends on the range $\alpha$. See Theorems 3.3 and 3.4 for more details.

details. In particular, the recent pioneering work of Ba et al. (2022) analyzed two-layer neural networks, trained with one gradient step on the first layer. They showed that when the step size is small, after one gradient step, the resulting two-layer neural network can learn linear features. However, it still behaves as a noisy linear model and does not capture non-linear components of a teacher function. Moreover, they showed that for a sufficiently large step size, under certain conditions, the one-step updated random features model can outperform linear and kernel predictors. However, the effects of a large gradient step size on the features is unknown. What happens in the intermediate step size regime also remains unexplored. In this paper, we focus on the following key questions in this area:

> *What nonlinear features are learned by a two-layer neural network after one gradient update? How are these features reflected in the singular values and vectors of the feature matrix, and how does this depend on the scaling of the step size? What exactly is the improvement in the loss due to the nonlinear features learned?*

**Main Contributions.** Toward answering the above questions, we make the following contributions:

- We study feature learning in two-layer fully-connected neural networks. Specifically, we follow the training procedure introduced in Ba et al. (2022) where one step of gradient descent with step size $\eta$ is applied to the first layer weights, and the second layer weights are found by solving ridge regression on the updated features. We consider a step size $\eta \asymp n^{\alpha}, \alpha \in (0, \frac{1}{2})$ that grows with the sample size $n$ and examine how the learned features change with $\alpha$ (Section 2.1).

- In Section 3, we present a spectral analysis of the updated feature matrix. We first show that the spectrum of the feature matrix undergoes phase transitions depending on the range of $\alpha$. In particular, we find that if $\alpha \in (\frac{\ell-1}{2\ell}, \frac{\ell}{2\ell+2})$ for some $\ell \in \{1, 2, \ldots\}$, then $\ell$ separated singular values—*spikes*—will be added to the spectrum of the initial feature matrix (Theorem 3.3). Figure 1 illustrates this finding.

- Building on perturbation theory for singular vectors, we argue that the left singular vectors (principal components) associated with the $\ell$ spikes are asymptotically aligned with polynomial features of different degrees (Theorem 3.4). In other words, the updated feature matrix will contain information about the degree-$\ell$ polynomial component of the target function.

- In Section 4.1, we establish equivalence theorems (Theorem 4.1 and 4.2) which state that the training and test errors of the updated neural networks are fully characterized by the initial feature matrix and the $\ell$ spikes.

- We use the equivalence theorems from Section 4.1 to fully characterize asymptotics of the training loss for different $\ell$ (Theorem 4.4). Notably, we show that in the simple case where $\ell = 1$, the neural network does not learn non-linear functions. However, in the $\ell = 2$ regime, the neural network in fact learns quadratic components of the target function (Corollary 4.5).

## 1.1 RELATED WORKS

**Theory of shallow neural networks.** Random features models (Rahimi & Recht, 2007) have been used to study various aspects of deep learning, such as generalization (Mei & Montanari, 2022; Adlam et al., 2022; Lin & Dobriban, 2021; Mel & Pennington, 2021), adversarial robustness (Hassani & Javanmard, 2022; Bombari et al., 2023), transfer learning (Tripuraneni et al., 2021), out-of-distribution performance estimation (Lee et al., 2023), uncertainty quantification (Clarté et al., 2023), stability, and privacy (Bombari & Mondelli, 2023). This line of work builds upon nonlinear random matrix theory (see e.g., Pennington & Worah (2017); Louart et al. (2018); Fan & Wang (2020); Benigni & Péché (2021), etc.) studying the spectrum of the feature matrix of two-layer neural networks at initialization. See Section A for more discussion on related work in deep learning theory.

**Feature learning.** The problem of feature learning has been gaining a lot of attention recently (see e.g., Damian et al. (2022); Nichani et al. (2023); Zhenmei et al. (2022), etc.). Please refer to Section A for a more detailed discussion of the prior work.

Wang et al. (2022) empirically show that if learning rate is sufficiently large, an outlier in the spectrum of the weight and feature matrix emerges with the corresponding singular vector aligned to the structure of the training data. Recently, Ba et al. (2022) show that in two-layer neural networks, when the dimension, sample size and hidden layer size are proportional, one gradient step with a constant step size on the first layer weights can lead to feature learning. However, non-linear components of a single-index target function are still not learned. They further show that with a sufficiently large step size, for teacher functions with information exponent (leap index) $\kappa = 1$, and under certain conditions, the updated neural networks can outperform linear and kernel methods. However, the precise effects of large gradient step sizes on learning nonlinear features, and their precise effects on the loss remain unexplored. Dandi et al. (2023) show that for single index models with information exponent $\kappa$, there are hard directions whose learning requires a sample size of order $\Theta(d^\kappa)$. They also show that with one gradient step, and a sample size $\Theta(d)$, only a single direction of a multi-index target function can be learned. In the present work, we study the problem of learning nonlinear components of a single-index target function with $\kappa = 1$.

**High-dimensional asymptotics.** We use tools developed in work on high-dimensional asymptotics, which dates back at least to the 1960s (Raudys, 1967; Deev, 1970; Raudys, 1972). Recently, these tools have been used in a wide range of areas such as wireless communications (e.g., Tulino & Verdú (2004); Couillet & Debbah (2011), etc.), high-dimensional statistics (e.g., Raudys & Young (2004); Serdobolskii (2007); Paul & Aue (2014); Yao et al. (2015); Dobriban & Wager (2018), etc.), and machine learning (e.g., Györgyi & Tishby (1990); Opper (1995); Opper & Kinzel (1996); Couillet & Liao (2022); Engel & Van den Broeck (2001), etc.). In particular, the spectrum of so-called information plus noise random matrices that arise in Gaussian equivalence results has been studied in Dozier & Silverstein (2007); Péché (2019) and its spikes in Capitaine (2014).

## 2 PRELIMINARIES

**Notation**. We let $\mathbb{N} = \{1, 2, \ldots\}$ be the set of positive integers. For a positive integer $d \geq 1$, we denote $[d] = \{1, \ldots, d\}$. We use $O(\cdot)$ and $o(\cdot)$ for the standard big-O and little-o notation. For a matrix $\mathbf{A}$ and a non-negative integer $k$, $\mathbf{A}^{\circ k} = \mathbf{A} \circ \mathbf{A} \circ \ldots \circ \mathbf{A}$ is the matrix of the $k$-th powers of the elements of $\mathbf{A}$. For positive sequences $(A_n)_{n\geq 1}, (B_n)_{n\geq 1}$, we write $A_n = \Theta(B_n)$ or $A_n \asymp B_n$ or $A_n \equiv B_n$ if there is $C, C' > 0$ such that $CB_n \geq A_n \geq C'B_n$ for all $n$. We use $O_{\mathbb{P}}(\cdot), o_{\mathbb{P}}(\cdot)$, and $\Theta_{\mathbb{P}}(\cdot)$ for the same notions holding in probability. The symbol $\to_P$ denotes convergence in probability.

## 2.1 PROBLEM SETTING

In this paper, we study a supervised learning problem with training data $(\boldsymbol{x}_i, y_i) \in \mathbb{R}^d \times \mathbb{R}$, for $i \in [2n]$, where $d$ is the feature dimension and $n \geq 2$ is the sample size. We assume that the data is generated according to

$$\boldsymbol{x}_i \overset{\text{i.i.d.}}{\sim} \mathsf{N}(0, \mathbf{I}_d), \text{ and } y_i = f_\star(\boldsymbol{x}_i) + \varepsilon_i, \tag{1}$$

in which $f_\star$ is the ground truth or *teacher function*, and $\varepsilon_i \overset{\text{i.i.d.}}{\sim} \mathsf{N}(0, \sigma_\varepsilon^2)$ is additive noise.

We fit a model to the data in order to predict outcomes for unlabeled examples at test time; using a two-layer neural network. We let the width of the internal layer be $N \in \mathbb{N}$. For a weight matrix $\mathbf{W} \in \mathbb{R}^{N \times d}$, an activation function $\sigma : \mathbb{R} \to \mathbb{R}$ applied element-wise, and the weights $\boldsymbol{a} \in \mathbb{R}^N$ of a linear layer, we define the two-layer neural network as $f_{\mathbf{W}, \boldsymbol{a}}(\boldsymbol{x}) = \boldsymbol{a}^\top \sigma(\mathbf{W}\boldsymbol{x})$.

Following Ba et al. (2022), for the convenience of the theoretical analysis, we split the training data into two parts: $\mathbf{X} = [\boldsymbol{x}_1, \ldots, \boldsymbol{x}_n]^\top \in \mathbb{R}^{n \times d}, \boldsymbol{y} = (y_1, \ldots, y_n)^\top \in \mathbb{R}^n$ and $\tilde{\mathbf{X}} = [\boldsymbol{x}_{n+1}, \ldots, \boldsymbol{x}_{2n}]^\top \in \mathbb{R}^{n \times d}, \tilde{\boldsymbol{y}} = (y_{n+1}, \ldots, y_{2n})^\top \in \mathbb{R}^n$. We train the two layer neural network as follows. First, we initialize $\boldsymbol{a} = (a_1, \ldots, a_N)^\top$ with $a_i \overset{\text{i.i.d.}}{\sim} \mathsf{N}(0, 1/N)$ and initialize $\mathbf{W}$ with

$$\mathbf{W}_0 = [\boldsymbol{w}_{0,1}, \ldots, \boldsymbol{w}_{0,N}]^\top \in \mathbb{R}^{N \times d}, \quad \boldsymbol{w}_{0,i} \overset{\text{i.i.d.}}{\sim} \text{Unif}(\mathbb{S}^{d-1}),$$

where $\mathbb{S}^{d-1}$ is the unit sphere in $\mathbb{R}^d$ and $\text{Unif}(\mathbb{S}^{d-1})$ is the uniform measure over it. Although we choose this initialization for a simpler analysis, many arguments can be shown to hold if we switch from the uniform distribution over the sphere to a Gaussian. For example, see Section N.5. Fixing $\boldsymbol{a}$ at initialization, we perform *one step of gradient descent* on $\mathbf{W}$ with respect to the squared loss computed on $(\mathbf{X}, \boldsymbol{y})$. Recalling that $\circ$ denotes element-wise multiplication, the negative gradient can be written as

$$\mathbf{G} := -\frac{\partial}{\partial \mathbf{W}} \left[ \frac{1}{2n} \left\| \boldsymbol{y} - \sigma(\mathbf{X}\mathbf{W}^\top)\boldsymbol{a} \right\|_2^2 \right]_{\mathbf{W}=\mathbf{W}_0} = \frac{1}{n} \left[ (\boldsymbol{a}\boldsymbol{y}^\top - \boldsymbol{a}\boldsymbol{a}^\top \sigma(\mathbf{W}_0\mathbf{X}^\top)) \circ \sigma'(\mathbf{W}_0\mathbf{X}^\top) \right] \mathbf{X},$$

and the one-step update is $\mathbf{W} = [\boldsymbol{w}_1, \ldots, \boldsymbol{w}_N]^\top = \mathbf{W}_0 + \eta \, \mathbf{G}$ for a *learning rate* or *step size* $\eta$.

After the update on $\mathbf{W}$, we perform ridge regression on $\boldsymbol{a}$ using $(\tilde{\mathbf{X}}, \tilde{\boldsymbol{y}})$. Let $\mathbf{F} = \sigma(\tilde{\mathbf{X}}\mathbf{W}^\top) \in \mathbb{R}^{n \times N}$ be the feature matrix after the one-step update. For a regularization parameter $\lambda > 0$, we set

$$\hat{\boldsymbol{a}} = \hat{\boldsymbol{a}}(\mathbf{F}) = \arg\min_{\boldsymbol{a} \in \mathbb{R}^N} \frac{1}{n} \|\tilde{\boldsymbol{y}} - \mathbf{F}\boldsymbol{a}\|_2^2 + \lambda \|\boldsymbol{a}\|_2^2 = \left( \mathbf{F}^\top \mathbf{F} + \lambda n \mathbf{I}_N \right)^{-1} \mathbf{F}^\top \tilde{\boldsymbol{y}}. \tag{2}$$

Then, for a test datapoint with features $\boldsymbol{x}$, we predict the outcome $\hat{y} = f_{\mathbf{W}, \hat{\boldsymbol{a}}}(\boldsymbol{x}) = \hat{\boldsymbol{a}}^\top \sigma(\mathbf{W}\boldsymbol{x})$.

## 2.2 CONDITIONS

Our theoretical analysis applies under the following conditions:

**Condition 2.1** (Asymptotic setting). *We assume that the sample size $n$, dimension $d$, and width of hidden layer $N$ all tend to infinity with*

$$d/n \to \phi > 0, \quad \text{and} \quad d/N \to \psi > 0.$$

We require the following conditions on the teacher function $f_\star$.

**Condition 2.2.** *We let $f_\star : \mathbb{R}^d \to \mathbb{R}$ be a single-neuron model $f_\star(\boldsymbol{x}) = \sigma_\star(\boldsymbol{x}^\top \boldsymbol{\beta}_\star)$, where $\boldsymbol{\beta}_\star \in \mathbb{R}^d$ is an unknown parameter with $\boldsymbol{\beta}_\star \sim \mathsf{N}(0, \frac{1}{d}\mathbf{I}_d)$ and $\sigma_\star : \mathbb{R} \to \mathbb{R}$ is a teacher activation function. We further assume that $\sigma_\star : \mathbb{R} \to \mathbb{R}$ is $\Theta(1)$-Lipschitz.*

We let $H_k$, $k \geq 1$ be the (probabilist's) Hermite polynomials on $\mathbb{R}$ defined by

$$H_k(x) = (-1)^k \exp(x^2/2) \frac{d^k}{dx^k} \exp(-x^2/2),$$

for any $x \in \mathbb{R}$. These polynomials form an orthogonal basis in the Hilbert space $L^2$ of measurable functions $f : \mathbb{R} \to \mathbb{R}$ such that $\int f^2(x) \exp(-x^2/2)dx < \infty$ with inner product $\langle f, g \rangle = \int f(x)g(x) \exp(-x^2/2)dx$. The first few Hermite polynomials are $H_0(x) = 1, H_1(x) = x$, and $H_2(x) = x^2 - 1$.

**Condition 2.3.** *The activation function $\sigma : \mathbb{R} \to \mathbb{R}$ has the following Hermite expansion in $L^2$:*

$$\sigma(z) = \sum_{k=1}^\infty c_k H_k(z), \quad c_k = \frac{1}{k!} \mathbb{E}_{Z \sim \mathsf{N}(0,1)}[\sigma(Z) H_k(Z)].$$

*The coefficients satisfy $c_1 \neq 0$ and $c_k^2 k! \leq Ck^{-\frac{3}{2}-\omega}$ for some $C, \omega > 0$ and for all $k \geq 1$. Moreover, the first three derivatives of $\sigma$ almost surely exist and are bounded.*

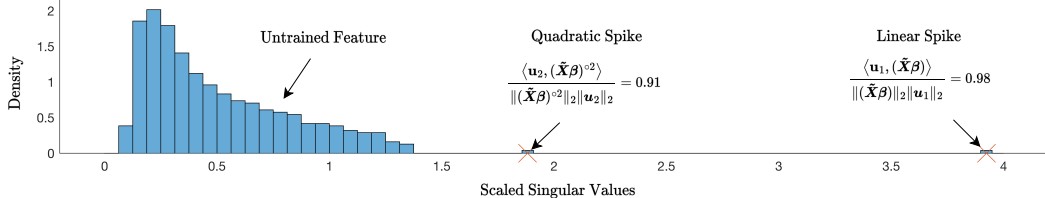

Figure 2: Histogram of the scaled singular values (divided by $\sqrt{n}$) of the feature matrix after the update with step size $\eta = n^{0.29}$ ($\ell = 2$). In this regime, two isolated spikes appear in the spectrum as stated in Theorem 3.3. The top two left singular vectors $\boldsymbol{u}_1$ and $\boldsymbol{u}_2$ are aligned with $\tilde{\mathbf{X}}\boldsymbol{\beta}$ and $(\tilde{\mathbf{X}}\boldsymbol{\beta})^{\circ 2}$, respectively. See Section 5 for the simulation details.

We remark that the above condition requires $c_0 = 0$, i.e., that $\mathbb{E}\sigma(Z) = 0$ for $Z \sim \mathsf{N}(0,1)$. This condition is in line with prior work in the area (e.g., Adlam & Pennington (2020a); Ba et al. (2022), etc.), and could be removed at the expense of more complicated formulas and theoretical analysis. The smoothness assumption on $\sigma$ is also in line with prior work in the area (see e.g., Hu & Lu (2023); Ba et al. (2022), etc.). Note that the above condition is satisfied by many popular activation functions (after shifting) such as the ReLU $\sigma(x) = \max\{x, 0\} - \frac{1}{\sqrt{2\pi}}$, hyperbolic tangent $\sigma(x) = \frac{e^x - e^{-x}}{e^x + e^{-x}}$, and sigmoid $\sigma(x) = \frac{1}{1+e^{-x}} - \frac{1}{2}$. We also make similar assumptions on the teacher activation:

**Condition 2.4.** *The teacher activation $\sigma_\star : \mathbb{R} \to \mathbb{R}$ has the following Hermite expansion in $L^2$:*

$$\sigma_\star(z) = \sum_{k=1}^{\infty} c_{\star,k} H_k(z), \quad c_{\star,k} = \frac{1}{k!} \mathbb{E}_{Z \sim \mathsf{N}(0,1)}[\sigma_\star(Z) H_k(Z)].$$

*Also, we define $c_\star = (\sum_{k=1}^{\infty} k! c_{\star,k}^2)^{\frac{1}{2}}$.*

## 3 ANALYSIS OF THE FEATURE MATRIX

The first step in analyzing the spectrum of the feature matrix $\mathbf{F}$ is to study the negative gradient $\mathbf{G}$. It is shown in (Ba et al., 2022, Proposition 2) that in operator norm, the matrix $\mathbf{G}$ can be approximated by the rank-one matrix $c_1 \boldsymbol{a}\boldsymbol{\beta}^\top$ with high probability, where the Hermite coefficient $c_1$ of the activation $\sigma$ is defined in Condition 2.3, and $\boldsymbol{\beta} = \frac{1}{n}\mathbf{X}^\top \boldsymbol{y} \in \mathbb{R}^d$. As the following proposition suggests, $\boldsymbol{\beta}$ can be understood as a noisy estimate of $\boldsymbol{\beta}_\star$ (see also Lemma K.1).

**Proposition 3.1.** *If Conditions 2.1–2.4 hold, then*

$$\frac{|\boldsymbol{\beta}_\star^\top \boldsymbol{\beta}|}{\|\boldsymbol{\beta}_\star\|_2 \|\boldsymbol{\beta}\|_2} \to_P \frac{|c_{\star,1}|}{\sqrt{c_{\star,1}^2 + \phi(c_\star^2 + \sigma_\varepsilon^2)}}.$$

In particular, if the number of samples used for the gradient update is very large; i.e., $\phi \to 0$, $\boldsymbol{\beta}$ will converge to being completely aligned to $\boldsymbol{\beta}_\star$.

Building on this result, we can prove the following rank-one approximation lemma. Note that the updated feature matrix can be written as $\mathbf{F} = \sigma(\tilde{\mathbf{X}}(\mathbf{W}_0 + \eta\mathbf{G})^\top)$ and terms of the form $(\tilde{\mathbf{X}}\mathbf{G}^\top)^{\circ k}$, $k \in \mathbb{N}$, will appear in polynomial and Taylor expansions of $\mathbf{F}$ around $\mathbf{F}_0$. In the following lemma, we show that for any fixed power $k$, these terms can be approximated by rank one terms.

**Lemma 3.2** (Rank-one approximation). *If Conditions 2.1–2.4 hold, then there exists $C > 0$ such that for $c_1$ from Condition 2.3, for any fixed $k \in \mathbb{N}$,*

$$\|(\tilde{\mathbf{X}}\mathbf{G}^\top)^{\circ k} - c_1^k(\tilde{\mathbf{X}}\boldsymbol{\beta})^{\circ k}(\boldsymbol{a}^{\circ k})^\top\|_{\mathrm{op}} \leq C^k n^{-\frac{k}{2}} \log^{2k} n$$

*with probability $1 - o(1)$.*

Next, we will show that after the gradient step, the spectrum of the feature matrix $\mathbf{F}$ will consist of a bulk of singular values that stick close together—given by the spectrum of the initial feature

matrix $\mathbf{F}_0 = \sigma(\tilde{\mathbf{X}}\mathbf{W}_0^\top)$—and $\ell$ separated spikes[1], where $\ell$ is an integer that depends on the step size used in the gradient update. Specifically, when the step size is $\eta \asymp n^\alpha$ with $\frac{\ell-1}{2\ell} < \alpha < \frac{\ell}{2\ell+2}$ for some $\ell \in \mathbb{N}$, the feature matrix $\mathbf{F}$ can be approximated in operator norm by the untrained features $\mathbf{F}_0 = \sigma(\tilde{\mathbf{X}}\mathbf{W}_0^\top)$ plus $\ell$ rank-one terms, where the left singular vectors of the rank-one terms are aligned with the non-linear features $\tilde{\mathbf{X}} \mapsto (\tilde{\mathbf{X}}\boldsymbol{\beta})^{\circ k}$, for $k \in [\ell]$. See Figure 2.

**Theorem 3.3** (Spectrum of feature matrix). *Let $\eta \asymp n^\alpha$ with $\frac{\ell-1}{2\ell} < \alpha < \frac{\ell}{2\ell+2}$ for some $\ell \in \mathbb{N}$. If Conditions 2.1-2.4 hold, then for $c_k$ from Condition 2.3 and $\mathbf{F}_0 = \sigma(\tilde{\mathbf{X}}\mathbf{W}_0^\top)$,*

$$\mathbf{F} = \mathbf{F}_\ell + \boldsymbol{\Delta}, \qquad \text{with} \qquad \mathbf{F}_\ell := \mathbf{F}_0 + \sum_{k=1}^\ell c_1^k c_k \eta^k (\tilde{\mathbf{X}}\boldsymbol{\beta})^{\circ k} (\boldsymbol{a}^{\circ k})^\top, \qquad (3)$$

*where $\|\boldsymbol{\Delta}\|_{\mathrm{op}} = o(\sqrt{n})$ with probability $1 - o(1)$.*

To understand $(\tilde{\mathbf{X}}\boldsymbol{\beta})^{\circ k}(\boldsymbol{a}^{\circ k})^\top$, notice that for a datapoint with features $\tilde{\boldsymbol{x}}_i$, the activation of each neuron is proportional to the polynomial feature $(\tilde{\boldsymbol{x}}_i^\top \boldsymbol{\beta})^k$, with coefficients given by $\boldsymbol{a}^{\circ k}$ for the neurons. The spectrum of the initial feature matrix $\mathbf{F}_0$ is fully characterized in Pennington & Worah (2017); Benigni & Péché (2021; 2022); Louart et al. (2018); Fan & Wang (2020), and its operator norm is known to be $\Theta_{\mathbb{P}}(\sqrt{n})$. Moreover, it follows from the proof that the operator norm of each of the terms $c_1^k c_k \eta^k (\tilde{\mathbf{X}}\boldsymbol{\beta})^{\circ k}(\boldsymbol{a}^{\circ k})^\top$, $k \in [l]$ is with high probability of order larger than $\sqrt{n}$. Thus, Theorem 3.3 identifies the spikes in the spectrum of the feature matrix.

**Proof Idea.** We approximate the feature matrix $\mathbf{F} = \sigma(\tilde{\mathbf{X}}(\mathbf{W}_0 + \eta\mathbf{G})^\top)$ by a polynomial using its Hermite expansion. Next, we use the binomial expansion and apply Lemma 3.2 to approximate $(\tilde{\mathbf{X}}\mathbf{G}^\top)^{\circ k}$ by $c_1^k (\tilde{\mathbf{X}}\boldsymbol{\beta})^{\circ k}(\boldsymbol{a}^{\circ k})^\top$, for all $k$. Then, spike terms with $k \geq \ell + 1$ are negligible since we can show that their norm is $O_{\mathbb{P}}(n^{k\alpha + \frac{1}{2} - \frac{k-1}{2}}) = o_{\mathbb{P}}(\sqrt{n})$.

The special case where $\alpha = 0$ is discussed in (Ba et al., 2022, Section 3), which focuses on the spectrum of the updated weight matrix $\mathbf{W} = \mathbf{W}_0 + \eta\mathbf{G}$. However, here we study the updated feature matrix $\mathbf{F} = \sigma(\tilde{\mathbf{X}}(\mathbf{W}_0 + \eta\mathbf{G})^\top)$ because that is more directly related to the learning problem—as we will discuss in the consequences for the training and test risk below.

In the following theorem, we argue that the subspace spanned by the non-linear features $\{\sigma(\tilde{\mathbf{X}}\boldsymbol{w}_i)\}_{i \in [N]}$ can be approximated by the subspace spanned by the monomials $\{(\tilde{\mathbf{X}}\boldsymbol{\beta})^{\circ k}\}_{k \in [\ell]}$. For two $\ell$-dimensional subspaces $\mathcal{U}_1, \mathcal{U}_2 \subseteq \mathbb{R}^n$, with orthonormal bases $\mathbf{U}_1, \mathbf{U}_2 \in \mathbb{R}^{n \times \ell}$, recall the principal angle distance between $\mathcal{U}_1, \mathcal{U}_2$ defined by $d(\mathcal{U}_1, \mathcal{U}_2) = \min_{\mathbf{Q}} \|\mathbf{U}_1 - \mathbf{U}_2\mathbf{Q}\|_{\mathrm{op}}$, where the minimum is over $\ell \times \ell$ orthogonal matrices (Stewart & Sun, 1990). This definition is invariant to the choice of the orthonormal bases $\mathbf{U}_1, \mathbf{U}_2$.

**Theorem 3.4.** *Let $\mathcal{F}_\ell$ be the $\ell$-dimensional subspace of $\mathbb{R}^n$ spanned by top-$\ell$ left singular vectors (principal components) of $\mathbf{F}$. Under the conditions of Theorem 3.3, we have*

$$d(\mathcal{F}_\ell, \mathrm{span}\{(\tilde{\mathbf{X}}\boldsymbol{\beta})^{\circ k}\}_{k \in [\ell]}) \to_P 0.$$

This result shows that after one step of gradient descent with step size $\eta \asymp n^\alpha$ with $\frac{\ell-1}{2\ell} < \alpha < \frac{\ell}{2\ell+2}$, the subspace of the top-$\ell$ left singular vectors carries information from the polynomials $\{(\tilde{\mathbf{X}}\boldsymbol{\beta})^{\circ k}\}_{k \in [\ell]}$. Also, recall that by Proposition 3.1, the vector $\boldsymbol{\beta}$ is aligned with $\boldsymbol{\beta}_\star$. Hence, it is shown that $\mathcal{F}_\ell$ carries information from the first $\ell$ polynomial components of the teacher function.

**Proof Idea.** We use Wedin's theorem (Wedin, 1972) to characterize the distance between the left singular vector space of $\sum_{k=1}^\ell c_1^k c_k \eta^k (\tilde{\mathbf{X}}\boldsymbol{\beta})^{\circ k}(a^{\circ k})^\top$ and that of $\mathbf{F}$. Here, we consider the matrix $\mathbf{F}_0 + \boldsymbol{\Delta}$ as the perturbation term.

## 4 LEARNING HIGHER-DEGREE POLYNOMIALS

In the previous section, we studied the feature matrix $\mathbf{F}$ and showed that when $\eta \asymp n^\alpha$ with $\frac{\ell-1}{2\ell} < \alpha < \frac{\ell}{2\ell+2}$, it can be approximated by $\mathbf{F}_0 = \sigma(\tilde{\mathbf{X}}\mathbf{W}_0^\top)$ plus $\ell$ rank-one or spike terms. We

---

[1]Using terminology from random matrix theory (Bai & Silverstein, 2010; Yao et al., 2015).

also saw that the left singular vectors of the spike terms are aligned with the non-linear functions $\tilde{\mathbf{X}} \mapsto (\tilde{\mathbf{X}}\boldsymbol{\beta})^{\circ k}$. Intuitively, this result suggests that after the gradient update, the trained weights are becoming aligned with the teacher model and we should expect the ridge regression estimator on the learned features to achieve better performance. In particular, when $\alpha > 0$, we expect the ridge regression estimator to—partially—capture the non-linear part of the teacher function. This is impossible for $\eta = O(1)$ (Ba et al., 2022) or $\eta = 0$ (Hu & Lu, 2023; Mei & Montanari, 2022).

In this section, we aim to make this intuition rigorous and show that the spikes in the feature matrix lead to a decrease in the loss achieved by the estimator. Moreover, for large enough step sizes, the model can fit non-linear components of the teacher function. For this, we first need to prove *equivalence theorems* showing that instead of the true feature matrix $\mathbf{F}$, the approximations from Theorem 3.3 can be used to compute error terms (i.e., the effect of $\boldsymbol{\Delta}$ on the error is negligible).

### 4.1 Equivalence Theorems

Given a regularization parameter $\lambda > 0$, recalling the ridge estimator $\hat{\boldsymbol{a}}(\mathbf{F})$ from equation 2,

we define the training loss

$$\mathcal{L}_{\text{tr}}(\mathbf{F}) = \frac{1}{n}\|\tilde{\boldsymbol{y}} - \mathbf{F}\hat{\boldsymbol{a}}(\mathbf{F})\|_2^2 + \lambda\|\hat{\boldsymbol{a}}(\mathbf{F})\|_2^2.$$

In the next theorem, we show that when $\eta \asymp n^\alpha$ with $\frac{\ell-1}{2\ell} < \alpha < \frac{\ell}{2\ell+2}$, the training loss $\mathcal{L}_{\text{tr}}(\mathbf{F})$ can be approximated with negligible error by $\mathcal{L}_{\text{tr}}(\mathbf{F}_\ell)$.

In other words, the approximation of the feature matrix in Theorem 3.3 can be used to derive the asymptotics of the training loss.

**Theorem 4.1** (Training loss equivalence). *Let $\eta \asymp n^\alpha$ with $\frac{\ell-1}{2\ell} < \alpha < \frac{\ell}{2\ell+2}$ for some $\ell \in \mathbb{N}$ and recall $\mathbf{F}_\ell$ from equation 3. If Conditions 2.1-2.4 hold, then for any fixed $\lambda > 0$, we have $\mathcal{L}_{\text{tr}}(\mathbf{F}) - \mathcal{L}_{\text{tr}}(\mathbf{F}_\ell) = o(1)$, with probability $1 - o(1)$.*

Similar equivalence results can also be proved for the test risk, i.e., the average test loss. For any $\boldsymbol{a} \in \mathbb{R}^N$, we define the test risk of $\boldsymbol{a}$ as $\mathcal{L}_{\text{te}}(\boldsymbol{a}) = \mathbb{E}_{\boldsymbol{f},y}(y - \boldsymbol{f}^\top \boldsymbol{a})^2$, in which the expectation is taken over $(\boldsymbol{x}, y)$ where $\boldsymbol{f} = \sigma(\mathbf{W}\boldsymbol{x})$ with $\boldsymbol{x} \sim \mathsf{N}(0, \mathbf{I}_d)$ and $y = f_\star(\boldsymbol{x}) + \varepsilon$ with $\varepsilon \sim \mathsf{N}(0, \sigma_\varepsilon^2)$. The next theorem shows that one can also use the approximation of the feature matrix from Theorem 3.3 to derive the asymptotics of the test risk.

**Theorem 4.2** (Test risk equivalence). *Let $\eta \asymp n^\alpha$ with $\frac{\ell-1}{2\ell} < \alpha < \frac{\ell}{2\ell+2}$ for some $\ell \in \mathbb{N}$ and $\mathbf{F}_\ell$ be defined as in equation 3. If Conditions 2.1-2.4 hold, then for any $\lambda > 0$, if $\mathcal{L}_{\text{te}}(\hat{\boldsymbol{a}}(\mathbf{F})) \to_P \mathcal{L}_{\mathbf{F}}$ and $\mathcal{L}_{\text{te}}(\hat{\boldsymbol{a}}(\mathbf{F}_\ell)) \to_P \mathcal{L}_{\mathbf{F}_\ell}$, we have $\mathcal{L}_{\mathbf{F}} = \mathcal{L}_{\mathbf{F}_\ell}$.*

**Proof Idea.** For theorem 4.1 we argue that the error introduced by swapping the feature matrix $\mathbf{F}$ with $\mathbf{F}_\ell$ is small, using a *free-energy trick* (Abbasi et al., 2019; Hu & Lu, 2023; Hassani & Javanmard, 2022). We first extend Theorem 4.1 and show that for any $\lambda, \zeta > 0$, the minima over $\boldsymbol{a}$ of

$$\mathcal{R}_\zeta(\boldsymbol{a}, \bar{\mathbf{F}}) = \frac{1}{n}\|\tilde{\boldsymbol{y}} - \bar{\mathbf{F}}\boldsymbol{a}\|_2^2 + \lambda\|\boldsymbol{a}\|_2^2 + \zeta\mathcal{L}_{\text{te}}(\boldsymbol{a}),$$

for $\bar{\mathbf{F}} = \mathbf{F}$ and $\bar{\mathbf{F}} = \mathbf{F}_\ell$ are close. Then, we use this to argue that the limiting test loss are also close.

With Theorem 4.1 and 4.2 in hand, for $\eta \asymp n^\alpha$, we can use the approximation $\mathbf{F}_\ell$—with the appropriate $\ell$—of the feature matrix $\mathbf{F}$ to analyze the train loss and the test risk.

### 4.2 Analysis of Training Loss

In this section, we quantify the discrepancy between the training loss of the ridge estimator trained on the new—learned—feature map $\mathbf{F}$ and the same ridge estimator trained on the unlearned feature map $\mathbf{F}_0$. We will do this for the step size $\eta \asymp n^\alpha$ with $\frac{\ell-1}{2\ell} < \alpha < \frac{\ell}{2\ell+2}$ for various $\ell \in \mathbb{N}$.

Our results depend on the limits of traces of the matrices $(\mathbf{F}_0\mathbf{F}_0^\top + \lambda n \mathbf{I}_n)^{-1}$ and $\tilde{\mathbf{X}}^\top(\mathbf{F}_0\mathbf{F}_0^\top + \lambda n \mathbf{I}_n)^{-1}\tilde{\mathbf{X}}$. These limits have been determined in Adlam et al. (2022); Adlam & Pennington (2020a),

see also Pennington & Worah (2017); Péché (2019), and depend on the values $m_1, m_2 > 0$, which are the unique solutions of the following system of coupled equations, for $\lambda > 0$:

$$\begin{cases} \phi \left( m_1 - m_2 \right) \left( c_{>1}^2 m_1 + c_1^2 m_2 \right) + c_1^2 m_1 m_2 \left( \lambda \frac{\psi}{\phi} m_1 - 1 \right) = 0, \\ \frac{\phi}{\psi} \left( c_1^2 m_1 m_2 + \phi \left( m_2 - m_1 \right) \right) + c_1^2 m_1 m_2 \left( \lambda \frac{\psi}{\phi} m_1 - 1 \right) = 0, \end{cases} \quad (4)$$

where $c_{>1} = \left( \sum_{k=2}^{\infty} k! c_k^2 \right)^{1/2}$. For instance, we leverage that $\lim_{d,n,N \to \infty} \text{tr}(\tilde{\mathbf{X}}^\top (\mathbf{F}_0 \mathbf{F}_0^\top + \lambda n \mathbf{I}_n)^{-1} \tilde{\mathbf{X}})/d = \psi m_2/\phi > 0$ and $\lim_{d,n,N \to \infty} \text{tr}((\mathbf{F}_0 \mathbf{F}_0^\top + \lambda n \mathbf{I}_n)^{-1}) = \psi m_1/\phi > 0$.

See Lemma K.4 and its proof for more details. For instance, as argued in Pennington & Worah (2017); Adlam et al. (2022) these can be reduced to a quartic equation for $m_1$ and are convenient to solve numerically. However, the existence of these limits does not imply our results; on the contrary, the proofs of our results require extensive additional calculations and several novel ideas. Moreover, our results also rely on the following Gaussian equivalence conjecture for the untrained feature matrix, which is commonly used in the theory of random features models. See Section J for related work and further discussion; in particular Gaussian Equivalence has been broadly supported by prior theoretical and empirical results.

**Conjecture 4.3** (Gaussian Equivalence). *The limiting behavior of the training error is unchanged if we replace the untrained feature matrix $\mathbf{F}_0 = \sigma(\tilde{\mathbf{X}} \mathbf{W}_0^\top)$ with $\mathbf{F}_0 = c_1 \tilde{\mathbf{X}} \mathbf{W}_0^\top + c_{>1} \mathbf{Z}$, where $\mathbf{Z} \in \mathbb{R}^{n \times d}$ is an independent random matrix with i.i.d. $\mathsf{N}(0,1)$ entries. Specifically, the limiting behavior of the quantities listed in Section J is unchanged.*

**Theorem 4.4.** *If Conditions 2.1-2.4 are satisfied, and the Gaussian equivalence conjecture 4.3 hold, while we also have $c_1, \cdots, c_\ell \neq 0$, and $\eta \asymp n^\alpha$ with $\frac{\ell-1}{2\ell} < \alpha < \frac{\ell}{2\ell+2}$, then for the learned feature map $\mathbf{F}$ and the untrained feature map $\mathbf{F}_0$, we have $\mathcal{L}_{\text{tr}}(\mathbf{F}_0) - \mathcal{L}_{\text{tr}}(\mathbf{F}) \to_P \Delta_\ell > 0$, where the explicit expression for $\Delta_\ell$ can be found in Section L.*

The expression for $\Delta_\ell$ is complex and given in Section L due to space limitations. For a better understanding of Theorem 4.4, we consider two specific cases of $\ell = 1$ and $\ell = 2$.

**Corollary 4.5.** *Under the assumptions of Theorem 4.4, for $\ell = 1$, we have*

$$\mathcal{L}_{\text{tr}}(\mathbf{F}_0) - \mathcal{L}_{\text{tr}}(\mathbf{F}) \to_P \Delta_1 := \frac{\psi \lambda c_{\star,1}^4 m_2}{\phi[c_{\star,1}^2 + \phi(c_\star^2 + \sigma_\varepsilon^2)]} > 0. \quad (5)$$

*Similarly, for $\ell = 2$, we have*

$$\mathcal{L}_{\text{tr}}(\mathbf{F}_0) - \mathcal{L}_{\text{tr}}(\mathbf{F}) \to_P \Delta_2 := \Delta_1 + \frac{4\psi \lambda c_{\star,1}^4 c_{\star,2}^2 m_1}{3\phi[\phi(c_\star^2 + \sigma_\varepsilon^2) + c_{\star,1}^2]^2} > 0. \quad (6)$$

The above result confirms our intuition that training the first-layer parameters improves the performance of the trained model. For example, when $\ell = 1$, the improvement in the loss is increasing in the strength of the linear component $c_{\star,1}$ keeping the signal strength $c_\star$ fixed; and not so for the strength of the non-linear component $c_{\star,>1}^2 = c_\star^2 - c_{\star,1}^2$. When we further increase the step size to the $\ell = 2$ regime, the loss of the trained model will drop by an additional positive value, depending on the strength $c_{\star,2}$ of the quadratic signal, which supports our claim that the quadratic component of the target function is also being learned.

Given $\ell \in \mathbb{N}$, the loss of the trained model is asymptotically constant for all $\eta = cn^\alpha$ with $\frac{\ell-1}{2\ell} < \alpha < \frac{\ell}{2\ell+2}$ and $c \in \mathbb{R}$. There are sharp jumps at the edges between regimes of $\alpha$, whose size is precisely characterized above. See Figure 3 (**Right**).

**Proof Idea.** We first show that $\mathcal{L}_{\text{tr}}(\mathbf{F}) = \lambda \tilde{\mathbf{y}}^\top (\mathbf{F}\mathbf{F}^\top + \lambda n \mathbf{I}_n)^{-1} \tilde{\mathbf{y}}$. Then using Theorem 4.1 and by application of the Woodbury formula, we decompose the matrix $\bar{\mathbf{R}} = (\mathbf{F}\mathbf{F}^\top + \lambda n \mathbf{I}_n)^{-1}$ as $\bar{\mathbf{R}}_0 = (\mathbf{F}_0 \mathbf{F}_0^\top + \lambda n \mathbf{I}_n)^{-1}$ plus rank-one terms involving $\bar{\mathbf{R}}_0$ and the non-linear spikes from Theorem 3.3. Then, we show that the interactions between the first $\ell$ components of $\tilde{\mathbf{y}}$ and the terms involving the non-linear spikes in the expansion of $\bar{\mathbf{R}}$ will result in non-vanishing terms corresponding to learning different components of the target function $f_\star$.

## 5 NUMERICAL SIMULATIONS

To support and illustrate our theoretical results, we present some numerical simulations. We use the shifted ReLU activation $\sigma(x) = \max(x, 0) - 1/\sqrt{2\pi}$, $n = 1000$, $N = 500$, $d = 300$, and the regularization parameter $\lambda = 0.01$.

**Singular Value Spectrum of F.** We let the the teacher function $f_\star(\boldsymbol{x}) = H_1(\boldsymbol{\beta}_\star^\top \boldsymbol{x}) + H_2(\boldsymbol{\beta}_\star^\top \boldsymbol{x})$ be, set the noise variance $\sigma_\varepsilon^2 = 0.5$, and the step size to $\eta = n^{0.29}$, so $\ell = 2$. We plot the histogram of singular values of the updated feature matrix $\mathbf{F}$. In Figure 2, we see two spikes corresponding to $\tilde{\mathbf{X}}\boldsymbol{\beta}, (\tilde{\mathbf{X}}\boldsymbol{\beta})^{\circ 2}$ as suggested by Theorem 3.3 and 3.4. Since $f_\star$ has a linear component $H_1$ and a quadratic component $H_2$, these spikes will lead to feature learning.

**Quadratic Feature Learning.** To support the findings of Corollary 4.5 for $\ell = 2$, we consider the following two settings:

$$\textbf{Setting 1}: y = H_1(\boldsymbol{\beta}_\star^\top \boldsymbol{x}) + \varepsilon, \quad \varepsilon \sim \mathsf{N}(0, 1), \quad \textbf{Setting 2}: y = H_1(\boldsymbol{\beta}_\star^\top \boldsymbol{x}) + \frac{1}{\sqrt{2}} H_2(\boldsymbol{\beta}_\star^\top \boldsymbol{x}).$$

Note that $c_{\star,1}$ and $c_\star + \sigma_\varepsilon^2$ are same in these two settings. This ensures that the improvement due to learning the linear component is the same. We plot the training and test errors of the two-layer neural networks trained with the procedure described in Section 2.1 as functions of $\log(\eta)/\log(n)$. In Figure 3 (**Left**), we see that the errors decrease in the range $\log(\eta)/\log(n) \in (0, \frac{1}{4})$ as the model learns the linear component $H_1(\boldsymbol{\beta}_\star^\top \boldsymbol{x})$. In the range $\log(\eta)/\log(n) \in (\frac{1}{4}, \frac{1}{3})$, the model starts to learn the quadratic feature. However since the quadratic feature is not present in Setting 1, the errors under the two settings diverge. These results are consistent with Corollary 4.5.

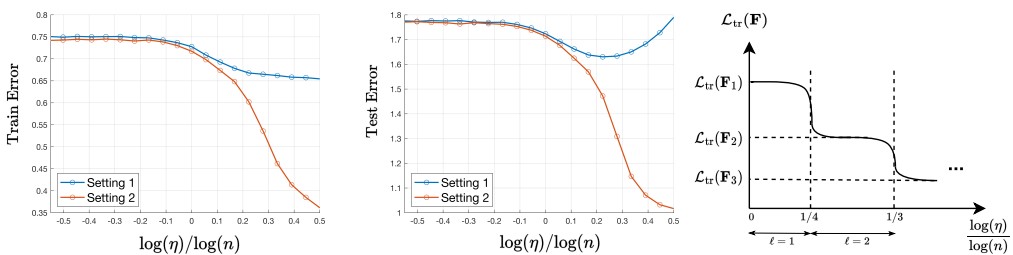

Figure 3: **(Left, Middle)** Training and test errors after one gradient as functions of $\log(\eta)/\log(n)$. **(Right)** Theoretical training error curve as a function of $\log(\eta)/\log(n)$.

## 6 CONCLUSION

In this work, we study feature learning in two-layer neural networks under one-step gradient descent with the step size $\eta \asymp n^\alpha, \alpha \in (0, \frac{1}{2})$. We show that the singular value spectrum of the updated feature matrix exhibits different behaviors for different ranges of $\alpha$. Specifically, if $\alpha \in (\frac{\ell-1}{2\ell}, \frac{\ell}{2\ell+2})$, then the gradient update will add $\ell$ separated singular values to the initial feature matrix spectrum. We then derive the improvement in the loss in the proportional limit and show that non-linear features can be learned in certain examples.

**Limitations and Future Work.** First, our analysis requires that the teacher activation function $\sigma_\star$ has information exponent $\kappa = 1$. This assumption is necessary to learn $\boldsymbol{\beta}_\star$ with one step of gradient and with the sample size $n \asymp d$. We believe that learning $\boldsymbol{\beta}_\star$ from a teacher activation with higher information exponent will require either multiple steps of gradient or a larger sample size. Second, we only derived the limiting training loss in our result. This is mainly because the test error does not allow a simple expression such as Lemma K.2, and deriving its asymptotics would require much more laborious calculation. We hope to address this issue in future work. Third, we only study the problem when $\eta \asymp n^\alpha$ with $\alpha \in (\frac{\ell-1}{2\ell}, \frac{\ell}{2\ell+2})$. The case where $\eta \asymp n^{\frac{\ell-1}{2\ell}}$ is an interesting problem and is left as future work. Finally, our results in Section 4.2 rely on a Gaussian equivalence conjecture for the untrained features $\mathbf{F}_0$. The Gaussian equivalence conjecture we use, despite being related to the results discussed in Section J, does not directly follow from prior work.

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
