# A ADDITIONAL RELATED WORK

**Theory of Shallow Neural Networks.** Two-layer neural networks have been studied extensively in the mean-field regime (see e.g., Mei et al. (2018; 2019); Sirignano & Spiliopoulos (2020), etc.), and the neural tangent kernel (NTK) regime (see e.g., Jacot et al. (2018); Lee et al. (2019); Huang & Yau (2020), etc.). However, these results often require the neural net to have an extremely large width.

In particular, in the NTK regime, this large width will result in features not evolving over the course of training. Ghorbani et al. (2021) show that for NTKs, with a sample size linear in size of the input, non-linear functions cannot be learned. See also (Misiakiewicz, 2022; Xiao et al., 2022; Lu & Yau, 2022). Goel & Klivans (2019) provide a polynomial time algorithm that learns neural networks with two non-linear layers. Our setting is different because we do not apply a non-linear activation after the second layer. Chen et al. (2022) show that learning two-hidden-layer neural networks from noise-free Gaussian data requires superpolynomially many statistical queries.

**Feature Learning.** Damian et al. (2022) study the problem of learning polynomials with only a few relevant directions and show a sample complexity improvement over kernel methods. Nichani et al. (2023) provide theoretical evidence that three-layer neural networks have provably richer feature learning capabilities than their two-layer counterparts. Zhenmei et al. (2022) show that neural networks trained by gradient descent can succeed on problems where the labels are determined by a set of class-relevant patterns and if these patterns are removed, no polynomial algorithm in the Statistical Query model can learn even weakly.

# B ADDITIONAL NOTATION AND TERMINOLOGY

In the appendix, we use the following additional notations. We let $\mathbb{N}_0 = \{0, 1, 2, \ldots\}$ be the set of non-negative integers. For a set $X$ and $x_1, x_2 \in X$, $\delta_{x_1, x_2}$ is the Kronecker delta, which equals unity if $x_1 = x_2$, and is zero otherwise. We use $\tilde{O}(\cdot)$ for the standard big-O notation up to logarithmic factors in $n$. For a positive integer $k$, $k!!$ is the product of all the positive integers up to $n$ with the same parity as $n$. For two random quantities $X, Y$, $X \perp\!\!\!\perp Y$ denotes that $X$ is independent of $Y$. By orderwise analysis, we mean bounding a term by the triangle inequality and the inequality $\|Ab\|_2 \leq \|A\|_{\mathrm{op}}\|b\|_2$ for a conformable matrix-vector pair $A, b$, to reduce it to operator norms of matrices and Euclidean norms of vectors, and then use simple bounds for those quantities. Constants such as $C, c'$, etc., can change from line to line unless specified otherwise. For two random quantities $A, B$, $A =_d B$ denotes that $A$ and $B$ have the same distribution. Limits of random variables are understood in probability. For two matrices $\mathbf{A}, \mathbf{B}$ with equal shape, we write $\mathbf{A} \circ \mathbf{B}$ to denote their entry-wise (Hadamard) product.

We denote $\mathbf{X}\boldsymbol{\beta} = \boldsymbol{\theta}$, $\tilde{\mathbf{X}}\boldsymbol{\beta} = \tilde{\boldsymbol{\theta}}$, $\mathbf{X}\boldsymbol{\beta}_\star = \boldsymbol{\theta}_\star$, and $\tilde{\mathbf{X}}\boldsymbol{\beta}_\star = \tilde{\boldsymbol{\theta}}_\star$. We also define $\bar{\mathbf{R}}_0 = (\mathbf{F}_0\mathbf{F}_0^\top + \lambda n\mathbf{I}_n)^{-1}$ and $\mathbf{R}_0 = (\mathbf{F}_0^\top\mathbf{F}_0 + \lambda n\mathbf{I}_N)^{-1}$.

# C BASIC LEMMAS

**Lemma C.1** (Orthogonality of Hermite polynomials). *Let $(Z_1, Z_2)$ be jointly Gaussian with $\mathbb{E}[Z_1] = \mathbb{E}[Z_2] = 0$, $\mathbb{E}[Z_1^2] = \mathbb{E}[Z_2^2] = 1$, and $\mathbb{E}[Z_1 Z_2] = \rho$. Then for any $k_1, k_2 \in \mathbb{N}_0$,*

$$\mathbb{E}[H_{k_1}(Z_1)H_{k_2}(Z_2)] = k_1! \rho^{k_1} \delta_{k_1, k_2}.$$

*In particular, if for some positive integer $d$, $\mathbf{Z} \sim \mathsf{N}(0, \mathbf{I}_d)$, and if $\boldsymbol{a}, \boldsymbol{b} \in \mathbb{S}^{d-1}$, then*

$$\mathbb{E}[H_{k_1}(\boldsymbol{a}^\top\mathbf{Z})H_{k_2}(\boldsymbol{b}^\top\mathbf{Z})] = k_1! (\boldsymbol{a}^\top\boldsymbol{b})^{k_1} \delta_{k_1, k_2}.$$

*Proof.* See (O'Donnell, 2014, Chapter 11.2). □

**Lemma C.2** (Taylor expansion of Hermite polynomials). *For any $k \in \mathbb{N}_0$ and $x, y \in \mathbb{R}$,*

$$H_k(x + y) = \sum_{j=0}^{k} \binom{k}{j} x^j H_{k-j}(y).$$

*Proof.* Note that $\frac{d}{dx}H_k(x) = kH_{k-1}(x)$ (Abramowitz & Stegun, 1968, Equation 22.8.8) and thus $\frac{d^j}{dx^j}H_k(x) = \frac{k!}{(k-j)!}H_{k-j}(x)$. By Taylor expanding $H_k(x+y)$ at $y$, we find

$$H_k(x+y) = \sum_{j=0}^{k} \frac{x^j}{j!}\frac{d^j}{dy^j}H_k(y) = \sum_{j=0}^{k}\binom{k}{j}x^j H_{k-j}(y).$$

$\square$

The following Lemma, proved in Section N.1, provides several bounds used in the proofs.

**Lemma C.3.** *Under Conditions 2.1-2.4, there exists $C > 0$ such that the following holds with probability $1 - o(1)$.*

(a) $M_{\boldsymbol{a}} := \max_{1 \le i \le N}|a_i| \le Cn^{-\frac{1}{2}}\log^{\frac{1}{2}} n$,

(b) $M_{\boldsymbol{\beta}} := \max_{1 \le i \le n}|\langle \tilde{\boldsymbol{x}}_i, \boldsymbol{\beta}\rangle| \le C\log^{\frac{1}{2}} n$,

(c) $M_{\mathbf{W}_0} := \sup_{k \ge 1}\|(\mathbf{W}_0\mathbf{W}_0^\top)^{\circ k}\|_{\mathrm{op}} \le C$,

(d) $\|\tilde{\mathbf{X}}\|_{\mathrm{op}} \le C\sqrt{n}$.

## D   PROOF OF PROPOSITION 3.1

*Proof.* By Lemma K.1 with $\mathbf{v} = \boldsymbol{\beta}_\star$ and $\mathbf{D} = \mathbf{I}_d$, we have

$$\boldsymbol{\beta}_\star^\top\boldsymbol{\beta} \to_P c_{\star,1}\|\boldsymbol{\beta}_\star\|_2^2 = c_{\star,1},$$
$$\|\boldsymbol{\beta}\|_2^2 = \boldsymbol{\beta}^\top\boldsymbol{\beta} \to_P \phi(c_\star^2 + \sigma_\varepsilon^2) + c_{\star,1}^2\boldsymbol{\beta}_\star^\top\boldsymbol{\beta}_\star = c_{\star,1}^2 + \phi(c_\star^2 + \sigma_\varepsilon^2).$$

By the continuous mapping theorem, we conclude

$$\frac{|\boldsymbol{\beta}_\star^\top\boldsymbol{\beta}|}{\|\boldsymbol{\beta}_\star\|_2\|\boldsymbol{\beta}\|_2} \to_P \frac{|c_{\star,1}|}{\sqrt{c_{\star,1}^2 + \phi(c_\star^2 + \sigma_\varepsilon^2)}}.$$

$\square$

## E   PROOF OF LEMMA 3.2

*Proof.* For $k = 1$, we have by (Ba et al., 2022, Proposition 2)—substituting our $c_1$ for their $\mu_1$ and using that $\boldsymbol{\beta} = \frac{1}{n}\mathbf{X}^\top\boldsymbol{y}$; as well as noting that by the discussion below (Ba et al., 2022, Proposition 2), $\|\mathbf{G}\|_{\mathrm{op}} = O_{\mathbb{P}}(1)$—and by Lemma C.3 (d), that with probability $1 - o(1)$,

$$\|\tilde{\mathbf{X}}\mathbf{G}^\top - c_1\tilde{\mathbf{X}}\boldsymbol{\beta}\boldsymbol{a}^\top\|_{\mathrm{op}} = O(n^{-\frac{1}{2}}\log^2 n). \tag{7}$$

For $k \ge 2$, expanding $(\tilde{\mathbf{X}}\mathbf{G}^\top)^{\circ k} = (\tilde{\mathbf{X}}\mathbf{G}^\top - c_1\tilde{\mathbf{X}}\boldsymbol{\beta}\boldsymbol{a}^\top + c_1\tilde{\mathbf{X}}\boldsymbol{\beta}\boldsymbol{a}^\top)^{\circ k}$ using the binomial formula, we have

$$(\tilde{\mathbf{X}}\mathbf{G}^\top)^{\circ k} - c_1^k(\tilde{\mathbf{X}}\boldsymbol{\beta})^{\circ k}(\boldsymbol{a}^{\circ k})^\top = \sum_{j=1}^{k}\binom{k}{j}(\tilde{\mathbf{X}}\mathbf{G}^\top - c_1\tilde{\mathbf{X}}\boldsymbol{\beta}\boldsymbol{a}^\top)^{\circ j} \circ (c_1\tilde{\mathbf{X}}\boldsymbol{\beta}\boldsymbol{a}^\top)^{\circ(k-j)}$$

$$= \sum_{j=1}^{k}\binom{k}{j}c_1^{k-j}\mathrm{diag}(\tilde{\mathbf{X}}\boldsymbol{\beta})^{k-j}(\tilde{\mathbf{X}}\mathbf{G}^\top - c_1\tilde{\mathbf{X}}\boldsymbol{\beta}\boldsymbol{a}^\top)^{\circ j}\mathrm{diag}(\boldsymbol{a})^{k-j}.$$

Recalling $M_{\boldsymbol{a}}, M_{\boldsymbol{\beta}}$ from Lemma C.3, and using that

$$\|(\tilde{\mathbf{X}}\mathbf{G}^\top - c_1\tilde{\mathbf{X}}\boldsymbol{\beta}\boldsymbol{a}^\top)^{\circ j}\|_{\mathrm{op}} \le \|\tilde{\mathbf{X}}\mathbf{G}^\top - c_1\tilde{\mathbf{X}}\boldsymbol{\beta}\boldsymbol{a}^\top\|_{\mathrm{op}}^j$$

(see e.g., (Bai & Silverstein, 2010, Corollary A.21)), we have

$$\| \operatorname{diag}(\tilde{\mathbf{X}}\boldsymbol{\beta})^{k-j}(\tilde{\mathbf{X}}\mathbf{G}^{\top} - c_1\tilde{\mathbf{X}}\boldsymbol{\beta}\boldsymbol{a}^{\top})^{\circ j} \operatorname{diag}(\boldsymbol{a})^{k-j}\|_{\mathrm{op}}$$
$$\leq \| \operatorname{diag}(\tilde{\mathbf{X}}\boldsymbol{\beta})^{k-j}\|_{\mathrm{op}}\|(\tilde{\mathbf{X}}\mathbf{G}^{\top} - c_1\tilde{\mathbf{X}}\boldsymbol{\beta}\boldsymbol{a}^{\top})^{\circ j}\|_{\mathrm{op}}\| \operatorname{diag}(\boldsymbol{a})^{k-j}\|_{\mathrm{op}}$$
$$\leq (M_{\boldsymbol{a}}M_{\boldsymbol{\beta}})^{k-j}\|\tilde{\mathbf{X}}\mathbf{G}^{\top} - c_1\tilde{\mathbf{X}}\boldsymbol{\beta}\boldsymbol{a}^{\top}\|_{\mathrm{op}}^{j}.$$

Hence, by the triangle inequality,

$$\|(\tilde{\mathbf{X}}\mathbf{G}^{\top})^{\circ k} - c_1^k(\tilde{\mathbf{X}}\boldsymbol{\beta})^{\circ k}(\boldsymbol{a}^{\circ k})^{\top}\|_{\mathrm{op}} \leq \sum_{j=1}^{k}\binom{k}{j}(c_1 M_{\boldsymbol{a}}M_{\boldsymbol{\beta}})^{k-j}\|\tilde{\mathbf{X}}\mathbf{G}^{\top} - c_1\tilde{\mathbf{X}}\boldsymbol{\beta}\boldsymbol{a}^{\top}\|_{\mathrm{op}}^{j}.$$

By Lemma C.3 (a), (b) and equation 7, there exists $C > 0$ such that for any $k \in \mathbb{N}$,

$$\sum_{j=1}^{k}\binom{k}{j}(c_1 M_{\boldsymbol{a}}M_{\boldsymbol{\beta}})^{k-j}\|\tilde{\mathbf{X}}\mathbf{G}^{\top} - c_1\tilde{\mathbf{X}}\boldsymbol{\beta}\boldsymbol{a}^{\top}\|_{\mathrm{op}}^{j} \leq (C/2)^k \sum_{j=1}^{k}\binom{k}{j}(n^{-\frac{1}{2}}\log n)^{k-j}(n^{-\frac{1}{2}}\log^2 n)^j$$
$$\leq C^k n^{-\frac{k}{2}}\log^{2k} n$$

with probability $1 - o(1)$. $\qquad\qquad\square$

## F  PROOF OF THEOREM 3.3

*Proof.* We consider any fixed $\mathbf{W}_0$ such that the event $\Omega = \{\sup_{k \geq 1}\|(\mathbf{W}_0\mathbf{W}_0^{\top})^{\circ k}\|_{\mathrm{op}} \leq C\}$ from Lemma C.3 (c) holds. By Lemma C.1, each row of $H_j(\tilde{\mathbf{X}}\mathbf{W}_0^{\top})$ has second moment matrix

$$\mathbb{E}_{\boldsymbol{x} \sim \mathsf{N}(0,\mathbf{I}_d)}[H_j(\mathbf{W}_0\boldsymbol{x})H_j(\mathbf{W}_0\boldsymbol{x})^{\top}] = j!(\mathbf{W}_0\mathbf{W}_0^{\top})^{\circ j},$$

whose operator norm is $O(j!)$ on $\Omega$. Thus by (Vershynin, 2012, Theorem 5.48) and Markov's inequality, for any $j \in [L]$, for $t \geq (Cnj!)^{1/2}$, and with $M = \mathbb{E}\max_{i=1}^{n}\|H_j(\mathbf{W}_0\tilde{\boldsymbol{x}}_i)\|^2$, $\delta = C\sqrt{M\log\min(n,N)}$,

$$P(\|H_j(\tilde{\mathbf{X}}\mathbf{W}_0^{\top})\|_{\mathrm{op}} \geq t) \leq P(\|H_j(\tilde{\mathbf{X}}\mathbf{W}_0^{\top})H_j(\tilde{\mathbf{X}}\mathbf{W}_0^{\top})^{\top}/n - j!(\mathbf{W}_0\mathbf{W}_0^{\top})^{\circ j}\|_{\mathrm{op}} \geq t^2/n - Cj!)$$
$$\leq \frac{\mathbb{E}\|H_j(\tilde{\mathbf{X}}\mathbf{W}_0^{\top})H_j(\tilde{\mathbf{X}}\mathbf{W}_0^{\top})^{\top}/n - j!(\mathbf{W}_0\mathbf{W}_0^{\top})^{\circ j}\|_{\mathrm{op}}}{t^2/n - Cj!} \leq \frac{\delta\max((Cj!)^{1/2},\delta)}{t^2/n - Cj!}.$$

Next, we observe that since $H_j$ is a $j$-th degree polynomial and the normal absolute moments increase with $j$, $M = \mathbb{E}\max_{i=1}^{n}\|H_j(\mathbf{W}_0\tilde{\boldsymbol{x}}_i)\|^2 \leq C_j\mathbb{E}\max_{i=1}^{n}\|(\mathbf{W}_0\tilde{\boldsymbol{x}}_i)^{\circ j}\|^2$. Now, note that for any vectors $\boldsymbol{x}_1, \boldsymbol{x}_2$, we have $\|\boldsymbol{x}_1 \circ \boldsymbol{x}_2\|^2 \leq \|\boldsymbol{x}_1\|^2\|\boldsymbol{x}_2\|^2$ by simply expanding the norms. Thus, on the event $\Omega$, one can verify that for all $\boldsymbol{x}$, $\|(\mathbf{W}_0\boldsymbol{x})^{\circ j}\|^2 \leq C_j'\|\boldsymbol{x}\|^{2j}$ for some $C_j' > 0$. Also, we have that $A_i = \|\tilde{\boldsymbol{x}}_i\|^{2j}/N$ for $i \in [n]$ are sub-Weibull random variables with tail parameter $1/(2j)$, see e.g., Vladimirova et al. (2020); Zhang & Chen (2021). Thus, by the maximal inequality for sub-Weibull random variables (Kuchibhotla & Chakrabortty, 2022, Proposition A.6 and Remark A.1), it follows that for all $j \geq 1$, there is $C_j > 0$ such that $\mathbb{E}\max_{i=1}^{n} A_i \leq C_j(\log n)^{2j}$. Hence, $M \leq C_j''N(\log n)^{2j}$.

Thus, choosing $t = C'\sqrt{nj!}(\log n)^j$ for sufficiently large $C'$ leads to

$$\|H_j(\tilde{\mathbf{X}}\mathbf{W}_0^{\top})\|_{\mathrm{op}} = O\left(\sqrt{nj!}(\log n)^j\right) \tag{8}$$

with probability $1 - o(1)$.

Define, for all $z \in \mathbb{R}$, $\sigma_L(z) = \sum_{k=0}^{L} c_k H_k(z)$, where $L = \max\{\ell, \frac{\log n}{4(\ell+1)\log\log n}\}$. Each row of $(\sigma - \sigma_L)(\tilde{\mathbf{X}}\mathbf{W}_0^{\top})$ has second moment matrix

$$\mathbb{E}_{\boldsymbol{x} \sim \mathsf{N}(0,\mathbf{I}_d)}[(\sigma - \sigma_L)(\mathbf{W}_0\boldsymbol{x})(\sigma - \sigma_L)(\mathbf{W}_0\boldsymbol{x})^{\top}] = \sum_{k=L+1}^{\infty} k!c_k^2(\mathbf{W}_0\mathbf{W}_0^{\top})^{\circ k},$$

whose operator norm is $O(L^{-\frac{1}{2}-\omega})$ by Lemma C.3 (c) and Condition 2.3. Therefore,

$$\|(\sigma - \sigma_L)(\tilde{\mathbf{X}}\mathbf{W}_0^\top)\|_{\mathrm{op}} = O(\sqrt{n \log n} L^{-\frac{1}{2}-\omega}) = o(\sqrt{n}) \tag{9}$$

with probability $1 - o(1)$. Since $\eta = o(\sqrt{n})$, the rows of have $\mathbf{W}$ norm of $O_\mathbb{P}(1)$. Thus, we can repeat the same argument to show that with probability $1 - o(1)$, we have

$$\|(\sigma - \sigma_L)(\tilde{\mathbf{X}}\mathbf{W}^\top)\|_{\mathrm{op}} = O(\sqrt{n \log n} L^{-\frac{1}{2}-\omega}) = o(\sqrt{n}). \tag{10}$$

Let $\mathbf{F}^{(L)} := \sigma_L(\tilde{\mathbf{X}}\mathbf{W}^\top)$ and $\mathbf{F}_0^{(L)} := \sigma_L(\tilde{\mathbf{X}}\mathbf{W}_0^\top)$. We can write

$$\mathbf{F}^{(L)} = \mathbf{F}_0^{(L)} + \sum_{k=1}^{L} c_k (H_k(\tilde{\mathbf{X}}\mathbf{W}^\top) - H_k(\tilde{\mathbf{X}}\mathbf{W}_0^\top)).$$

By Lemma C.2, using $\mathbf{W} = \mathbf{W}_0 + \eta \mathbf{G}$ so that $\tilde{\mathbf{X}}\mathbf{W}^\top = \tilde{\mathbf{X}}\mathbf{W}_0^\top + \eta\tilde{\mathbf{X}}\mathbf{G}^\top$, and using that $H_0(z) = 1$ for all $z \in \mathbb{R}$,

$$H_k(\tilde{\mathbf{X}}\mathbf{W}^\top) - H_k(\tilde{\mathbf{X}}\mathbf{W}_0^\top) = \eta^k (\tilde{\mathbf{X}}\mathbf{G}^\top)^{\circ k} + \sum_{j=1}^{k-1} \binom{k}{j} \eta^j H_{k-j}(\tilde{\mathbf{X}}\mathbf{W}_0^\top) \circ (\tilde{\mathbf{X}}\mathbf{G}^\top)^{\circ j}.$$

Therefore,

$$\mathbf{F}^{(L)} = \mathbf{F}_0^{(L)} + \sum_{k=1}^{\ell} c_1^k c_k \eta^k (\tilde{\mathbf{X}}\boldsymbol{\beta})^{\circ k} (\boldsymbol{a}^{\circ k})^\top + \underbrace{\sum_{k=1}^{L} c_k \eta^k \left[ (\tilde{\mathbf{X}}\mathbf{G}^\top)^{\circ k} - c_1^k (\tilde{\mathbf{X}}\boldsymbol{\beta})^{\circ k} (\boldsymbol{a}^{\circ k})^\top \right]}_{\boldsymbol{\Delta}_1}$$

$$+ \underbrace{\sum_{k=\ell+1}^{L} c_1^k c_k \eta^k (\tilde{\mathbf{X}}\boldsymbol{\beta})^{\circ k} (\boldsymbol{a}^{\circ k})^\top}_{\boldsymbol{\Delta}_2} + \underbrace{\sum_{k=1}^{L}\sum_{j=1}^{k-1} c_k \binom{k}{j} \eta^j H_{k-j}(\tilde{\mathbf{X}}\mathbf{W}_0^\top) \circ \left[ (\tilde{\mathbf{X}}\mathbf{G}^\top)^{\circ j} - c_1^j (\tilde{\mathbf{X}}\boldsymbol{\beta})^{\circ j} (\boldsymbol{a}^{\circ j})^\top \right]}_{\boldsymbol{\Delta}_3}$$

$$+ \underbrace{\sum_{k=1}^{L}\sum_{j=1}^{k-1} c_1^j c_k \binom{k}{j} \eta^j H_{k-j}(\tilde{\mathbf{X}}\mathbf{W}_0^\top) \circ \left[ (\tilde{\mathbf{X}}\boldsymbol{\beta})^{\circ j} (\boldsymbol{a}^{\circ j})^\top \right]}_{\boldsymbol{\Delta}_4}.$$

We will show that each of $\|\boldsymbol{\Delta}_1\|_{\mathrm{op}}, \|\boldsymbol{\Delta}_2\|_{\mathrm{op}}, \|\boldsymbol{\Delta}_3\|_{\mathrm{op}}, \|\boldsymbol{\Delta}_4\|_{\mathrm{op}}$ is $o(\sqrt{n})$ with probability $1 - o(1)$.

By Lemma 3.2,

$$\|\boldsymbol{\Delta}_1\|_{\mathrm{op}} \leq \sum_{k=1}^{L} c_k C^k \eta^k n^{-\frac{k}{2}} \log^{2k} n = \tilde{O}(\eta/\sqrt{n}) = o(\sqrt{n})$$

with probability $1 - o(1)$.

By Lemma C.3 (a) and (b), using that $\alpha < \frac{\ell}{2\ell+2}$,

$$\|\boldsymbol{\Delta}_2\|_{\mathrm{op}} \leq \sum_{k=\ell+1}^{L} c_1^k c_k \eta^k \|(\tilde{\mathbf{X}}\boldsymbol{\beta})^{\circ k}\|_2 \|\boldsymbol{a}^{\circ k}\|_2 \leq \sum_{k=\ell+1}^{L} c_1^k c_k \eta^k n M_{\boldsymbol{a}}^k M_{\boldsymbol{\beta}}^k = \tilde{O}(n(\eta/\sqrt{n})^{\ell+1}) = o(\sqrt{n})$$

with probability $1 - o(1)$.

By (Bai & Silverstein, 2010, Corollary A.21), equation 8, and Lemma 3.2,

$$\|\boldsymbol{\Delta}_3\|_{\mathrm{op}} \leq \sum_{k=1}^{L}\sum_{j=1}^{k-1} c_k C^j \binom{k}{j} \sqrt{(k-j)!} \eta^j n^{-\frac{j}{2}+\frac{1}{2}} \log^{k+j} n = \tilde{O}(\eta) = o(\sqrt{n}).$$

Finally, since

$$\|H_{k-j}(\tilde{\mathbf{X}}\mathbf{W}_0^\top) \circ (\tilde{\mathbf{X}}\boldsymbol{\beta})^{\circ j} (\boldsymbol{a}^{\circ j})^\top\|_{\mathrm{op}} \leq (M_{\boldsymbol{a}} M_{\boldsymbol{\beta}})^j \sqrt{(k-j)!} n^{\frac{1}{2}} \log^{k-j} n$$
$$\leq C^j \sqrt{(k-j)!} n^{-\frac{j}{2}+\frac{1}{2}} \log^k n,$$

we also have

$$\|\boldsymbol{\Delta}_4\|_{\mathrm{op}} \leq \sum_{k=1}^{L}\sum_{j=1}^{k-1} c_k C^j \binom{k}{j} \sqrt{(k-j)!}\eta^j n^{-\frac{j}{2}+\frac{1}{2}} \log^k n = \tilde{O}(\eta) = o(\sqrt{n}).$$

This proves that with probability $1-o(1)$, we have $\mathbf{F}^{(L)} = \mathbf{F}_0^{(L)} + \sum_{k=1}^{\ell} c_1^k c_k \eta^k (\tilde{\mathbf{X}}\boldsymbol{\beta})^{\circ k}(\boldsymbol{a}^{\circ k})^{\top} + \boldsymbol{\Delta}$, with $\|\boldsymbol{\Delta}\|_{\mathrm{op}} = o(1)$. This, alongside equation 9 and equation 10, concludes the proof. $\qquad\square$

## G  PROOF OF THEOREM 3.4

By Theorem 3.3, letting $\mathbf{E} = \mathbf{F}_0 + \boldsymbol{\Delta}$, we have $\|\mathbf{E}\|_{\mathrm{op}} = O_{\mathbb{P}}(\sqrt{n})$. Note that $\sum_{k=1}^{\ell} c_1^k c_k \eta^k (\tilde{\mathbf{X}}\boldsymbol{\beta})^{\circ k}(a^{\circ k})^{\top}$ has rank $\ell$ almost surely and its left singular vector space is $\mathrm{span}\{(\tilde{\mathbf{X}}\boldsymbol{\beta})^{\circ k}\}_{k\in[\ell]}$. Also, the subspace spanned by the top-$\ell$ left singular vectors of $\mathbf{F}$ is $\mathcal{F}_{\ell}$. By Wedin's theorem (Wedin, 1972), (Chen et al., 2021, Theorem 2.9), and as $\alpha > \frac{\ell-1}{2\ell}$, we have

$$d(\mathcal{F}_{\ell}, \mathrm{span}\{(\tilde{\mathbf{X}}\boldsymbol{\beta})^{\circ k}\}_{k\in[\ell]}) = O_{\mathbb{P}}\left(\frac{\|\mathbf{E}\|_{\mathrm{op}}}{\eta^{\ell} n^{\frac{1}{2}-\frac{\ell-1}{2}} - \|\mathbf{E}\|_{\mathrm{op}}}\right) = O_{\mathbb{P}}(n^{\frac{\ell-1}{2}-\alpha\ell}) = o_{\mathbb{P}}(1).$$

## H  PROOF OF THEOREM 4.1

*Proof.* By the definition of $\hat{\boldsymbol{a}}(\mathbf{F})$, we have

$$\max\left\{\frac{1}{n}\|\tilde{\boldsymbol{y}} - \mathbf{F}\hat{\boldsymbol{a}}(\mathbf{F})\|_2^2, \lambda\|\hat{\boldsymbol{a}}(\mathbf{F})\|_2^2\right\} \leq \frac{1}{n}\|\tilde{\boldsymbol{y}} - \mathbf{F}\hat{\boldsymbol{a}}(\mathbf{F})\|_2^2 + \lambda\|\hat{\boldsymbol{a}}(\mathbf{F})\|_2^2$$

$$\leq \frac{1}{n}\|\tilde{\boldsymbol{y}} - \mathbf{F}\cdot\boldsymbol{0}\|_2^2 + \lambda\|\boldsymbol{0}\|_2^2 = \frac{1}{n}\|\tilde{\boldsymbol{y}}\|_2^2 = O_{\mathbb{P}}(1).$$

Thus,

$$\|\hat{\boldsymbol{a}}(\mathbf{F})\|_2 = O_{\mathbb{P}}(1), \quad \|\tilde{\boldsymbol{y}} - \mathbf{F}\hat{\boldsymbol{a}}(\mathbf{F})\|_2 = O_{\mathbb{P}}(\sqrt{n}). \tag{11}$$

A similar argument gives

$$\|\hat{\boldsymbol{a}}(\mathbf{F}_{\ell})\|_2 = O_{\mathbb{P}}(1), \quad \|\tilde{\boldsymbol{y}} - \mathbf{F}_{\ell}\hat{\boldsymbol{a}}(\mathbf{F}_{\ell})\|_2 = O_{\mathbb{P}}(\sqrt{n}). \tag{12}$$

Also, by the triangle inequality, and using equation 12 and Theorem 3.3, which states $\|\mathbf{F}_{\ell} - \mathbf{F}\|_{\mathrm{op}} = o_{\mathbb{P}}(\sqrt{n})$, we have

$$\|\tilde{\boldsymbol{y}} - \mathbf{F}\hat{\boldsymbol{a}}(\mathbf{F}_{\ell})\|_2 \leq \|\tilde{\boldsymbol{y}} - \mathbf{F}_{\ell}\hat{\boldsymbol{a}}(\mathbf{F}_{\ell})\|_2 + \|(\mathbf{F}_{\ell} - \mathbf{F})\hat{\boldsymbol{a}}(\mathbf{F}_{\ell})\|_2$$

$$\leq \|\tilde{\boldsymbol{y}} - \mathbf{F}_{\ell}\hat{\boldsymbol{a}}(\mathbf{F}_{\ell})\|_2 + \|\mathbf{F}_{\ell} - \mathbf{F}\|_{\mathrm{op}}\|\hat{\boldsymbol{a}}(\mathbf{F}_{\ell})\|_2 = O_{\mathbb{P}}(\sqrt{n}). \tag{13}$$

Similarly, we can prove that

$$\|\tilde{\boldsymbol{y}} - \mathbf{F}_{\ell}\hat{\boldsymbol{a}}(\mathbf{F})\|_2 = O_{\mathbb{P}}(\sqrt{n}). \tag{14}$$

For $\boldsymbol{a} = \hat{\boldsymbol{a}}(\mathbf{F})$ or $\boldsymbol{a} = \hat{\boldsymbol{a}}(\mathbf{F}_{\ell})$,

$$\frac{1}{n}\left(\|\tilde{\boldsymbol{y}} - \mathbf{F}\boldsymbol{a}\|_2^2 - \|\tilde{\boldsymbol{y}} - \mathbf{F}_{\ell}\boldsymbol{a}\|_2^2\right) = \frac{1}{n}\langle(\mathbf{F}_{\ell} - \mathbf{F})\boldsymbol{a}, \tilde{\boldsymbol{y}} - \mathbf{F}\boldsymbol{a} + \tilde{\boldsymbol{y}} - \mathbf{F}_{\ell}\boldsymbol{a}\rangle$$

$$\leq \frac{1}{n}\|\mathbf{F}_{\ell} - \mathbf{F}\|_{\mathrm{op}}\|\boldsymbol{a}\|_2\left(\|\tilde{\boldsymbol{y}} - \mathbf{F}\boldsymbol{a}\|_2 + \|\tilde{\boldsymbol{y}} - \mathbf{F}_{\ell}\boldsymbol{a}\|_2\right) = o_{\mathbb{P}}(1)$$

by equation 11, equation 12, equation 13, equation 14, and Theorem 3.3. Therefore, using the definition of $\hat{\boldsymbol{a}}(\mathbf{F}_{\ell})$,

$$\frac{1}{n}\|\tilde{\boldsymbol{y}} - \mathbf{F}_{\ell}\hat{\boldsymbol{a}}(\mathbf{F}_{\ell})\|_2^2 + \lambda\|\hat{\boldsymbol{a}}(\mathbf{F}_{\ell})\|_2^2 \leq \frac{1}{n}\|\tilde{\boldsymbol{y}} - \mathbf{F}_{\ell}\hat{\boldsymbol{a}}(\mathbf{F})\|^2 + \lambda\|\hat{\boldsymbol{a}}(\mathbf{F})\|_2^2$$

$$= \frac{1}{n}\|\tilde{\boldsymbol{y}} - \mathbf{F}\hat{\boldsymbol{a}}(\mathbf{F})\|_2^2 + \lambda\|\hat{\boldsymbol{a}}(\mathbf{F})\|_2^2 + o_{\mathbb{P}}(1)$$

and using the definition of $\hat{\boldsymbol{a}}(\mathbf{F})$,

$$\frac{1}{n}\|\tilde{\boldsymbol{y}} - \mathbf{F}\hat{\boldsymbol{a}}(\mathbf{F})\|_2^2 + \lambda\|\hat{\boldsymbol{a}}(\mathbf{F})\|_2^2 \leq \frac{1}{n}\|\tilde{\boldsymbol{y}} - \mathbf{F}\hat{\boldsymbol{a}}(\mathbf{F}_{\ell})\|^2 + \lambda\|\hat{\boldsymbol{a}}(\mathbf{F}_{\ell})\|_2^2$$

$$= \frac{1}{n}\|\tilde{\boldsymbol{y}} - \mathbf{F}_{\ell}\hat{\boldsymbol{a}}(\mathbf{F}_{\ell})\|_2^2 + \lambda\|\hat{\boldsymbol{a}}(\mathbf{F}_{\ell})\|_2^2 + o_{\mathbb{P}}(1).$$

These together prove the theorem. $\qquad\square$

## I  PROOF OF THEOREM 4.2

First, we will prove a general lemma regarding the equivalence of an augmented training loss. We will later use this result to prove the equivalence of the test loss.

**Lemma I.1.** *Let $\eta \asymp n^{\alpha}$ with $\frac{\ell-1}{2\ell} < \alpha < \frac{\ell}{2\ell+2}$ for some $\ell \in \mathbb{N}$ and $\mathbf{F}_{\ell}$ be defined as in equation 3. For the test risk $\mathcal{L}_{\text{te}}$ from Section 4.1, define*

$$\mathcal{R}_{\zeta}(\boldsymbol{a}, \mathbf{F}) = \frac{1}{n}\|\tilde{\boldsymbol{y}} - \mathbf{F}\boldsymbol{a}\|_2^2 + \lambda\|\boldsymbol{a}\|_2^2 + \zeta\mathcal{L}_{\text{te}}(\boldsymbol{a}).$$

*Then, for any $\lambda > 0$, $\zeta > 0$, we have*

$$\left|\min_{\boldsymbol{a}} \mathcal{R}_{\zeta}(\boldsymbol{a}, \mathbf{F}_{\ell}) - \min_{\boldsymbol{a}} \mathcal{R}_{\zeta}(\boldsymbol{a}, \mathbf{F})\right| = o(1), \tag{15}$$

*with probability $1 - o(1)$.*

*Proof of Lemma I.1.* Letting $\hat{\boldsymbol{a}}_{\zeta}(\mathbf{F}) = \arg\min_{\boldsymbol{a}} \mathcal{R}_{\zeta}(\boldsymbol{a}, \mathbf{F})$, we can write

$$\max\left\{\frac{1}{n}\|\tilde{\boldsymbol{y}} - \mathbf{F}\hat{\boldsymbol{a}}_{\zeta}(\mathbf{F})\|_2^2, \lambda\|\hat{\boldsymbol{a}}_{\zeta}(\mathbf{F})\|_2^2, \zeta\mathcal{L}_{\text{te}}(\hat{\boldsymbol{a}}_{\zeta}(\mathbf{F}))\right\}$$

$$\leq \frac{1}{n}\|\tilde{\boldsymbol{y}} - \mathbf{F}\hat{\boldsymbol{a}}_{\zeta}(\mathbf{F})\|_2^2 + \lambda\|\hat{\boldsymbol{a}}_{\zeta}(\mathbf{F})\|_2^2 + \zeta\mathcal{L}_{\text{te}}(\hat{\boldsymbol{a}}_{\zeta}(\mathbf{F}))$$

$$\leq \frac{1}{n}\|\tilde{\boldsymbol{y}} - \mathbf{F}\cdot\boldsymbol{0}\|_2^2 + \lambda\|\boldsymbol{0}\|_2^2 + \zeta\mathcal{L}_{\text{te}}(\boldsymbol{0}) = O_{\mathbb{P}}(1).$$

Thus,

$$\mathcal{L}_{\text{te}}(\hat{\boldsymbol{a}}_{\zeta}(\mathbf{F})) = O_{\mathbb{P}}(1), \quad \|\hat{\boldsymbol{a}}_{\zeta}(\mathbf{F})\|_2 = O_{\mathbb{P}}(1), \quad \|\tilde{\boldsymbol{y}} - \mathbf{F}\hat{\boldsymbol{a}}_{\zeta}(\mathbf{F})\|_2 = O_{\mathbb{P}}(\sqrt{n}). \tag{16}$$

A similar argument gives

$$\mathcal{L}_{\text{te}}(\hat{\boldsymbol{a}}_{\zeta}(\mathbf{F}_{\ell})) = O_{\mathbb{P}}(1), \quad \|\hat{\boldsymbol{a}}_{\zeta}(\mathbf{F}_{\ell})\|_2 = O_{\mathbb{P}}(1), \quad \|\tilde{\boldsymbol{y}} - \mathbf{F}_{\ell}\hat{\boldsymbol{a}}_{\zeta}(\mathbf{F}_{\ell})\|_2 = O_{\mathbb{P}}(\sqrt{n}). \tag{17}$$

Also by the triangle inequality, equation 17 and Theorem 3.3, which states $\|\mathbf{F}_{\ell} - \mathbf{F}\|_{\text{op}} = o_{\mathbb{P}}(\sqrt{n})$,

$$\|\tilde{\boldsymbol{y}} - \mathbf{F}\hat{\boldsymbol{a}}_{\zeta}(\mathbf{F}_{\ell})\|_2 \leq \|\tilde{\boldsymbol{y}} - \mathbf{F}_{\ell}\hat{\boldsymbol{a}}_{\zeta}(\mathbf{F}_{\ell})\|_2 + \|(\mathbf{F}_{\ell} - \mathbf{F})\hat{\boldsymbol{a}}_{\zeta}(\mathbf{F}_{\ell})\|_2$$

$$\leq \|\tilde{\boldsymbol{y}} - \mathbf{F}_{\ell}\hat{\boldsymbol{a}}_{\zeta}(\mathbf{F}_{\ell})\|_2 + \|\mathbf{F}_{\ell} - \mathbf{F}\|_{\text{op}}\|\hat{\boldsymbol{a}}_{\zeta}(\mathbf{F}_{\ell})\|_2 = O_{\mathbb{P}}(\sqrt{n}). \tag{18}$$

Similarly, we can show that

$$\|\tilde{\boldsymbol{y}} - \mathbf{F}_{\ell}\hat{\boldsymbol{a}}_{\zeta}(\mathbf{F})\|_2 = O_{\mathbb{P}}(\sqrt{n}). \tag{19}$$

For $\boldsymbol{a} = \hat{\boldsymbol{a}}_{\zeta}(\mathbf{F})$ or $\boldsymbol{a} = \hat{\boldsymbol{a}}_{\zeta}(\mathbf{F}_{\ell})$,

$$\frac{1}{n}\left(\|\tilde{\boldsymbol{y}} - \mathbf{F}\boldsymbol{a}\|_2^2 - \|\tilde{\boldsymbol{y}} - \mathbf{F}_{\ell}\boldsymbol{a}\|_2^2\right) = \frac{1}{n}\langle(\mathbf{F}_{\ell} - \mathbf{F})\boldsymbol{a}, \tilde{\boldsymbol{y}} - \mathbf{F}\boldsymbol{a} + \tilde{\boldsymbol{y}} - \mathbf{F}_{\ell}\boldsymbol{a}\rangle$$

$$\leq \frac{1}{n}\|\mathbf{F}_{\ell} - \mathbf{F}\|_{\text{op}}\|\boldsymbol{a}\|_2\left(\|\tilde{\boldsymbol{y}} - \mathbf{F}\boldsymbol{a}\|_2 + \|\tilde{\boldsymbol{y}} - \mathbf{F}_{\ell}\boldsymbol{a}\|_2\right) = o_{\mathbb{P}}(1)$$

by equation 16, equation 17, equation 18, equation 19, and Theorem 3.3. Therefore, using the definition of $\hat{\boldsymbol{a}}_{\zeta}(\mathbf{F}_{\ell})$,

$$\frac{1}{n}\|\tilde{\boldsymbol{y}} - \mathbf{F}_{\ell}\hat{\boldsymbol{a}}_{\zeta}(\mathbf{F}_{\ell})\|_2^2 + \lambda\|\hat{\boldsymbol{a}}_{\zeta}(\mathbf{F}_{\ell})\|_2^2 + \zeta\mathcal{L}_{\text{te}}(\hat{\boldsymbol{a}}_{\zeta}(\mathbf{F}_{\ell}))$$

$$\leq \frac{1}{n}\|\tilde{\boldsymbol{y}} - \mathbf{F}_{\ell}\hat{\boldsymbol{a}}_{\zeta}(\mathbf{F})\|^2 + \lambda\|\hat{\boldsymbol{a}}_{\zeta}(\mathbf{F})\|_2^2 + \zeta\mathcal{L}_{\text{te}}(\hat{\boldsymbol{a}}_{\zeta}(\mathbf{F}))$$

$$= \frac{1}{n}\|\tilde{\boldsymbol{y}} - \mathbf{F}\hat{\boldsymbol{a}}_{\zeta}(\mathbf{F})\|_2^2 + \lambda\|\hat{\boldsymbol{a}}_{\zeta}(\mathbf{F})\|_2^2 + \zeta\mathcal{L}_{\text{te}}(\hat{\boldsymbol{a}}_{\zeta}(\mathbf{F})) + o_{\mathbb{P}}(1),$$

and using the definition of $\hat{\boldsymbol{a}}_{\zeta}(\mathbf{F})$,

$$\frac{1}{n}\|\tilde{\boldsymbol{y}} - \mathbf{F}\hat{\boldsymbol{a}}_{\zeta}(\mathbf{F})\|_2^2 + \lambda\|\hat{\boldsymbol{a}}_{\zeta}(\mathbf{F})\|_2^2 + \zeta\mathcal{L}_{\text{te}}(\hat{\boldsymbol{a}}_{\zeta}(\mathbf{F}))$$

$$\leq \frac{1}{n}\|\tilde{\boldsymbol{y}} - \mathbf{F}\hat{\boldsymbol{a}}_{\zeta}(\mathbf{F}_{\ell})\|^2 + \lambda\|\hat{\boldsymbol{a}}_{\zeta}(\mathbf{F}_{\ell})\|_2^2 + \zeta\mathcal{L}_{\text{te}}(\hat{\boldsymbol{a}}_{\zeta}(\mathbf{F}_{\ell}))$$

$$= \frac{1}{n}\|\tilde{\boldsymbol{y}} - \mathbf{F}_{\ell}\hat{\boldsymbol{a}}_{\zeta}(\mathbf{F}_{\ell})\|_2^2 + \lambda\|\hat{\boldsymbol{a}}_{\zeta}(\mathbf{F}_{\ell})\|_2^2 + \zeta\mathcal{L}_{\text{te}}(\hat{\boldsymbol{a}}_{\zeta}(\mathbf{F}_{\ell})) + o_{\mathbb{P}}(1).$$

Putting these together, we have

$$\left| \min_{\boldsymbol{a}} \mathcal{R}_\zeta(\boldsymbol{a}, \mathbf{F}_\ell) - \min_{\boldsymbol{a}} \mathcal{R}_\zeta(\boldsymbol{a}, \mathbf{F}) \right| = o_{\mathbb{P}}(1), \tag{20}$$

which concludes the proof. □

Now, we use this lemma to prove the equivalence of the test loss.

*Proof of Theorem 4.2.* We will argue by contradiction. Assume that $\mathcal{L}_{\mathbf{F}} \neq \mathcal{L}_{\mathbf{F}_\ell}$ and let $\mathcal{L} = \frac{1}{2}(\mathcal{L}_{\mathbf{F}} + \mathcal{L}_{\mathbf{F}_\ell})$. Now, consider the following two optimization problems:

$$\mathcal{L}_1 = \min_{\mathcal{L}_{\text{te}}(\boldsymbol{a}) \leq \mathcal{L}} \frac{1}{n} \|\tilde{\boldsymbol{y}} - \mathbf{F}\boldsymbol{a}\|_2^2 + \lambda \|\boldsymbol{a}\|_2^2, \qquad \mathcal{L}_2 = \min_{\mathcal{L}_{\text{te}}(\boldsymbol{a}) \leq \mathcal{L}} \frac{1}{n} \|\tilde{\boldsymbol{y}} - \mathbf{F}_\ell \boldsymbol{a}\|_2^2 + \lambda \|\boldsymbol{a}\|_2^2.$$

Without loss of generality, assume that $\mathcal{L}_{\mathbf{F}} < \mathcal{L}_{\mathbf{F}_\ell}$. The solution of the first optimization problem will still converge to $\mathcal{L}_{\text{tr}}(\mathbf{F})$ because $\mathcal{L}_{\mathbf{F}} < \mathcal{L}$. However, the solution of the second optimization problem will converge to a value greater than $\mathcal{L}_{\text{tr}}(\mathbf{F}_\ell)$, because $\mathcal{L}_{\mathbf{F}_\ell} > \mathcal{L}$ and the objective is $\lambda$-strongly convex. Note that by Theorem 4.1, we asymptotically have $\mathcal{L}_{\text{tr}}(\mathbf{F}_\ell) = \mathcal{L}_{\text{tr}}(\mathbf{F})$. Thus $\mathcal{L}_1$ and $\mathcal{L}_2$ converge to different quantities as $n \to \infty$. However, using the minimax theorem and since the objectives are $\lambda$-strongly convex, we can write

$$\mathcal{L}_1 = \max_{\zeta > 0} -\zeta \mathcal{L} + \min_{\boldsymbol{a}} \left[ \frac{1}{n} \|\tilde{\boldsymbol{y}} - \mathbf{F}\boldsymbol{a}\|_2^2 + \lambda \|\boldsymbol{a}\|_2^2 + \zeta \mathcal{L}_{\text{te}}(\boldsymbol{a}) \right],$$

$$\mathcal{L}_2 = \max_{\zeta > 0} -\zeta \mathcal{L} + \min_{\boldsymbol{a}} \left[ \frac{1}{n} \|\tilde{\boldsymbol{y}} - \mathbf{F}_\ell \boldsymbol{a}\|_2^2 + \lambda \|\boldsymbol{a}\|_2^2 + \zeta \mathcal{L}_{\text{te}}(\boldsymbol{a}) \right].$$

According to Lemma I.1, the two minima above converge to the same value for any fixed $\zeta$. Note that, as functions of $\zeta$, both maxima are concave as they are minima of linear functions of $\zeta$. Hence, by using the concave version of (Abbasi et al., 2019, Lemma 1), we have that $\mathcal{L}_1$ and $\mathcal{L}_2$ converge to the same value, which is a contradiction. □

## J   GAUSSIAN EQUIVALENCE CONJECTURE

Results similar in spirit to Gaussian equivalence (Conjecture 4.3) for non-linear random matrices were introduced in El Karoui (2010); Cheng & Singer (2013); Fan & Montanari (2019). They have been repeatedly used in recent studies of random feature models Mei & Montanari (2022); Montanari et al. (2019); Adlam & Pennington (2020a;b); Tripuraneni et al. (2021); Goldt et al. (2022); Mel & Pennington (2021); d'Ascoli et al. (2021); Loureiro et al. (2021); Lee et al. (2023); Hassani & Javanmard (2022); Hu & Lu (2023); Montanari & Saeed (2022). Also, there has been progress on proving the Gaussian equivalence property for a multi-layer network with only the final layer trained Bosch et al. (2023); Cui et al. (2023).

In more distantly related work in random matrix theory literature, the phenomenon that eigenvalue statistics in the bulk spectrum of a random matrix do not depend on the specific law of the matrix entries is referred to as "bulk universality" Wigner (1955); Gaudin (1961); Mehta (2004); Dyson (1962); Erdös et al. (2010; 2012); El Karoui (2010); Tao & Vu (2011).

Erdös (2019) shows that local spectral laws of correlated random Hermitian matrices can be fully determined by their first and second moments, through the matrix Dyson equation. Also, Banna et al. (2015; 2020) show that spectral distributions of correlated symmetric random matrices can be characterized by Gaussian matrices with matching correlation structures.

In our case, we apply the Gaussian equivalence conjecture to the following quantities for $p, q \in \mathbb{N}_0$ and $\boldsymbol{\beta}_1, \boldsymbol{\beta}_2 \in \{\boldsymbol{\beta}, \boldsymbol{\beta}_\star\}$: $H_p(\tilde{\mathbf{X}}\boldsymbol{\beta}_1)^\top \bar{\mathbf{R}}_0 H_q(\tilde{\mathbf{X}}\boldsymbol{\beta}_2)$, and $\frac{1}{\sqrt{N}} H_p(\tilde{\mathbf{X}}\boldsymbol{\beta}_1)^\top \bar{\mathbf{R}}_0 \mathbf{F}_0 H_q(N^{1/2}\boldsymbol{a})$.

## K   PROOFS OF RESULTS FROM SECTION 4.2

Here, we will prove the results in Section 4.2. First, we will provide several lemmas, which will be used in our proofs. The first lemma allows us to approximate linear and quadratic forms of $\boldsymbol{\beta}$ in terms of $\boldsymbol{\beta}_\star$; the quadratic form result is from Ba et al. (2022). Its proof is in Section N.2.

**Lemma K.1.** *For any $d \in \mathbb{N}$, let $\mathbf{v} \in \mathbb{R}^d$ and $\mathbf{D} \in \mathbb{R}^{d \times d}$ be vectors and matrices, fixed or independent of $\mathbf{X}, \boldsymbol{\beta}_\star, \varepsilon_1, \ldots, \varepsilon_n$, and satisfy $\|\mathbf{v}\|_2, \|\mathbf{D}\|_{\mathrm{op}} \leq C$ almost surely, uniformly for some constant $C > 0$. Under Condition 2.1, we have*

$$\left| \mathbf{v}^\top \boldsymbol{\beta} - c_{\star,1} \mathbf{v}^\top \boldsymbol{\beta}_\star \right| \to 0, \quad \left| \boldsymbol{\beta}^\top \mathbf{D} \boldsymbol{\beta} - \frac{1}{n}(c_\star^2 + \sigma_\varepsilon^2) \operatorname{tr} \mathbf{D} - c_{\star,1}^2 \boldsymbol{\beta}_\star^\top \mathbf{D} \boldsymbol{\beta}_\star \right| \to 0$$

*in probability as $d \to \infty$.*

We will use the expression derived for the training loss in the following lemma; see Section N.3 for the proof.

**Lemma K.2.** *The training loss $\mathcal{L}_{\mathrm{tr}}(\mathbf{F})$ can be written as*

$$\mathcal{L}_{\mathrm{tr}}(\mathbf{F}) = \lambda \tilde{\boldsymbol{y}}^\top (\mathbf{F}\mathbf{F}^\top + \lambda n \mathbf{I}_n)^{-1} \tilde{\boldsymbol{y}}.$$

The following lemma will be used in proving concentration of certain quadratic forms appearing in the proofs; see Section N.4 for the proof.

**Lemma K.3.** *Let $g : \mathbb{R} \to \mathbb{R}$ be a polynomial, $\mathbf{D} \in \mathbb{R}^{n \times n}$ be a matrix with $\|\mathbf{D}\|_{\mathrm{op}} = O_{\mathbb{P}}(1/n)$, and $\mathbf{Z} \in \mathbb{R}^n$ be a vector of i.i.d. Gaussian random variables with bounded variance independent of $\mathbf{D}$. We have*

$$\left| g(\mathbf{Z})^\top \mathbf{D} \ g(\mathbf{Z}) - \mathbb{E}[g(\mathbf{Z})^\top \mathbf{D} \ g(\mathbf{Z})] \right| \to_P 0,$$

*in which $g$ is applied elementwise.*

The limiting values of two key quadratic forms appearing in the proof are derived in the following lemma, whose proof is deferred to Section N.5.

**Lemma K.4.** *Let $m_1$ and $m_2$ be the solutions to the system of fixed point equations from equation 4. Then, the following holds:*

*(a)* $\boldsymbol{\beta}^\top \tilde{\mathbf{X}}^\top \bar{\mathbf{R}}_0 \tilde{\mathbf{X}} \boldsymbol{\beta} = \psi(c_\star^2 + \sigma_\varepsilon^2) m_2 + \frac{\psi}{\phi} c_{\star,1}^2 m_2 + o_{\mathbb{P}}(1) = \Theta_{\mathbb{P}}(1).$

*(b)* $\boldsymbol{a}^\top \mathbf{F}_0^\top \bar{\mathbf{R}}_0 \mathbf{F}_0 \boldsymbol{a} - \|\boldsymbol{a}\|_2^2 = -\lambda \frac{\psi^2}{\phi^2} m_1 + \frac{\psi}{\phi} - 1 + o_{\mathbb{P}}(1) = \Theta_{\mathbb{P}}(1).$

*In particular, $\psi(c_\star^2 + \sigma_\varepsilon^2) m_2 + \frac{\psi}{\phi} c_{\star,1}^2 m_2 \neq 0$ and $-\lambda \frac{\psi^2}{\phi^2} m_1 + \frac{\psi}{\phi} - 1 \neq 0$.*

The following lemmas will be used in the computations. We defer the proofs of these lemmas to Sections N.6, N.7, N.8, and N.9 respectively.

**Lemma K.5.** *For any $p, q \in \mathbb{N}_0$, $p \neq q$ and any vector $\boldsymbol{u} \in \mathbb{R}^n$, with $\|\boldsymbol{u}\|_2 = 1$ independent of $\bar{\mathbf{R}}_0$, we have $H_q(\tilde{\mathbf{X}}\boldsymbol{u})^\top \bar{\mathbf{R}}_0 H_p(\tilde{\mathbf{X}}\boldsymbol{u}) = o_{\mathbb{P}}(1)$.*

**Lemma K.6.** *For any $p \in \mathbb{N}$, we have*

*(a)* $\sqrt{N} H_p(\tilde{\mathbf{X}}\boldsymbol{\beta}_\star) \bar{\mathbf{R}}_0 \mathbf{F}_0 \boldsymbol{a}^{\circ 2} = o_{\mathbb{P}}(1),$

*(b)* $\sqrt{N} H_p(\tilde{\mathbf{X}}\boldsymbol{\beta}) \bar{\mathbf{R}}_0 \mathbf{F}_0 \boldsymbol{a}^{\circ 2} = o_{\mathbb{P}}(1).$

**Lemma K.7.** *For $s \in \{1, 2\}$, $p \in \mathbb{N}$, and $p \neq s$, we have $H_p(\tilde{\mathbf{X}}\boldsymbol{\beta}_\star)^\top \bar{\mathbf{R}}_0 (\tilde{\mathbf{X}}\boldsymbol{\beta})^{\circ s} = o_{\mathbb{P}}(1)$. Further,*

$$\lim_{n,N,d \to \infty} H_2(\tilde{\mathbf{X}}\boldsymbol{\beta}_\star)^\top \bar{\mathbf{R}}_0 (\tilde{\mathbf{X}}\boldsymbol{\beta})^{\circ 2} = 2 c_{\star,1}^2 \frac{\psi m_1}{\phi}.$$

**Lemma K.8.** *We have*

$$\lim_{n,N,d \to \infty} (\tilde{\mathbf{X}}\boldsymbol{\beta})^{\circ 2 \top} \bar{\mathbf{R}}_0 (\tilde{\mathbf{X}}\boldsymbol{\beta})^{\circ 2} = \frac{3 \psi m_1}{\phi} [\phi(c_\star^2 + \sigma_\varepsilon^2) + c_{\star,1}^2]^2.$$

Now, we will first provide a proof of Theorem 4.4 in the case of $\ell = 1$ and $\ell = 2$ for a better insight into the proof techniques. We will then prove the general form in Section L.

### K.1 PROOF FOR $\ell = 1$

*Proof.* In the $\ell = 1$ regime, due to Theorem 4.1, we can replace $\mathbf{F}$ by $\mathbf{F}_1$ (defined in equation 3) to compute the training loss. Hence, from now on we let $\mathbf{F} = \mathbf{F}_1$. We can write $\mathbf{F}\mathbf{F}^\top = \mathbf{F}_0\mathbf{F}_0^\top + \mathbf{U}\mathbf{K}\mathbf{U}^\top$ where $\mathbf{U} = [\,\mathbf{F}_0\boldsymbol{a} \mid \tilde{\mathbf{X}}\boldsymbol{\beta}\,]$ and

$$\mathbf{K} = \begin{bmatrix} 0 & c_1^2\eta \\ c_1^2\eta & c_1^4\eta^2\|\boldsymbol{a}\|_2^2 \end{bmatrix}.$$

Based on Lemma K.2, the training loss depends on $\bar{\mathbf{R}} = (\mathbf{F}\mathbf{F}^\top + \lambda n\mathbf{I}_n)^{-1}$. Using the Woodbury formula, this matrix can be written in terms of $\bar{\mathbf{R}}_0 = (\mathbf{F}_0\mathbf{F}_0^\top + \lambda n\mathbf{I}_n)^{-1}$ as

$$\bar{\mathbf{R}} = \bar{\mathbf{R}}_0 - \bar{\mathbf{R}}_0\mathbf{U}(\mathbf{K}^{-1} + \mathbf{U}^\top\bar{\mathbf{R}}_0\mathbf{U})^{-1}\mathbf{U}^\top\bar{\mathbf{R}}_0. \tag{21}$$

Defining $\mathbf{T} = (\mathbf{K}^{-1} + \mathbf{U}^\top\bar{\mathbf{R}}_0\mathbf{U})^{-1} \in \mathbb{R}^{2\times 2}$ and substituting $\bar{\mathbf{R}} = \bar{\mathbf{R}}_0 - \bar{\mathbf{R}}_0\mathbf{U}\mathbf{T}\mathbf{U}^\top\bar{\mathbf{R}}_0$ in the formula for training loss in Lemma K.2, we find

$$\mathcal{L}_{\mathrm{tr}}(\mathbf{F}_0) - \mathcal{L}_{\mathrm{tr}}(\mathbf{F}) = \lambda\tilde{\boldsymbol{y}}^\top\bar{\mathbf{R}}_0\mathbf{U}\mathbf{T}\mathbf{U}^\top\bar{\mathbf{R}}_0\tilde{\boldsymbol{y}}. \tag{22}$$

Using equation 22 and $\mathbf{U} = [\,\mathbf{F}_0\boldsymbol{a} \mid \tilde{\mathbf{X}}\boldsymbol{\beta}\,]$, the loss difference can be written as

$$\mathcal{L}_{\mathrm{tr}}(\mathbf{F}_0) - \mathcal{L}_{\mathrm{tr}}(\mathbf{F}) = \lambda\Big[T_{11}(\tilde{\boldsymbol{y}}^\top\bar{\mathbf{R}}_0\mathbf{F}_0\boldsymbol{a})^2$$

$$+ (T_{12} + T_{21})\tilde{\boldsymbol{y}}^\top\bar{\mathbf{R}}_0\tilde{\mathbf{X}}\boldsymbol{\beta}\cdot\boldsymbol{a}^\top\mathbf{F}_0^\top\bar{\mathbf{R}}_0\tilde{\boldsymbol{y}} + T_{22}(\tilde{\boldsymbol{y}}^\top\bar{\mathbf{R}}_0\tilde{\mathbf{X}}\boldsymbol{\beta})^2\Big], \tag{23}$$

in which $T_{ij}$ are the elements of the matrix $\mathbf{T}$. Using

$$\mathbf{T} = \frac{\begin{bmatrix} \boldsymbol{\beta}^\top\tilde{\mathbf{X}}^\top\bar{\mathbf{R}}_0\tilde{\mathbf{X}}\boldsymbol{\beta} & -\frac{1}{c_1^2\eta} - \boldsymbol{a}^\top\mathbf{F}_0^\top\bar{\mathbf{R}}_0\tilde{\mathbf{X}}\boldsymbol{\beta} \\ -\frac{1}{c_1^2\eta} - \boldsymbol{\beta}^\top\tilde{\mathbf{X}}^\top\bar{\mathbf{R}}_0\mathbf{F}_0\boldsymbol{a} & \boldsymbol{a}^\top\mathbf{F}_0^\top\bar{\mathbf{R}}_0\mathbf{F}_0\boldsymbol{a} - \|\boldsymbol{a}\|_2^2 \end{bmatrix}}{\left(\boldsymbol{\beta}^\top\tilde{\mathbf{X}}^\top\bar{\mathbf{R}}_0\tilde{\mathbf{X}}\boldsymbol{\beta}\right)\left(\boldsymbol{a}^\top\mathbf{F}_0^\top\bar{\mathbf{R}}_0\mathbf{F}_0\boldsymbol{a} - \|\boldsymbol{a}\|_2^2\right) - \left(\frac{1}{c_1^2\eta} + \boldsymbol{a}^\top\mathbf{F}_0^\top\bar{\mathbf{R}}_0\tilde{\mathbf{X}}\boldsymbol{\beta}\right)^2},$$

we will compute the limit of each term appearing in equation 23 separately:

**Term 1.** The first term can be written as

$$\delta_1 = \lambda T_{11}(\tilde{\boldsymbol{y}}^\top\bar{\mathbf{R}}_0\mathbf{F}_0\boldsymbol{a})^2$$

$$= \frac{\lambda\left(\boldsymbol{\beta}^\top\tilde{\mathbf{X}}^\top\bar{\mathbf{R}}_0\tilde{\mathbf{X}}\boldsymbol{\beta}\right)(\tilde{\boldsymbol{y}}^\top\bar{\mathbf{R}}_0\mathbf{F}_0\boldsymbol{a})^2}{\left(\boldsymbol{\beta}^\top\tilde{\mathbf{X}}^\top\bar{\mathbf{R}}_0\tilde{\mathbf{X}}\boldsymbol{\beta}\right)\left(\boldsymbol{a}^\top\mathbf{F}_0^\top\bar{\mathbf{R}}_0\mathbf{F}_0\boldsymbol{a} - \|\boldsymbol{a}\|_2^2\right) - \left(\frac{1}{c_1^2\eta} + \boldsymbol{a}^\top\mathbf{F}_0^\top\bar{\mathbf{R}}_0\tilde{\mathbf{X}}\boldsymbol{\beta}\right)^2}.$$

Based on Lemma K.4, we know that $\boldsymbol{\beta}^\top\tilde{\mathbf{X}}^\top\bar{\mathbf{R}}_0\tilde{\mathbf{X}}\boldsymbol{\beta}$ and $\boldsymbol{a}^\top\mathbf{F}_0^\top\bar{\mathbf{R}}_0\mathbf{F}_0\boldsymbol{a} - \|\boldsymbol{a}\|_2^2$ are $\Theta_\mathbb{P}(1)$. Also, it can easily be seen that

$$\|\mathbf{F}_0^\top\bar{\mathbf{R}}_0\tilde{\mathbf{X}}\boldsymbol{\beta}\|_2 \leq \|\mathbf{F}_0\|_{\mathrm{op}}\|\bar{\mathbf{R}}_0\|_{\mathrm{op}}\|\tilde{\mathbf{X}}\boldsymbol{\beta}\|_2 = O_\mathbb{P}(1).$$

Hence,

$$\left(\frac{1}{c_1^2\eta} + \boldsymbol{a}^\top\mathbf{F}_0^\top\bar{\mathbf{R}}_0\tilde{\mathbf{X}}\boldsymbol{\beta}\right)^2 = o_\mathbb{P}(1) \tag{24}$$

because $\boldsymbol{a} \perp\!\!\!\perp \mathbf{F}_0^\top\bar{\mathbf{R}}_0\tilde{\mathbf{X}}\boldsymbol{\beta}$. Also, using that $\bar{\mathbf{R}}_0\mathbf{F}_0\mathbf{F}_0^\top = (\mathbf{F}_0\mathbf{F}_0^\top + \lambda n\mathbf{I}_n)^{-1}\mathbf{F}_0\mathbf{F}_0^\top = \mathbf{I} - \lambda n\bar{\mathbf{R}}_0$,

$$(\tilde{\boldsymbol{y}}^\top\bar{\mathbf{R}}_0\mathbf{F}_0\boldsymbol{a})^2 = \tilde{\boldsymbol{y}}^\top\bar{\mathbf{R}}_0\mathbf{F}_0\mathbb{E}_{\boldsymbol{a}}[\boldsymbol{a}\boldsymbol{a}^\top]\mathbf{F}_0^\top\bar{\mathbf{R}}_0\tilde{\boldsymbol{y}} + o_\mathbb{P}(1)$$

$$= \frac{1}{N}\tilde{\boldsymbol{y}}^\top\bar{\mathbf{R}}_0\mathbf{F}_0\mathbf{F}_0^\top\bar{\mathbf{R}}_0\tilde{\boldsymbol{y}} + o_\mathbb{P}(1) = \frac{1}{N}\tilde{\boldsymbol{y}}^\top\bar{\mathbf{R}}_0\tilde{\boldsymbol{y}} - \frac{\lambda n}{N}\tilde{\boldsymbol{y}}^\top\bar{\mathbf{R}}_0^2\tilde{\boldsymbol{y}} + o_\mathbb{P}(1) = o_\mathbb{P}(1),$$

where in the last inequality, we used the fact that $\tilde{\boldsymbol{y}}^\top\bar{\mathbf{R}}_0\tilde{\boldsymbol{y}} \leq \frac{1}{\lambda n}\|\tilde{\boldsymbol{y}}\|_2^2 = O_\mathbb{P}(1)$ and $\tilde{\boldsymbol{y}}^\top\bar{\mathbf{R}}_0^2\tilde{\boldsymbol{y}} \leq \frac{1}{(\lambda n)^2}\|\tilde{\boldsymbol{y}}\|_2^2 = o_\mathbb{P}(1)$. Putting everything together, it follows that $\delta_1 = o_\mathbb{P}(1)$ in probability.

**Term 2 and Term 3.** The second and third terms can be written as

$$\delta_2 = \delta_3 = \lambda T_{12} \tilde{\boldsymbol{y}}^\top \bar{\mathbf{R}}_0 \tilde{\mathbf{X}} \boldsymbol{\beta} \boldsymbol{a}^\top \mathbf{F}_0^\top \bar{\mathbf{R}}_0 \tilde{\boldsymbol{y}}$$

$$= \frac{\lambda \left( -\frac{1}{c_1^2 \eta} - \boldsymbol{a}^\top \mathbf{F}_0^\top \bar{\mathbf{R}}_0 \tilde{\mathbf{X}} \boldsymbol{\beta} \right) (\tilde{\boldsymbol{y}}^\top \bar{\mathbf{R}}_0 \tilde{\mathbf{X}} \boldsymbol{\beta} \boldsymbol{a}^\top \mathbf{F}_0^\top \bar{\mathbf{R}}_0 \tilde{\boldsymbol{y}})}{\left( \boldsymbol{\beta}^\top \tilde{\mathbf{X}}^\top \bar{\mathbf{R}}_0 \tilde{\mathbf{X}} \boldsymbol{\beta} \right) \left( \boldsymbol{a}^\top \mathbf{F}_0^\top \bar{\mathbf{R}}_0 \mathbf{F}_0 \boldsymbol{a} - \|\boldsymbol{a}\|_2^2 \right) - \left( \frac{1}{c_1^2 \eta} + \boldsymbol{a}^\top \mathbf{F}_0^\top \bar{\mathbf{R}}_0 \tilde{\mathbf{X}} \boldsymbol{\beta} \right)^2}.$$

Recall from the above argument that the denominator is $\Theta_{\mathbb{P}}(1)$ and that $\frac{1}{c_1^2 \eta} + \boldsymbol{a}^\top \mathbf{F}_0^\top \bar{\mathbf{R}}_0 \tilde{\mathbf{X}} \boldsymbol{\beta} = o_{\mathbb{P}}(1)$. Also, $\tilde{\boldsymbol{y}}^\top \bar{\mathbf{R}}_0 \tilde{\mathbf{X}} \boldsymbol{\beta} \boldsymbol{a}^\top \mathbf{F}_0^\top \bar{\mathbf{R}}_0 \tilde{\boldsymbol{y}} \leq \frac{1}{(\lambda n)^2} \|\tilde{\boldsymbol{y}}\|_2^2 \|\tilde{\mathbf{X}} \boldsymbol{\beta}\|_2 \|\boldsymbol{a}\|_2 \|\mathbf{F}_0\|_{\mathrm{op}} = O_{\mathbb{P}}(1)$. Therefore, we find $\delta_2 = \delta_3 = o_{\mathbb{P}}(1)$.

**Term 4.** This term can be written as

$$\delta_4 = \lambda T_{22} (\tilde{\boldsymbol{y}}^\top \bar{\mathbf{R}}_0 \tilde{\mathbf{X}} \boldsymbol{\beta})^2$$

$$= \frac{\lambda \left( \boldsymbol{a}^\top \mathbf{F}_0^\top \bar{\mathbf{R}}_0 \mathbf{F}_0 \boldsymbol{a} - \|\boldsymbol{a}\|_2^2 \right) (\tilde{\boldsymbol{y}}^\top \bar{\mathbf{R}}_0 \tilde{\mathbf{X}} \boldsymbol{\beta})^2}{\boldsymbol{\beta}^\top \tilde{\mathbf{X}}^\top \bar{\mathbf{R}}_0 \tilde{\mathbf{X}} \boldsymbol{\beta} \left( \boldsymbol{a}^\top \mathbf{F}_0^\top \bar{\mathbf{R}}_0 \mathbf{F}_0 \boldsymbol{a} - \|\boldsymbol{a}\|_2^2 \right) - \left( \frac{1}{c_1^2 \eta} + \boldsymbol{a}^\top \mathbf{F}_0^\top \bar{\mathbf{R}}_0 \tilde{\mathbf{X}} \boldsymbol{\beta} \right)^2} = \lambda \frac{(\tilde{\boldsymbol{y}}^\top \bar{\mathbf{R}}_0 \tilde{\mathbf{X}} \boldsymbol{\beta})^2}{\boldsymbol{\beta}^\top \tilde{\mathbf{X}}^\top \bar{\mathbf{R}}_0 \tilde{\mathbf{X}} \boldsymbol{\beta}} + o_{\mathbb{P}}(1),$$

since $\frac{1}{c_1^2 \eta} + \boldsymbol{a}^\top \mathbf{F}_0^\top \bar{\mathbf{R}}_0 \tilde{\mathbf{X}} \boldsymbol{\beta} = o_{\mathbb{P}}(1)$ and $\boldsymbol{a}^\top \mathbf{F}_0^\top \bar{\mathbf{R}}_0 \mathbf{F}_0 \boldsymbol{a} - \|\boldsymbol{a}\|_2^2 = \Theta_{\mathbb{P}}(1) \neq 0$ by Lemma K.4. By equation 1 and Condition 2.4, we can write

$$\tilde{\boldsymbol{y}} = \sum_{p=1}^{\infty} c_{\star, p} H_p(\tilde{\mathbf{X}} \boldsymbol{\beta}_\star) + \varepsilon,$$

where $\varepsilon \in \mathbb{R}^n$ is additive Gaussian noise. Note that

$$\tilde{\boldsymbol{y}}^\top \bar{\mathbf{R}}_0 \tilde{\mathbf{X}} \boldsymbol{\beta} = \sum_{p=1}^{\infty} c_{\star, p} H_p(\tilde{\mathbf{X}} \boldsymbol{\beta}_\star)^\top \bar{\mathbf{R}}_0 \tilde{\mathbf{X}} \boldsymbol{\beta} + \varepsilon^\top \bar{\mathbf{R}}_0 \tilde{\mathbf{X}} \boldsymbol{\beta} = c_{\star, 1} \boldsymbol{\beta}_\star^\top \tilde{\mathbf{X}}^\top \bar{\mathbf{R}}_0 \tilde{\mathbf{X}} \boldsymbol{\beta} + o_{\mathbb{P}}(1)$$

by Lemma K.7 and since $\|\bar{\mathbf{R}}_0 \tilde{\mathbf{X}} \boldsymbol{\beta}\|_2 = O_{\mathbb{P}}(1/\sqrt{n})$ and $\varepsilon \perp\!\!\!\perp \bar{\mathbf{R}}_0 \tilde{\mathbf{X}} \boldsymbol{\beta}$.

Further by Lemma K.1,

$$c_{\star, 1} \boldsymbol{\beta}_\star^\top \tilde{\mathbf{X}}^\top \bar{\mathbf{R}}_0 \tilde{\mathbf{X}} \boldsymbol{\beta} = c_{\star, 1}^2 \boldsymbol{\beta}_\star^\top \tilde{\mathbf{X}}^\top \bar{\mathbf{R}}_0 \tilde{\mathbf{X}} \boldsymbol{\beta}_\star + o_{\mathbb{P}}(1).$$

By summing up the fours terms computed above and using Lemma K.4, we get

$$\mathcal{L}_{\mathrm{tr}}(\mathbf{F}_0) - \mathcal{L}_{\mathrm{tr}}(\mathbf{F}) \to_P \Delta_1 = \frac{\psi \lambda c_{\star, 1}^4 m_2}{\phi[c_{\star, 1}^2 + \phi(c_\star^2 + \sigma_\varepsilon^2)]} > 0, \tag{25}$$

which concludes the proof for $\ell = 1$. $\qquad\square$

### K.2 PROOF FOR $\ell = 2$

In the $\ell = 2$ regime, based on Theorem 4.1, we can replace $\mathbf{F}$ with $\mathbf{F}_2$ (defined in equation 3) to compute the training loss. Hence, from now on we let $\mathbf{F} = \mathbf{F}_2$. We can write $\mathbf{F}\mathbf{F}^\top = \mathbf{F}_0 \mathbf{F}_0^\top + \mathbf{U}\mathbf{K}\mathbf{U}^\top$ where $\mathbf{U} = [\, \mathbf{F}_0 \boldsymbol{a} \mid \mathbf{F}_0 \boldsymbol{a}^{\circ 2} \sqrt{N} \mid \tilde{\mathbf{X}} \boldsymbol{\beta} \mid (\tilde{\mathbf{X}} \boldsymbol{\beta})^{\circ 2} \,]$ and

$$\mathbf{K} = \begin{bmatrix} 0 & 0 & c_1^2 \eta & 0 \\ 0 & 0 & 0 & c_1^2 c_2 \eta^2 / \sqrt{N} \\ c_1^2 \eta & 0 & c_1^4 \eta^2 \|\boldsymbol{a}\|_2^2 & c_1^4 c_2 \eta^3 \langle \boldsymbol{a}, \boldsymbol{a}^{\circ 2} \rangle \\ 0 & c_1^2 c_2 \eta^2 / \sqrt{N} & c_1^4 c_2 \eta^3 \langle \boldsymbol{a}^{\circ 2}, \boldsymbol{a} \rangle & c_1^4 c_2^2 \eta^4 \langle \boldsymbol{a}^{\circ 2}, \boldsymbol{a}^{\circ 2} \rangle \end{bmatrix}.$$

Recalling $\bar{\mathbf{R}} = (\mathbf{F}\mathbf{F}^\top + \lambda n \mathbf{I}_n)^{-1}$ and $\bar{\mathbf{R}}_0 = (\mathbf{F}_0 \mathbf{F}_0^\top + \lambda n \mathbf{I}_n)^{-1}$, we still have equation 21. Defining $\mathbf{T} = (\mathbf{K}^{-1} + \mathbf{U}^\top \bar{\mathbf{R}}_0 \mathbf{U})^{-1} \in \mathbb{R}^{4 \times 4}$, we have the following analogue to equation 22:

$$\mathcal{L}_{\mathrm{tr}}(\mathbf{F}_0) - \mathcal{L}_{\mathrm{tr}}(\mathbf{F}) = \lambda \tilde{\boldsymbol{y}}^\top \bar{\mathbf{R}}_0 \mathbf{U} \mathbf{T} \mathbf{U}^\top \bar{\mathbf{R}}_0 \tilde{\boldsymbol{y}}. \tag{26}$$

Denoting in what follows $\mathbf{Q} = \mathbf{F}_0^\top \bar{\mathbf{R}}_0 \mathbf{F}_0$, the inverse $\mathbf{T}^{-1}$ can be written as follows:

$$
\begin{bmatrix}
\boldsymbol{a}^\top \mathbf{Q} \boldsymbol{a} - \|\boldsymbol{a}\|_2^2 & N^{\frac{1}{2}} \boldsymbol{a}^\top (\mathbf{Q} - \mathbf{I}) \boldsymbol{a}^{\circ 2} & \boldsymbol{a}^\top \mathbf{F}_0^\top \bar{\mathbf{R}}_0 \tilde{\boldsymbol{\theta}} + \frac{1}{c_1^2 \eta} & \boldsymbol{a}^\top \mathbf{F}_0^\top \bar{\mathbf{R}}_0 \tilde{\boldsymbol{\theta}}^{\circ 2} \\[2mm]
N^{\frac{1}{2}} \boldsymbol{a}^\top (\mathbf{Q} - \mathbf{I}) \boldsymbol{a}^{\circ 2} & N \boldsymbol{a}^{\circ 2 \top} \mathbf{Q} \boldsymbol{a}^{\circ 2} - N \|\boldsymbol{a}^{\circ 2}\|_2^2 & N^{\frac{1}{2}} \boldsymbol{a}^{\circ 2 \top} \mathbf{F}_0^\top \bar{\mathbf{R}}_0 \tilde{\boldsymbol{\theta}} & N^{\frac{1}{2}} \boldsymbol{a}^{\circ 2 \top} \mathbf{F}_0^\top \bar{\mathbf{R}}_0 \tilde{\boldsymbol{\theta}}^{\circ 2} + \frac{N^{\frac{1}{2}}}{c_1^2 c_2 \eta^2} \\[2mm]
\tilde{\boldsymbol{\theta}}^\top \bar{\mathbf{R}}_0 \mathbf{F}_0 \boldsymbol{a} + \frac{1}{c_1^2 \eta} & N^{\frac{1}{2}} \tilde{\boldsymbol{\theta}}^\top \bar{\mathbf{R}}_0 \mathbf{F}_0 \boldsymbol{a}^{\circ 2} & \tilde{\boldsymbol{\theta}}^\top \bar{\mathbf{R}}_0 \tilde{\boldsymbol{\theta}} & \tilde{\boldsymbol{\theta}}^\top \bar{\mathbf{R}}_0 \tilde{\boldsymbol{\theta}}^{\circ 2} \\[2mm]
\tilde{\boldsymbol{\theta}}^{\circ 2 \top} \bar{\mathbf{R}}_0 \mathbf{F}_0 \boldsymbol{a} & N^{\frac{1}{2}} \tilde{\boldsymbol{\theta}}^{\circ 2 \top} \bar{\mathbf{R}}_0 \mathbf{F}_0 \boldsymbol{a}^{\circ 2} + \frac{N^{\frac{1}{2}}}{c_1^2 c_2 \eta^2} & \tilde{\boldsymbol{\theta}}^{\circ 2 \top} \bar{\mathbf{R}}_0 \tilde{\boldsymbol{\theta}} & \tilde{\boldsymbol{\theta}}^{\circ 2 \top} \bar{\mathbf{R}}_0 \tilde{\boldsymbol{\theta}}^{\circ 2}
\end{bmatrix}.
$$

### K.2.1 Analysis of Terms in $\mathbf{T}^{-1}$ and $\mathbf{T}$

In the following section, we will first analyze the elements of $\mathbf{T}^{-1}$:

**(1,1):** The term $\boldsymbol{a}^\top \mathbf{Q} \boldsymbol{a} - \|\boldsymbol{a}\|_2^2$ has already been analyzed in Lemma K.4 and is $\Theta_{\mathbb{P}}(1)$.

**(1,2) and (2,1):** Recalling $\mathbf{Q} = \mathbf{F}_0^\top \bar{\mathbf{R}}_0 \mathbf{F}_0$ and $\mathbf{R}_0 = (\mathbf{F}_0^\top \mathbf{F}_0 + \lambda n \mathbf{I}_N)^{-1}$, we can write

$$
\begin{aligned}
[\mathbf{T}^{-1}]_{1,2} = [\mathbf{T}^{-1}]_{2,1} &= \sqrt{N} \boldsymbol{a}^\top \mathbf{Q} \boldsymbol{a}^{\circ 2} - \sqrt{N} \langle \boldsymbol{a}, \boldsymbol{a}^{\circ 2} \rangle \\
&= -\lambda n \sqrt{N} \boldsymbol{a}^\top \mathbf{R}_0 \boldsymbol{a}^{\circ 2} = -\lambda n \sqrt{N} \boldsymbol{a}^\top \mathbf{R}_0 \left( \boldsymbol{a}^{\circ 2} - 1/N \mathbf{1}_N + 1/N \mathbf{1}_N \right).
\end{aligned}
$$

Introducing $\tilde{\boldsymbol{a}} = \sqrt{N} \boldsymbol{a} \sim \mathsf{N}(0, \mathbf{I}_N)$, and as $H_2(x) = x^2 - 1$ for all $x$, we find

$$
[\mathbf{T}^{-1}]_{1,2} = [\mathbf{T}^{-1}]_{2,1} = -\frac{\lambda n}{N} \tilde{\boldsymbol{a}}^\top \mathbf{R}_0 H_2(\tilde{\boldsymbol{a}}) - \frac{\lambda n}{\sqrt{N}} \boldsymbol{a}^\top \mathbf{R}_0 \mathbf{1}_N.
$$

The second term converges to zero as $n \to \infty$ because $\boldsymbol{a} \sim \mathsf{N}(0, \frac{1}{N} \mathbf{I}_N)$ is independent of $\mathbf{R}_0$, and $\left\| \frac{n}{\sqrt{N}} \mathbf{R}_0 \mathbf{1}_N \right\|_2 = O_{\mathbb{P}}(1)$. Moreover, the first term also converges to zero; indeed,

$$
\tilde{\boldsymbol{a}}^\top \mathbf{R}_0 H_2(\tilde{\boldsymbol{a}}) = \left( \frac{\tilde{\boldsymbol{a}}^\top + H_2(\tilde{\boldsymbol{a}})}{2} \right)^\top \mathbf{R}_0 \left( \frac{\tilde{\boldsymbol{a}}^\top + H_2(\tilde{\boldsymbol{a}})}{2} \right) - \left( \frac{\tilde{\boldsymbol{a}}^\top - H_2(\tilde{\boldsymbol{a}})}{2} \right)^\top \mathbf{R}_0 \left( \frac{\tilde{\boldsymbol{a}}^\top - H_2(\tilde{\boldsymbol{a}})}{2} \right).
$$

Lemma K.3 can be used with $\mathbf{D} = \mathbf{R}$ to prove the concentration of both term around their expectation. Note that the expectation of $\tilde{\boldsymbol{a}}^\top \mathbf{R}_0 H_2(\tilde{\boldsymbol{a}})$ is zero because of the orthogonality property of Hermite polynomials and the independence of $\tilde{\boldsymbol{a}}$ and $\mathbf{R}_0$. Putting everything together, we conclude that $[\mathbf{T}^{-1}]_{1,2} = [\mathbf{T}^{-1}]_{2,1} = o_{\mathbb{P}}(1)$.

**(1,3) and (3,1):** Recalling that $\tilde{\boldsymbol{\theta}} = \tilde{\mathbf{X}} \boldsymbol{\beta}$, it follows from equation 24 that this term is $o_{\mathbb{P}}(1)$.

**(1,4) and (4,1):** To bound $\boldsymbol{a}^\top \mathbf{F}_0^\top \bar{\mathbf{R}}_0 \tilde{\boldsymbol{\theta}}^{\circ 2}$, note that

$$
\|\mathbf{F}_0^\top \bar{\mathbf{R}}_0 \tilde{\boldsymbol{\theta}}^{\circ 2}\|_{\mathrm{op}} \leq \|\mathbf{F}_0\|_{\mathrm{op}} \|\bar{\mathbf{R}}_0\|_{\mathrm{op}} \|\tilde{\boldsymbol{\theta}}^{\circ 2}\|_2 = O_{\mathbb{P}}(1).
$$

Hence, because $\boldsymbol{a} \sim \mathsf{N}(0, \frac{1}{N} \mathbf{I}_N)$ is independent of $\mathbf{F}_0^\top \bar{\mathbf{R}}_0 \tilde{\boldsymbol{\theta}}^{\circ 2}$, we have

$$
[\mathbf{T}^{-1}]_{1,4} = [\mathbf{T}^{-1}]_{4,1} = \boldsymbol{a}^\top \mathbf{F}_0^\top \bar{\mathbf{R}}_0 \tilde{\boldsymbol{\theta}}^{\circ 2} = o_{\mathbb{P}}(1).
$$

**(2,2):** This term is $O_{\mathbb{P}}(1)$, because $\boldsymbol{a} \sim \mathsf{N}(0, \frac{1}{N} \mathbf{I}_N)$, so

$$
\begin{aligned}
[\mathbf{T}^{-1}]_{2,2} &= N \boldsymbol{a}^{\circ 2 \top} \mathbf{Q} \boldsymbol{a}^{\circ 2} - N \|\boldsymbol{a}^{\circ 2}\|_2^2 = -\lambda N n \, \boldsymbol{a}^{\circ 2 \top} (\mathbf{F}_0^\top \mathbf{F}_0 + \lambda n \mathbf{I}_N)^{-1} \boldsymbol{a}^{\circ 2} \\
&\leq \lambda N n \|\boldsymbol{a}^{\circ 2}\|_2^2 \cdot \|(\mathbf{F}_0^\top \mathbf{F}_0 + \lambda n \mathbf{I}_N)^{-1}\|_{\mathrm{op}} = O_{\mathbb{P}}(1).
\end{aligned}
$$

**(2,3) and (3,2):** To bound $\sqrt{N}\boldsymbol{a}^{\circ 2\top}\mathbf{F}_0^\top\bar{\mathbf{R}}_0\tilde{\boldsymbol{\theta}}$, note that

$$\|\sqrt{N}\boldsymbol{a}^{\circ 2\top}\mathbf{F}_0^\top\bar{\mathbf{R}}_0\tilde{\mathbf{X}}\|_2 \le \|\sqrt{N}\boldsymbol{a}^{\circ 2}\|_2\|\mathbf{F}_0\|_{\mathrm{op}}\|\mathbf{R}_0\|_{\mathrm{op}}\|\tilde{\mathbf{X}}\|_{\mathrm{op}}$$
$$\le C\cdot\sqrt{N}\cdot\frac{1}{n}\cdot\sqrt{N} = O_\mathbb{P}(1).$$

Also, by Lemma K.1, we have

$$[\mathbf{T}^{-1}]_{2,3} = [\mathbf{T}^{-1}]_{3,2} = \sqrt{N}\boldsymbol{a}^{\circ 2\top}\mathbf{F}_0^\top\bar{\mathbf{R}}_0\tilde{\mathbf{X}}\boldsymbol{\beta} = c_{\star,1}\sqrt{N}\boldsymbol{a}^{\circ 2\top}\mathbf{F}_0^\top\bar{\mathbf{R}}_0\tilde{\mathbf{X}}\boldsymbol{\beta}_\star + o_\mathbb{P}(1),$$

which converges to zero, because $\boldsymbol{\beta}_\star \sim \mathsf{N}(0,\frac{1}{d}\mathbf{I}_d)$ and is independent of $\sqrt{N}\boldsymbol{a}^{\circ 2\top}\mathbf{F}_0^\top\bar{\mathbf{R}}_0\tilde{\mathbf{X}}$, which has bounded norm in probability.

**(2,4) and (4,2):** First note that in the regime where $\ell = 2$, we have $\frac{\sqrt{N}}{\eta^2} \to 0$. Hence, we can write

$$[\mathbf{T}^{-1}]_{2,4} = \sqrt{N}(\tilde{\mathbf{X}}\boldsymbol{\beta})^{\circ 2\top}\bar{\mathbf{R}}_0\mathbf{F}_0\boldsymbol{a}^{\circ 2} + o_\mathbb{P}(1)$$
$$= \sqrt{N}H_2(\tilde{\mathbf{X}}\boldsymbol{\beta})^\top\bar{\mathbf{R}}_0\mathbf{F}_0\boldsymbol{a}^{\circ 2} + \sqrt{N}\mathbf{1}_n^\top\bar{\mathbf{R}}_0\mathbf{F}_0\boldsymbol{a}^{\circ 2} + o_\mathbb{P}(1). \tag{27}$$

By Lemma K.6, the first term converges in probability to zero. Moreover, $\boldsymbol{a} \sim \mathsf{N}(0,\frac{1}{N}\mathbf{I}_N)$ is independent of $\bar{\mathbf{R}}_0\mathbf{F}_0$, and $\|\mathbf{1}_n^\top\bar{\mathbf{R}}_0\mathbf{F}_0\|_2 = O_\mathbb{P}(1)$. Thus, we have that $\sqrt{N}\mathbf{1}_n^\top\bar{\mathbf{R}}_0\mathbf{F}_0\left(\boldsymbol{a}^{\circ 2} - 1/N\mathbf{1}_N\right) \to_P 0$. Hence, we find

$$[\mathbf{T}^{-1}]_{2,4} = \sqrt{N}\mathbf{1}_n^\top\bar{\mathbf{R}}_0\mathbf{F}_0\mathbf{1}_N/N + o_\mathbb{P}(1).$$

Based on Conjecture 4.3, we can replace $\mathbf{F}_0$ with $\mathbf{F}_0 = c_1\tilde{\mathbf{X}}\mathbf{W}_0^\top + c_{>1}\mathbf{Z}$, where $\mathbf{Z} \in \mathbb{R}^{n\times d}$ is an independent random matrix with $\mathsf{N}(0,1)$ entries, without changing the limit. Now, the linearized $\mathbf{F}_0$ is left-orthogonally invariant, hence $\mathbf{F}_0$ has the same distribution as $\mathbf{O}\mathbf{F}_0$, where $\mathbf{O}$ is uniformly distributed over the Haar measure of $d$-dimensional orthogonal matrices, independently of all other randomness. Hence,

$$N^{-1/2}\mathbf{1}_n^\top\bar{\mathbf{R}}_0\mathbf{F}_0\mathbf{1}_N =_d N^{-1/2}\mathbf{1}_n^\top\mathbf{O}\bar{\mathbf{R}}_0\mathbf{F}_0\mathbf{1}_N.$$

Now, $\mathbf{O}^\top\mathbf{1}_n =_d \sqrt{n}\boldsymbol{z}/\|\boldsymbol{z}\|_2$, where $\boldsymbol{z} \sim \mathsf{N}(0,\mathbf{I}_n)$. Moreover $\|\boldsymbol{z}\|_2 = \sqrt{n}(1 + o_\mathbb{P}(1))$, hence replacing $\mathbf{O}^\top\mathbf{1}_n$ with $\boldsymbol{z}^\top$ introduces negligible error. Hence,

$$[\mathbf{T}^{-1}]_{2,4} =_d N^{-1/2}\boldsymbol{z}^\top\bar{\mathbf{R}}_0\mathbf{F}_0\mathbf{1}_N + o_\mathbb{P}(1).$$

Now, $\boldsymbol{z}^\top\bar{\mathbf{R}}_0\mathbf{F}_0\mathbf{1}_N \sim \mathsf{N}(0,\|\bar{\mathbf{R}}_0\mathbf{F}_0\mathbf{1}_N\|_2^2)$, and $\|\bar{\mathbf{R}}_0\mathbf{F}_0\mathbf{1}_N\|_2 = O_\mathbb{P}(1)$, thus $[\mathbf{T}^{-1}]_{2,4} \to_P 0$.

**(3,3):** We have $\|\tilde{\boldsymbol{\theta}}\|_2 = O_\mathbb{P}(\sqrt{N})$ and $\|\bar{\mathbf{R}}_0\|_{\mathrm{op}} = O_\mathbb{P}(1/n)$. Thus, $[\mathbf{T}^{-1}]_{3,3} = O_\mathbb{P}(1)$.

**(3,4) and (4,3):** First, note that defining $\tilde{\boldsymbol{\beta}} = \frac{\boldsymbol{\beta}}{\|\boldsymbol{\beta}\|_2}$, and as $H_2(x) = x^2 - 1$ for all $x$, we can write

$$[\mathbf{T}^{-1}]_{3,4} = [\mathbf{T}^{-1}]_{4,3} = \tilde{\boldsymbol{\theta}}^\top\bar{\mathbf{R}}_0\tilde{\boldsymbol{\theta}}^{\circ 2} = \|\boldsymbol{\beta}\|_2^3\left((\tilde{\mathbf{X}}\tilde{\boldsymbol{\beta}})^\top\bar{\mathbf{R}}_0(\tilde{\mathbf{X}}\tilde{\boldsymbol{\beta}})^{\circ 2}\right)$$
$$= \|\boldsymbol{\beta}\|_2^3\left((\tilde{\mathbf{X}}\tilde{\boldsymbol{\beta}})^\top\bar{\mathbf{R}}_0 H_2(\tilde{\mathbf{X}}\tilde{\boldsymbol{\beta}})\right) + \|\boldsymbol{\beta}\|_2^2\left(\tilde{\boldsymbol{\theta}}^\top\bar{\mathbf{R}}_0\mathbf{1}_N\right).$$

Now, by Lemma K.1, we have

$$\tilde{\boldsymbol{\theta}}^\top\bar{\mathbf{R}}_0\mathbf{1}_N = c_{\star,1}\tilde{\boldsymbol{\theta}}_\star^\top\bar{\mathbf{R}}_0\mathbf{1}_N + o_\mathbb{P}(1).$$

Now, note that $\|\tilde{\mathbf{X}}\bar{\mathbf{R}}_0\mathbf{1}_N\|_2 = O_\mathbb{P}(1)$ and $\boldsymbol{\beta}_\star \sim \mathsf{N}(0,\frac{1}{d}\mathbf{I}_d)$ is independent of $\tilde{\mathbf{X}}\bar{\mathbf{R}}_0\mathbf{1}_N$, which implies that the second term converges to zero. By using Lemma K.5 for $\boldsymbol{u} = \tilde{\boldsymbol{\beta}}$, the first term also converges to zero. Putting these together, we have $[\mathbf{T}^{-1}]_{3,4} = [\mathbf{T}^{-1}]_{4,3} = o_\mathbb{P}(1)$.

**(4,4):** We have $\|\tilde{\boldsymbol{\theta}}^{\circ 2}\|_2 = O_\mathbb{P}(\sqrt{N})$ and $\|\bar{\mathbf{R}}_0\|_{\mathrm{op}} = O_\mathbb{P}(1/n)$. Thus, $[\mathbf{T}^{-1}]_{4,4} = O_\mathbb{P}(1)$.

Now, putting everything together, the matrix $\mathbf{T}^{-1}$ can be written as

$$\mathbf{T}^{-1} = \begin{bmatrix} [\mathbf{T}^{-1}]_{1,1} & 0 & 0 & 0 \\ 0 & [\mathbf{T}^{-1}]_{2,2} & 0 & 0 \\ 0 & 0 & [\mathbf{T}^{-1}]_{3,3} & 0 \\ 0 & 0 & 0 & [\mathbf{T}^{-1}]_{4,4} \end{bmatrix} + \boldsymbol{\Delta}_1,$$

where the all elements of $\boldsymbol{\Delta}_1$ are $o_{\mathbb{P}}(1)$. Thus the matrix $\mathbf{T}$ is equal to

$$\mathbf{T} = \begin{bmatrix} \frac{1}{[\mathbf{T}^{-1}]_{1,1}} & 0 & 0 & 0 \\ 0 & \frac{1}{[\mathbf{T}^{-1}]_{2,2}} & 0 & 0 \\ 0 & 0 & \frac{1}{[\mathbf{T}^{-1}]_{3,3}} & 0 \\ 0 & 0 & 0 & \frac{1}{[\mathbf{T}^{-1}]_{4,4}} \end{bmatrix} + \boldsymbol{\Delta}_2, \tag{28}$$

where the all elements of $\boldsymbol{\Delta}_2$ are $o_{\mathbb{P}}(1)$.

### K.2.2 Computing the training loss

Having computed the limit of the matrix $\mathbf{T}^{-1}$ and $\mathbf{T}$, we are now ready to put everything together and compute the limiting train loss. One can write the outcome vector $\tilde{\boldsymbol{y}}$ as $\tilde{\boldsymbol{y}} = \sigma_\star(\tilde{\mathbf{X}}\boldsymbol{\beta}_\star) + \boldsymbol{\varepsilon}$, where $\boldsymbol{\varepsilon} \in \mathbb{R}^n$ is the noise term. Thus, using equation 26, we find

$$\mathcal{L}_{\mathrm{tr}}(\mathbf{F}_0) - \mathcal{L}_{\mathrm{tr}}(\mathbf{F}) = \lambda \sigma_\star(\tilde{\mathbf{X}}\boldsymbol{\beta}_\star)^\top \bar{\mathbf{R}}_0 \mathbf{U} \mathbf{T} \mathbf{U}^\top \bar{\mathbf{R}}_0 \sigma_\star(\tilde{\mathbf{X}}\boldsymbol{\beta}_\star)$$
$$+ 2\lambda \sigma_\star(\tilde{\mathbf{X}}\boldsymbol{\beta}_\star)^\top \bar{\mathbf{R}}_0 \mathbf{U} \mathbf{T} \mathbf{U}^\top \bar{\mathbf{R}}_0 \boldsymbol{\varepsilon} + \lambda \boldsymbol{\varepsilon}^\top \bar{\mathbf{R}}_0 \mathbf{U} \mathbf{T} \mathbf{U}^\top \bar{\mathbf{R}}_0 \boldsymbol{\varepsilon}. \tag{29}$$

We will first argue the second and third term will go to zero in probability. To do this, we note that $\|\mathbf{T}\|_{\mathrm{op}} = O_{\mathbb{P}}(1)$ and also $\|\mathbf{U}^\top \bar{\mathbf{R}}_0\|_2 \le \|\mathbf{U}\|_{\mathrm{op}} \|\bar{\mathbf{R}}_0\|_{\mathrm{op}} = O_{\mathbb{P}}(1/\sqrt{n})$. We have $\boldsymbol{\varepsilon} \sim \mathsf{N}(0, \sigma_\varepsilon^2 \mathbf{I}_n)$ and it is independent of $\bar{\mathbf{R}}_0, \mathbf{U}, \mathbf{T}, \tilde{\mathbf{X}}$, and $\boldsymbol{\beta}_\star$. Also note that $\|\sigma_\star(\tilde{\mathbf{X}}\boldsymbol{\beta}_\star)^\top \bar{\mathbf{R}}_0 \mathbf{U}\|_2 = O_{\mathbb{P}}(1)$. Thus, the second and third term in equation 29 go to zero and we have

$$\mathcal{L}_{\mathrm{tr}}(\mathbf{F}_0) - \mathcal{L}_{\mathrm{tr}}(\mathbf{F}) = \lambda \sigma_\star(\tilde{\mathbf{X}}\boldsymbol{\beta}_\star)^\top \bar{\mathbf{R}}_0 \mathbf{U} \mathbf{T} \mathbf{U}^\top \bar{\mathbf{R}}_0 \sigma_\star(\tilde{\mathbf{X}}\boldsymbol{\beta}_\star) + o_{\mathbb{P}}(1).$$

If we expand $\sigma_\star(\tilde{\mathbf{X}}\boldsymbol{\beta}_\star)$ in the Hermite basis as $\sigma_\star(\tilde{\mathbf{X}}\boldsymbol{\beta}_\star) = \sum_{p=1}^\infty c_{\star,p} H_p(\tilde{\boldsymbol{\theta}}_\star)$, we can write

$$\mathcal{L}_{\mathrm{tr}}(\mathbf{F}_0) - \mathcal{L}_{\mathrm{tr}}(\mathbf{F}) = \lambda \sum_{p,q=1}^\infty c_{\star,p} c_{\star,q} H_p(\tilde{\boldsymbol{\theta}}_\star)^\top \bar{\mathbf{R}}_0 \mathbf{U} \mathbf{T} \mathbf{U}^\top \bar{\mathbf{R}}_0 H_q(\tilde{\boldsymbol{\theta}}_\star) + o_{\mathbb{P}}(1).$$

We define $\Delta_{p,q} = H_p(\tilde{\boldsymbol{\theta}}_\star)^\top \bar{\mathbf{R}}_0 \mathbf{U} \mathbf{T} \mathbf{U}^\top \bar{\mathbf{R}}_0 H_q(\tilde{\boldsymbol{\theta}}_\star) = \delta_1^{p,q} + \delta_2^{p,q} + \delta_3^{p,q} + \delta_4^{p,q}$ in which, with $T_{i,j}$ being the $(i,j)$-th elements of the matrix $\mathbf{T}$,

$$\delta_1^{p,q} = T_{1,1} H_p(\tilde{\boldsymbol{\theta}}_\star)^\top \bar{\mathbf{R}}_0 (\mathbf{F}_0 \boldsymbol{a}) (\mathbf{F}_0 \boldsymbol{a})^\top \bar{\mathbf{R}}_0 H_q(\tilde{\boldsymbol{\theta}}_\star)$$
$$+ T_{1,2} H_p(\tilde{\boldsymbol{\theta}}_\star)^\top \bar{\mathbf{R}}_0 (\mathbf{F}_0 \boldsymbol{a}) (\sqrt{N} \mathbf{F}_0 \boldsymbol{a}^{\circ 2})^\top \bar{\mathbf{R}}_0 H_q(\tilde{\boldsymbol{\theta}}_\star)$$
$$+ T_{1,3} H_p(\tilde{\boldsymbol{\theta}}_\star)^\top \bar{\mathbf{R}}_0 (\mathbf{F}_0 \boldsymbol{a}) \tilde{\boldsymbol{\theta}}^\top \bar{\mathbf{R}}_0 H_q(\tilde{\boldsymbol{\theta}}_\star)$$
$$+ T_{1,4} H_p(\tilde{\boldsymbol{\theta}}_\star)^\top \bar{\mathbf{R}}_0 (\mathbf{F}_0 \boldsymbol{a}) \tilde{\boldsymbol{\theta}}^{\circ 2 \top} \bar{\mathbf{R}}_0 H_q(\tilde{\boldsymbol{\theta}}_\star), \tag{30}$$

$$\delta_2^{p,q} = T_{2,1} H_p(\tilde{\boldsymbol{\theta}}_\star)^\top \bar{\mathbf{R}}_0 (\sqrt{N}\mathbf{F}_0 \boldsymbol{a}^{\circ 2})(\mathbf{F}_0 \boldsymbol{a})^\top \bar{\mathbf{R}}_0 H_q(\tilde{\boldsymbol{\theta}}_\star)$$
$$+ T_{2,2} H_p(\tilde{\boldsymbol{\theta}}_\star)^\top \bar{\mathbf{R}}_0 (\sqrt{N}\mathbf{F}_0 \boldsymbol{a}^{\circ 2})(\sqrt{N}\mathbf{F}_0 \boldsymbol{a}^{\circ 2})^\top \bar{\mathbf{R}}_0 H_q(\tilde{\boldsymbol{\theta}}_\star)$$
$$+ T_{2,3} H_p(\tilde{\boldsymbol{\theta}}_\star)^\top \bar{\mathbf{R}}_0 (\sqrt{N}\mathbf{F}_0 \boldsymbol{a}^{\circ 2})\tilde{\boldsymbol{\theta}}^\top \bar{\mathbf{R}}_0 H_q(\tilde{\boldsymbol{\theta}}_\star)$$
$$+ T_{2,4} H_p(\tilde{\boldsymbol{\theta}}_\star)^\top \bar{\mathbf{R}}_0 (\sqrt{N}\mathbf{F}_0 \boldsymbol{a}^{\circ 2})\tilde{\boldsymbol{\theta}}^{\circ 2\top} \bar{\mathbf{R}}_0 H_q(\tilde{\boldsymbol{\theta}}_\star), \tag{31}$$

$$\delta_3^{p,q} = T_{3,1} H_p(\tilde{\boldsymbol{\theta}}_\star)^\top \bar{\mathbf{R}}_0 \tilde{\boldsymbol{\theta}}(\mathbf{F}_0 \boldsymbol{a})^\top \bar{\mathbf{R}}_0 H_q(\tilde{\boldsymbol{\theta}}_\star)$$
$$+ T_{3,2} H_p(\tilde{\boldsymbol{\theta}}_\star)^\top \bar{\mathbf{R}}_0 \tilde{\boldsymbol{\theta}}(\sqrt{N}\mathbf{F}_0 \boldsymbol{a}^{\circ 2})^\top \bar{\mathbf{R}}_0 H_q(\tilde{\boldsymbol{\theta}}_\star)$$
$$+ T_{3,3} H_p(\tilde{\boldsymbol{\theta}}_\star)^\top \bar{\mathbf{R}}_0 \tilde{\boldsymbol{\theta}}\tilde{\boldsymbol{\theta}}^\top \bar{\mathbf{R}}_0 H_q(\tilde{\boldsymbol{\theta}}_\star)$$
$$+ T_{3,4} H_p(\tilde{\boldsymbol{\theta}}_\star)^\top \bar{\mathbf{R}}_0 \tilde{\boldsymbol{\theta}}\tilde{\boldsymbol{\theta}}^{\circ 2\top} \bar{\mathbf{R}}_0 H_q(\tilde{\boldsymbol{\theta}}_\star), \tag{32}$$

and

$$\delta_4^{p,q} = T_{4,1} H_p(\tilde{\boldsymbol{\theta}}_\star)^\top \bar{\mathbf{R}}_0 \tilde{\boldsymbol{\theta}}^{\circ 2}(\mathbf{F}_0 \boldsymbol{a})^\top \bar{\mathbf{R}}_0 H_q(\tilde{\boldsymbol{\theta}}_\star)$$
$$+ T_{4,2} H_p(\tilde{\boldsymbol{\theta}}_\star)^\top \bar{\mathbf{R}}_0 \tilde{\boldsymbol{\theta}}^{\circ 2}(\sqrt{N}\mathbf{F}_0 \boldsymbol{a}^{\circ 2})^\top \bar{\mathbf{R}}_0 H_q(\tilde{\boldsymbol{\theta}}_\star)$$
$$+ T_{4,3} H_p(\tilde{\boldsymbol{\theta}}_\star)^\top \bar{\mathbf{R}}_0 \tilde{\boldsymbol{\theta}}^{\circ 2}\tilde{\boldsymbol{\theta}}^\top \bar{\mathbf{R}}_0 H_q(\tilde{\boldsymbol{\theta}}_\star)$$
$$+ T_{4,4} H_p(\tilde{\boldsymbol{\theta}}_\star)^\top \bar{\mathbf{R}}_0 \tilde{\boldsymbol{\theta}}^{\circ 2}\tilde{\boldsymbol{\theta}}^{\circ 2\top} \bar{\mathbf{R}}_0 H_q(\tilde{\boldsymbol{\theta}}_\star). \tag{33}$$

We will now look at each $\delta_i^{p,q}$ for $i \in \{1, 2, 3, 4\}$.

**Term $\delta_1^{p,q}$:** To prove that the term in equation 30 are asymptotically negligible, note that $\boldsymbol{a} \sim \mathsf{N}(0, \frac{1}{N}\mathbf{I}_N)$ is independent of $H_p(\tilde{\boldsymbol{\theta}}_\star)\bar{\mathbf{R}}_0\mathbf{F}_0$ and we have $\|H_p(\tilde{\boldsymbol{\theta}}_\star)\bar{\mathbf{R}}_0\mathbf{F}_0\|_2 = O_\mathbb{P}(1)$. Thus, $H_p(\tilde{\boldsymbol{\theta}}_\star)\bar{\mathbf{R}}_0\mathbf{F}_0\boldsymbol{a} = o_\mathbb{P}(1)$ and all other terms multiplying this are $O_\mathbb{P}(1)$. This implies that for any $p, q \in \mathbb{N}$, we have $\delta_1^{p,q} = o_\mathbb{P}(1)$.

**Term $\delta_2^{p,q}$:** All four terms in equation 31 converge to zero. To prove this, we will use the Lemma K.6. In equation 31, all terms multiplied by $\sqrt{N}H_p(\tilde{\boldsymbol{\theta}}_\star)\bar{\mathbf{R}}_0\mathbf{F}_0\boldsymbol{a}^{\circ 2}$ are $O_\mathbb{P}(1)$. Thus, $\delta_2^{p,q} = o_\mathbb{P}(1)$ for any $p, q \in \mathbb{N}$.

**Term $\delta_3^{p,q}$:** The first term in equation 32 converges to zero in probability due to an argument similar to the arguments used for $\delta_1^{p,q}$; and the same holds for the second term in equation 32, by arguing similarly as for $\delta_2^{p,q}$. We have shown that $T_{3,4} = o_\mathbb{P}(1)$, and by a norm argument, we can see that $H_p(\tilde{\boldsymbol{\theta}}_\star)^\top\bar{\mathbf{R}}_0\tilde{\boldsymbol{\theta}}$ and $\tilde{\boldsymbol{\theta}}^{\circ 2\top}\bar{\mathbf{R}}_0 H_q(\tilde{\boldsymbol{\theta}}_\star)$ are $O_\mathbb{P}(1)$. Hence,

$$\delta_3^{p,q} = T_{3,3}\big(H_p(\tilde{\boldsymbol{\theta}}_\star)^\top\bar{\mathbf{R}}_0\tilde{\boldsymbol{\theta}}\big)\big(\tilde{\boldsymbol{\theta}}^\top\bar{\mathbf{R}}_0 H_q(\tilde{\boldsymbol{\theta}}_\star)\big) + o_\mathbb{P}(1).$$

**Term $\delta_4^{p,q}$:** The first two terms in equation 33 converge to zero by the same reasoning used for $\delta_1^{p,q}$ and $\delta_2^{p,q}$, respectively. The third term can also be shown to converge to zero by recalling that $T_{4,3} = o_\mathbb{P}(1)$. Hence, we can write

$$\delta_4^{p,q} = T_{4,4}\big(H_p(\tilde{\boldsymbol{\theta}}_\star)^\top\bar{\mathbf{R}}_0\tilde{\boldsymbol{\theta}}^{\circ 2}\big)\big(\tilde{\boldsymbol{\theta}}^{\circ 2\top}\bar{\mathbf{R}}_0 H_q(\tilde{\boldsymbol{\theta}}_\star)\big) + o_\mathbb{P}(1).$$

Putting everything together, we find

$$L_{\mathrm{tr}}(\mathbf{F}_0) - L_{\mathrm{tr}}(\mathbf{F}) = \lambda T_{3,3} \sum_{p,q=1}^\infty c_{\star,p}c_{\star,q}(H_p(\tilde{\boldsymbol{\theta}}_\star)^\top\bar{\mathbf{R}}_0\tilde{\boldsymbol{\theta}})\big(\tilde{\boldsymbol{\theta}}^\top\bar{\mathbf{R}}_0 H_q(\tilde{\boldsymbol{\theta}}_\star)\big)$$
$$+ \lambda T_{4,4} \sum_{p,q=1}^\infty c_{\star,p}c_{\star,q}\big(H_p(\tilde{\boldsymbol{\theta}}_\star)^\top\bar{\mathbf{R}}_0\tilde{\boldsymbol{\theta}}^{\circ 2}\big)\big(\tilde{\boldsymbol{\theta}}^{\circ 2\top}\bar{\mathbf{R}}_0 H_q(\tilde{\boldsymbol{\theta}}_\star)\big) + o_\mathbb{P}(1).$$

Using Lemma K.7, we know that in the sums above, the terms corresponding to $(p, q) = (1, 1)$ and $(p, q) = (2, 2)$ are the only non-negligible terms in the first and second sum respectively.

Hence, as $T_{3,3} = 1/(\tilde{\boldsymbol{\theta}}^\top \bar{\mathbf{R}}_0 \tilde{\boldsymbol{\theta}}) + o_{\mathbb{P}}(1)$ and $T_{4,4} = 1/(\tilde{\boldsymbol{\theta}}^{\circ 2\top} \bar{\mathbf{R}}_0 \tilde{\boldsymbol{\theta}}^{\circ 2}) + o_{\mathbb{P}}(1)$, from Lemmas K.1, K.4, K.7 and K.8, we can write,

$$
\mathcal{L}_{\mathrm{tr}}(\mathbf{F}) - \mathcal{L}_{\mathrm{tr}}(\mathbf{F}_0) = \lambda T_{3,3} c_{\star,1}^2 \big(\tilde{\boldsymbol{\theta}}_\star^\top \bar{\mathbf{R}}_0 \tilde{\boldsymbol{\theta}}\big)^2 + \lambda T_{4,4} c_{\star,2}^2 \big(H_2(\tilde{\boldsymbol{\theta}}_\star)^\top \bar{\mathbf{R}}_0 \tilde{\boldsymbol{\theta}}^{\circ 2}\big)^2 + o_{\mathbb{P}}(1)
$$

$$
= \lambda \frac{c_{\star,1}^2 \big(\tilde{\boldsymbol{\theta}}_\star^\top \bar{\mathbf{R}}_0 \tilde{\boldsymbol{\theta}}\big)^2}{\tilde{\boldsymbol{\theta}}^\top \bar{\mathbf{R}}_0 \tilde{\boldsymbol{\theta}}} + \lambda c_{\star,2}^2 \frac{\big(H_2(\tilde{\boldsymbol{\theta}}_\star)^\top \bar{\mathbf{R}}_0 \tilde{\boldsymbol{\theta}}^{\circ 2}\big)^2}{\tilde{\boldsymbol{\theta}}^{\circ 2\top} \bar{\mathbf{R}}_0 \tilde{\boldsymbol{\theta}}^{\circ 2}} + o_{\mathbb{P}}(1)
$$

$$
\to_P \Delta_2 = \frac{\psi \lambda c_{\star,1}^4 m_2}{\phi[c_{\star,1}^2 + \phi(c_\star^2 + \sigma_\varepsilon^2)]} + \frac{4\psi \lambda c_{\star,1}^4 c_{\star,2}^2 m_1}{3\phi[\phi(c_\star^2 + \sigma_\varepsilon^2) + c_{\star,1}^2]^2},
$$

proving the theorem for $\ell = 2$.

## L  ASYMPTOTICS OF THE TRAINING LOSS FOR GENERAL $\ell$

We define the values $\xi_{i,j}$ for all $i, j \in \{0, 1, \ldots\}$ such that for any $p \in \mathbb{N}$ and $x \in \mathbb{R}$, we have $x^p = \sum_{i=0}^p \xi_{p,i} H_i(x)$.

**Theorem L.1.** *Let $\ell \in \mathbb{N}$. If Conditions 2.1-2.4 and the Gaussian equivalence conjecture 4.3 hold, while we also have $c_1, \cdots, c_\ell \neq 0$, and $\eta \asymp n^\alpha$ with $\frac{\ell-1}{2\ell} < \alpha < \frac{\ell}{2\ell+2}$, then for the learned feature map $\mathbf{F}$ and the untrained feature map $\mathbf{F}_0$, we have $\mathcal{L}_{\mathrm{tr}}(\mathbf{F}_0) - \mathcal{L}_{\mathrm{tr}}(\mathbf{F}) \to_P \Delta_\ell > 0$, where*

$$
\Delta_\ell = \lambda \sum_{p=1}^\ell \sum_{q=1}^\ell c_{\star,p} c_{\star,q} r_p r_q \sum_{i=1}^\ell \sum_{j=1}^\ell \boldsymbol{\Omega}_{i,j} \big(\phi(c_\star^2 + \sigma_\varepsilon^2) + c_{\star,1}^2\big)^{(i+j)/2} \xi_{i,p} \xi_{j,q} + o_{\mathbb{P}}(1),
$$

*in which $\boldsymbol{\Omega}$ is an invertible matrix with*

$$
[\boldsymbol{\Omega}^{-1}]_{i,j} = \big(c_{\star,1}^2 + \phi(c_\star^2 + \sigma_\varepsilon^2)\big)^{(i+j)/2} \frac{\psi}{\phi} \left[ m_2 \xi_{i,1} \xi_{j,1} + m_1 \sum_{k=0,\ k\neq 1}^{\min(i,j)} k!\, \xi_{i,k} \xi_{j,k} \right], \quad \forall i, j \in [\ell],
$$

*and for $p \in \mathbb{N}$,*

$$
r_p = \begin{cases} \frac{p! \psi m_1}{\phi} \left( \frac{c_{\star,1}}{\sqrt{\phi(c_\star^2 + \sigma_\varepsilon^2) + c_{\star,1}^2}} \right)^p & p \neq 1 \\ \frac{\psi m_2}{\phi} \frac{c_{\star,1}}{\sqrt{\phi(c_\star^2 + \sigma_\varepsilon^2) + c_{\star,1}^2}} & p = 1 \end{cases}
$$

*Proof of Theorem L.1.* In the regime where $\eta \asymp n^\alpha$ with $\frac{\ell-1}{2\ell} < \alpha < \frac{\ell}{2\ell+2}$, according to the equivalence theorem 4.1, we can replace $\mathbf{F}$ with $\mathbf{F}_\ell$ when computing the limiting training loss. To compute the limiting training loss difference according to lemma K.2, we study the matrix $\bar{\mathbf{R}} = (\mathbf{FF}^\top + \lambda n \mathbf{I}_n)^{-1}$. Due to equation 3, we can write

$$
\mathbf{FF}^\top = \mathbf{F}_0 \mathbf{F}_0^\top + \sum_{k=1}^\ell c_1^k c_k \eta^k \tilde{\boldsymbol{\theta}}^{\circ k} (\mathbf{F}_0 \boldsymbol{a}^{\circ k})^\top
$$

$$
+ \sum_{k=1}^\ell c_1^k c_k \eta^k (\mathbf{F}_0 \boldsymbol{a}^{\circ k}) \tilde{\boldsymbol{\theta}}^{\circ k\top} + \sum_{j=1}^\ell \sum_{i=1}^\ell c_1^{i+j} c_i c_j \eta^{i+j} (\boldsymbol{a}^{\circ i})^\top (\boldsymbol{a}^{\circ j}) \tilde{\boldsymbol{\theta}}^{\circ i} \tilde{\boldsymbol{\theta}}^{\circ j\top}.
$$

Defining the matrix $\mathbf{U}$ as

$$
\mathbf{U} = \left[ \underbrace{\mathbf{F}_0 \boldsymbol{a} \mid \cdots \mid N^{(\ell-1)/2} \mathbf{F}_0 \boldsymbol{a}^{\circ \ell}}_{\ell \text{ columns}} \mid \underbrace{\tilde{\boldsymbol{\theta}} \mid \cdots \mid \tilde{\boldsymbol{\theta}}^{\circ \ell}}_{\ell \text{ columns}} \right] \in \mathbb{R}^{n \times 2\ell},
$$

we can write

$$
\mathbf{FF}^\top = \mathbf{F}_0 \mathbf{F}_0^\top + \mathbf{UKU}^\top, \text{ in which } \mathbf{K} = \begin{bmatrix} \mathbf{0}_{\ell \times \ell} & \mathbf{K}_o \\ \mathbf{K}_o & \tilde{\mathbf{K}} \end{bmatrix} \in \mathbb{R}^{2\ell \times 2\ell},
$$

where $\mathbf{K}_o = \mathrm{diag}\left(\frac{c_1 c_1 \eta}{N^0}, \ldots, \frac{c_1^\ell c_\ell \eta^\ell}{N^{(\ell-1)/2}}\right) \in \mathbb{R}^{\ell \times \ell}$, and $\tilde{\mathbf{K}} \in \mathbb{R}^{\ell \times \ell}$ with $[\tilde{\mathbf{K}}]_{i,j} = c_1^{i+j} c_i c_j \eta^{i+j} \langle \boldsymbol{a}^{\circ i}, \boldsymbol{a}^{\circ j} \rangle$, for all $i, j \in [\ell]$.

Using the Woodbury formula, the matrix $\bar{\mathbf{R}}$ can be written in terms of $\bar{\mathbf{R}}_0 = (\mathbf{F}_0 \mathbf{F}_0^\top + \lambda n \mathbf{I}_n)^{-1}$ and $\mathbf{T} = (\mathbf{K}^{-1} + \mathbf{U}^\top \bar{\mathbf{R}}_0 \mathbf{U})^{-1} \in \mathbb{R}^{2\ell \times 2\ell}$ as $\bar{\mathbf{R}} = \bar{\mathbf{R}}_0 - \bar{\mathbf{R}}_0 \mathbf{U} \mathbf{T} \mathbf{U}^\top \bar{\mathbf{R}}_0$. Now

$$\mathbf{K}^{-1} = \begin{bmatrix} \hat{\mathbf{K}} & \mathbf{K}_o^{-1} \\ \mathbf{K}_o^{-1} & \mathbf{0}_{\ell \times \ell} \end{bmatrix}, \quad \text{where} \quad \mathbf{K}_o^{-1} = \mathrm{diag}\left(\frac{N^0}{c_1 c_1 \eta}, \ldots, \frac{N^{\frac{\ell-1}{2}}}{c_1^\ell c_\ell \eta^\ell}\right),$$

and $[\hat{\mathbf{K}}]_{i,j} = -N^{(i-1)/2} N^{(j-1)/2} \langle \boldsymbol{a}^{\circ i}, \boldsymbol{a}^{\circ j} \rangle$, for all $i, j \in [\ell]$. We define $\mathbf{M}_1, \mathbf{M}_2, \mathbf{M}_o \in \mathbb{R}^{\ell \times \ell}$ as the following blocks of $\mathbf{T}^{-1}$:

$$\mathbf{T}^{-1} = \begin{bmatrix} \mathbf{M}_1 & \mathbf{M}_o \\ \mathbf{M}_o & \mathbf{M}_2 \end{bmatrix}.$$

Hence, we have

$$\begin{cases} [\mathbf{M}_1]_{i,j} = N^{(i-1)/2} N^{(j-1)/2} \boldsymbol{a}^{\circ i \top} (\mathbf{F}_0^\top \bar{\mathbf{R}}_0 \mathbf{F}_0 - \mathbf{I}) \boldsymbol{a}^{\circ j}, \\[2mm] [\mathbf{M}_o]_{i,j} = N^{(i-1)/2} \boldsymbol{a}^{\circ i \top} \mathbf{F}_0^\top \bar{\mathbf{R}}_0 \tilde{\boldsymbol{\theta}}^{\circ j} + o_{\mathbb{P}}(1), \\[2mm] [\mathbf{M}_2]_{i,j} = \tilde{\boldsymbol{\theta}}^{\circ i \top} \bar{\mathbf{R}}_0 \tilde{\boldsymbol{\theta}}^{\circ j}. \end{cases}$$

We can expand the monomials in terms of the Hermite polynomials, for scalars $\xi_{i,k}$, $k \in [i]$, as follows:

$$(N^{1/2} \boldsymbol{a})^{\circ i} = \sum_{k=0}^i \xi_{i,k} H_k(N^{1/2} \boldsymbol{a}), \quad \text{and} \quad (\tilde{\mathbf{X}} \boldsymbol{\beta})^{\circ i} = \|\boldsymbol{\beta}\|_2^i \sum_{k=0}^i \xi_{i,k} H_k(\tilde{\mathbf{X}} \boldsymbol{\beta} / \|\boldsymbol{\beta}\|_2).$$

Using these, we will analyze each matrix $\mathbf{M}_1, \mathbf{M}_2, \mathbf{M}_o$ separately.

**Analysis of $\mathbf{M}_1$.** It is easily seen that the elements of this matrix are $O_{\mathbb{P}}(1)$.

**Analysis of $\mathbf{M}_2$.** To analyze these terms, we need the following lemma, whose proof is deferred to Section N.10.

**Lemma L.2.** *For any $i, j \in \mathbb{N}_0$, we have*

$$(\tilde{\mathbf{X}} \boldsymbol{\beta})^{\circ i \top} \bar{\mathbf{R}}_0 (\tilde{\mathbf{X}} \boldsymbol{\beta})^{\circ j} \to_P \left(c_{\star,1}^2 + \phi(c_\star^2 + \sigma_\varepsilon^2)\right)^{(i+j)/2} \left[\xi_{i,1} \xi_{j,1} \frac{\psi m_2}{\phi} + \frac{\psi m_1}{\phi} \sum_{k=0,\, k \neq 1}^{\min(i,j)} k!\, \xi_{i,k} \xi_{j,k}\right].$$

Defining the matrix $\bar{\mathbf{M}}_2 \in \mathbb{R}^{\ell \times \ell}$ with entries

$$[\bar{\mathbf{M}}_2]_{i,j} = \left(c_{\star,1}^2 + \phi(c_\star^2 + \sigma_\varepsilon^2)\right)^{(i+j)/2} \left[\xi_{i,1} \xi_{j,1} \frac{\psi m_2}{\phi} + \frac{\psi m_1}{\phi} \sum_{k=0,\, k \neq 1}^{\min(i,j)} k!\, \xi_{i,k} \xi_{j,k}\right],$$

for all $i, j \in [\ell]$, we have $[\mathbf{M}_2]_{i,j} \to_P [\bar{\mathbf{M}}_2]_{i,j}$. Note that we can write

$$\bar{\mathbf{M}}_2 = \frac{\psi}{\phi} \mathbf{B} \mathbf{Z} \mathbf{M} \mathbf{Z}^\top \mathbf{B} + \frac{\psi m_1}{\phi} \boldsymbol{e} \boldsymbol{e}^\top,$$

where we define $b = (c_{\star,1}^2 + \phi(c_\star^2 + \sigma_\varepsilon^2))^{1/2}$, $\mathbf{B} = \mathrm{diag}(b^1, \cdots, b^\ell) \in \mathbb{R}^{\ell \times \ell}$, $\boldsymbol{e} = \mathbf{B}[\xi_{1,0}, \cdots, \xi_{\ell,0}]^\top$,

$$\mathbf{M} = \begin{bmatrix} 1!\, m_2 & 0 & \cdots & 0 \\ 0 & 2!\, m_1 & \cdots & 0 \\ \vdots & \vdots & \ddots & \vdots \\ 0 & 0 & \cdots & \ell!\, m_1 \end{bmatrix} \in \mathbb{R}^{\ell \times \ell}, \quad \text{and} \quad \mathbf{Z} = \begin{bmatrix} \xi_{1,1} & \cdots & \xi_{1,\ell} \\ \vdots & \ddots & \vdots \\ \xi_{\ell,1} & \cdots & \xi_{\ell,\ell} \end{bmatrix} \in \mathbb{R}^{\ell \times \ell}.$$

Recalling that for all $i, j \in \{0, 1, \ldots\}$, $\xi_{i,j}$ are such that such that for any $p \in \mathbb{N}$ and $x \in \mathbb{R}$, we have $x^p = \sum_{i=0}^p \xi_{p,i} H_i(x)$, it follows that the matrix $\mathbf{Z}$ is lower-triangular with unit diagonal; hence invertible. Thus, since $\mathbf{B}, \mathbf{M}$ are diagonal with positive entries, the matrix $\mathbf{B} \mathbf{Z} \mathbf{M} \mathbf{Z}^\top \mathbf{B}$ is positive definite. This implies that $\bar{\mathbf{M}}_2$ is invertible. We will denote $\boldsymbol{\Omega} = \bar{\mathbf{M}}_2^{-1}$.

**Analysis of $\mathbf{M}_o$.** We analyze $[\mathbf{M}_o]_{i,j}$ by writing $N^{(i-1)/2}\boldsymbol{a}^{\circ i}$ in the Hermite basis, finding

$$[\mathbf{M}_o]_{i,j} = \sum_{k=0}^{i} \frac{\xi_{i,k}}{\sqrt{N}} H_k(N^{1/2}\boldsymbol{a})^\top \mathbf{F}_0^\top \bar{\mathbf{R}}_0 \tilde{\boldsymbol{\theta}}^{\circ j} + o_\mathbb{P}(1).$$

The terms with $k > 0$ are all $o_\mathbb{P}(1)$ because $\frac{H_k(N^{1/2}\boldsymbol{a})}{\sqrt{N}}$ is a norm $O_\mathbb{P}(1)$ vector with mean zero, independent from the vector $\mathbf{F}_0^\top \bar{\mathbf{R}}_0 \tilde{\boldsymbol{\theta}}^{\circ j}$ with norm $O_\mathbb{P}(1)$. Thus, $[\mathbf{M}_o]_{i,j} = o_\mathbb{P}(1)$. The term with $k = 0$ can also be shown to be $o_\mathbb{P}(1)$ by using the fact that the linearized $\mathbf{F}_0$ is left-orthogonally invariant, via an argument identical to the one used to analyze equation 27.

Hence, putting these together, the matrix $\mathbf{T}$ can be written as

$$\mathbf{T} = \begin{bmatrix} \mathbf{M}_1^{-1} & \mathbf{0}_{\ell \times \ell} \\ \mathbf{0}_{\ell \times \ell} & \mathbf{M}_2^{-1} \end{bmatrix} + o_\mathbb{P}(1).$$

Using lemma K.2, we can write the training loss difference as $\mathcal{L}_{\mathrm{tr}}(\mathbf{F}_0) - \mathcal{L}_{\mathrm{tr}}(\mathbf{F}) = \lambda \boldsymbol{y}^\top \bar{\mathbf{R}}_0 \mathbf{U}\mathbf{T}\mathbf{U}^\top \bar{\mathbf{R}}_0 \boldsymbol{y}$. Plugging in the teacher function $f_\star$, we find

$$\mathcal{L}_{\mathrm{tr}}(\mathbf{F}_0) - \mathcal{L}_{\mathrm{tr}}(\mathbf{F}) = \sum_{p,q} \lambda c_{\star,p} c_{\star,q} H_p(\tilde{\boldsymbol{\theta}}_\star)^\top \bar{\mathbf{R}}_0 \mathbf{U}\mathbf{T}\mathbf{U}^\top \bar{\mathbf{R}}_0 H_q(\tilde{\boldsymbol{\theta}}_\star)$$

$$+ 2\lambda \sum_p \left( c_{\star,p} H_p(\tilde{\boldsymbol{\theta}}_\star)^\top \bar{\mathbf{R}}_0 \mathbf{U}\mathbf{T}\mathbf{U}^\top \bar{\mathbf{R}}_0 \boldsymbol{\varepsilon} \right) + \lambda \boldsymbol{\varepsilon}^\top \bar{\mathbf{R}}_0 \mathbf{U}\mathbf{T}\mathbf{U}^\top \bar{\mathbf{R}}_0 \boldsymbol{\varepsilon}.$$

Note that the second term can be shown to be $o_\mathbb{P}(1)$ because $\boldsymbol{\varepsilon} \sim \mathsf{N}(0, \sigma_\varepsilon^2 \mathbf{I}_n)$ and it is independent from $H_p(\tilde{\boldsymbol{\theta}}_\star)^\top \bar{\mathbf{R}}_0 \mathbf{U}\mathbf{T}\mathbf{U}^\top \bar{\mathbf{R}}_0$, and $\|H_p(\tilde{\boldsymbol{\theta}}_\star)^\top \bar{\mathbf{R}}_0 \mathbf{U}\mathbf{T}\mathbf{U}^\top \bar{\mathbf{R}}_0\|_{\mathrm{op}} = O_\mathbb{P}(1/\sqrt{N})$ with a simple order-wise analysis. The third can also be shown to be $o_\mathbb{P}(1)$ by noting that $\boldsymbol{\varepsilon}$ is independent from $\bar{\mathbf{R}}_0 \mathbf{U}$, $\|\bar{\mathbf{R}}_0 \mathbf{U}\|_{\mathrm{op}} = O_\mathbb{P}(1/\sqrt{n})$ and the fact that the elements of $\mathbf{T}$ are $O_\mathbb{P}(1)$.

To analyze the first term, we define $\delta_{p,q} = H_p(\tilde{\boldsymbol{\theta}}_\star)^\top \bar{\mathbf{R}}_0 \mathbf{U}\mathbf{T}\mathbf{U}^\top \bar{\mathbf{R}}_0 H_q(\tilde{\boldsymbol{\theta}}_\star)$ for all non-negative integers $p, q$. To analyze such terms, we first expand $\mathbf{U}\mathbf{T}\mathbf{U}^\top$ as

$$\mathbf{U}\mathbf{T}\mathbf{U}^\top = \sum_{i=1}^{\ell} \sum_{j=1}^{\ell} N^{(i+j)/2-1}[\mathbf{M}_1^{-1}]_{i,j}(\mathbf{F}_0\boldsymbol{a}^{\circ i})(\mathbf{F}_0\boldsymbol{a}^{\circ j})^\top + \sum_{i=1}^{\ell} \sum_{j=1}^{\ell} [\bar{\mathbf{M}}_2^{-1}]_{i,j}\tilde{\boldsymbol{\theta}}^{\circ i}\tilde{\boldsymbol{\theta}}^{\circ j \top}.$$

Thus, for any $p, q \in \mathbb{N}_0$, the terms $\delta_{p,q}$ can be written as

$$\delta_{p,q} = \sum_{i=1}^{\ell} \sum_{j=1}^{\ell} N^{(i+j)/2-1}[\mathbf{M}_1^{-1}]_{i,j} H_p(\tilde{\boldsymbol{\theta}}_\star)^\top \bar{\mathbf{R}}_0 (\mathbf{F}_0\boldsymbol{a}^{\circ i})(\mathbf{F}_0\boldsymbol{a}^{\circ j})^\top \bar{\mathbf{R}}_0 H_q(\tilde{\boldsymbol{\theta}}_\star)$$

$$+ \sum_{i=1}^{\ell} \sum_{j=1}^{\ell} [\bar{\mathbf{M}}_2^{-1}]_{i,j} H_p(\tilde{\boldsymbol{\theta}}_\star)^\top \bar{\mathbf{R}}_0 \tilde{\boldsymbol{\theta}}^{\circ i}\tilde{\boldsymbol{\theta}}^{\circ j \top}\bar{\mathbf{R}}_0 H_q(\tilde{\boldsymbol{\theta}}_\star).$$

By an argument identical to the argument for the terms in $\mathbf{M}_o$, the first sum goes to zero in probability. Denoting $\boldsymbol{\beta}/\|\boldsymbol{\beta}\|_2 := \tilde{\boldsymbol{\beta}}$, we can expand $(\tilde{\mathbf{X}}\boldsymbol{\beta})^{\circ i} = \|\boldsymbol{\beta}\|_2^i \sum_{k=0}^{i} \xi_{i,k} H_k(\tilde{\mathbf{X}}\boldsymbol{\beta}/\|\boldsymbol{\beta}\|_2)$, To analyze $\delta_{p,q}$, we need the following result, whose proof is deferred to Section N.11.

**Lemma L.3.** *For any $p, q \in \mathbb{N}_0$, we have*

$$H_p(\tilde{\mathbf{X}}\boldsymbol{\beta}_\star)^\top \bar{\mathbf{R}}_0 H_q(\tilde{\mathbf{X}}\tilde{\boldsymbol{\beta}}) \to_P \begin{cases} \frac{p!\psi m_1}{\phi} \left( \frac{c_{\star,1}}{\sqrt{\phi(c_\star^2+\sigma_\varepsilon^2)+c_{\star,1}^2}} \right)^p & p = q \neq 1 \\ \frac{\psi m_2}{\phi} \frac{c_{\star,1}}{\sqrt{\phi(c_\star^2+\sigma_\varepsilon^2)+c_{\star,1}^2}} & p = q = 1 \\ 0 & p \neq q. \end{cases}$$

We can now use Lemma L.3 and the fact that $\|\boldsymbol{\beta}\|_2 \to_P \left( \phi(c_\star^2 + \sigma_\varepsilon^2) + c_{\star,1}^2 \right)^{1/2}$ to write

$$\delta_{p,q} = \sum_{i=1}^{\ell} \sum_{j=1}^{\ell} [\bar{\mathbf{M}}_2^{-1}]_{i,j}\|\boldsymbol{\beta}\|_2^{i+j}\xi_{i,p}\xi_{j,q} H_p(\tilde{\boldsymbol{\theta}}_\star)^\top \bar{\mathbf{R}}_0 H_p(\tilde{\mathbf{X}}\tilde{\boldsymbol{\beta}}) \cdot H_q(\tilde{\mathbf{X}}\tilde{\boldsymbol{\beta}})^\top \bar{\mathbf{R}}_0 H_q(\tilde{\boldsymbol{\theta}}_\star) + o_\mathbb{P}(1)$$

$$= \sum_{i=1}^{\ell} \sum_{j=1}^{\ell} [\boldsymbol{\Omega}]_{i,j} \left( \phi(c_\star^2 + \sigma_\varepsilon^2) + c_{\star,1}^2 \right)^{(i+j)/2} \xi_{i,p}\xi_{j,q} r_p r_q + o_\mathbb{P}(1),$$

for $p, q \in [\ell]$, which concludes the proof. □

## M  INFINITE SAMPLE LIMIT

In the infinite sample limit, where $n \gg N, d$, we have $\phi \to 0$. In this extreme case, the expressions for $m_1, m_2$ will further simplify as $m_1, m_2 \to \phi/\lambda\psi$. Note that in this limit, we have $\mathcal{L}_{\mathrm{tr}}(\mathbf{F}_0) \to \sigma_\varepsilon^2 + c_\star^2$ (see e.g., (Mei & Montanari, 2022, Section 6). Using Corollary 4.5, we see that for example when $\ell = 2$, we have $\mathcal{L}(\mathbf{F}) \to \sigma_\varepsilon^2 + \frac{2c_{\star,2}^2}{3} + c_{\star,>2}^2$. In particular, the term corresponding to the linear component of the teacher function in $\mathcal{L}(\mathbf{F}_0)$ cancels out with the corresponding term in $\Delta_2$.

## N  PROOFS OF SUPPLEMENTARY LEMMAS

### N.1  PROOF OF LEMMA C.3

Recalling $a_i \overset{i.i.d.}{\sim} \mathsf{N}(0, 1/N)$ and $\langle \tilde{\boldsymbol{x}}_i, \boldsymbol{\beta} \rangle | \boldsymbol{\beta} \overset{i.i.d.}{\sim} \mathsf{N}(0, \|\boldsymbol{\beta}\|_2^2)$, claims (a) and (b) follow from standard Gaussian maximal inequalities (van der Vaart & Wellner, 2013, Section 2.2) and from $\|\boldsymbol{\beta}\|_2^2 = O_\mathbb{P}(1)$; the latter follows by writing $\boldsymbol{\beta} = n^{-1}\mathbf{X}^\top(\sigma_\star(\mathbf{X}\boldsymbol{\beta}_\star) + \varepsilon)$, where $\varepsilon = (\varepsilon_1, \ldots, \varepsilon_n)^\top$ and using our distributional assumptions on $\mathbf{X}, \varepsilon$, as well as Condition 2.2.

By (Vershynin, 2012, Theorem 5.39) and (Bai & Silverstein, 2010, Corollary A.21), we have $\|\mathbf{W}_0\mathbf{W}_0^\top\|_{\mathrm{op}}, \|(\mathbf{W}_0\mathbf{W}_0^\top)^{\circ 2}\|_{\mathrm{op}} = O_\mathbb{P}(1)$. Also, by (Vershynin, 2018, Theorem 3.4.6) and Gaussian maximal inequalities (van der Vaart & Wellner, 2013, Section 2.2), we have $\max_{1 \leq i \neq \leq j \leq N} \langle \boldsymbol{w}_{0,i}, \boldsymbol{w}_{0,j} \rangle = O_\mathbb{P}(n^{-\frac{1}{2}} \log^{\frac{1}{2}} n)$. For $k \geq 3$,

$$\|(\mathbf{W}_0\mathbf{W}_0^\top)^{\circ k}\|_{\mathrm{op}} \leq \|(\mathbf{W}_0\mathbf{W}_0^\top)^{\circ k} - \mathbf{I}_N\|_{\mathrm{op}} + 1 \leq \|(\mathbf{W}_0\mathbf{W}_0^\top)^{\circ k} - \mathbf{I}_N\|_{\mathrm{F}} + 1$$

$$\leq \left( \sum_{1 \leq i \neq j \leq N} \langle \boldsymbol{w}_{0,i}, \boldsymbol{w}_{0,j} \rangle^{2k} \right)^{\frac{1}{2}} + 1 = o_\mathbb{P}(1) + 1.$$

Therefore,

$$M_{W_0} \leq \max \left\{ \|\mathbf{W}_0\mathbf{W}_0^\top\|_{\mathrm{op}}, \|(\mathbf{W}_0\mathbf{W}_0^\top)^{\circ 2}\|_{\mathrm{op}}, \sup_{k \geq 3} \|(\mathbf{W}_0\mathbf{W}_0^\top)^{\circ k}\|_{\mathrm{op}} \right\} = O_\mathbb{P}(1).$$

Claim (d) is standard, see e.g. (Vershynin, 2018, Theorem 4.4.5).

### N.2  PROOF OF LEMMA K.1

We can write

$$\mathbf{v}^\top(\boldsymbol{\beta} - c_{\star,1}\boldsymbol{\beta}_\star) = n^{-1}\mathbf{v}^\top(\mathbf{X}^\top(\sigma_\star(\mathbf{X}\boldsymbol{\beta}_\star) + \varepsilon)) - c_{\star,1}\boldsymbol{\beta}_\star$$

$$= n^{-1}\sum_{i=1}^{n}(\mathbf{v}^\top\boldsymbol{x}_i\sigma_\star(\boldsymbol{x}_i^\top\boldsymbol{\beta}_\star) - c_{\star,1}\mathbf{v}^\top\boldsymbol{\beta}_\star) + n^{-1}\mathbf{v}^\top\boldsymbol{\varepsilon}.$$

Now $n^{-1}\mathbf{v}^\top\boldsymbol{\varepsilon} \sim \mathsf{N}(0, \sigma_\varepsilon^2\|\mathbf{v}\|_2^2)/n \to_P 0$. Moreover, by Condition 2.4, we can write $\sigma_\star(\boldsymbol{x}_i^\top\boldsymbol{\beta}_\star) = c_{\star,0} + c_{\star,1}\boldsymbol{x}_i^\top\boldsymbol{\beta}_\star + (P_{>1}\sigma_\star)(\boldsymbol{x}_i^\top\boldsymbol{\beta}_\star)$, where conditional on $\boldsymbol{\beta}_\star$, $(P_{>1}\sigma_\star)(\boldsymbol{x}_i^\top\boldsymbol{\beta}_\star)$ is orthogonal in $L^2$ to the constant function and to $\boldsymbol{x}_i^\top\boldsymbol{\beta}_\star$. Hence the first sum above equals

$$n^{-1}c_{\star,0}\mathbf{v}^\top\sum_{i=1}^{n}\boldsymbol{x}_i + n^{-1}c_{\star,1}\mathbf{v}^\top\left(\sum_{i=1}^{n}\boldsymbol{x}_i\boldsymbol{x}_i^\top - \mathbf{I}\right)\boldsymbol{\beta}_\star + n^{-1}\sum_{i=1}^{n}\mathbf{v}^\top\boldsymbol{x}_i(P_{>1}\sigma_\star)(\boldsymbol{x}_i^\top\boldsymbol{\beta}_\star).$$

For the first term, $n^{-1}c_{\star,0}\mathbf{v}^\top\sum_{i=1}^{n}\boldsymbol{x}_i \sim n^{-1}c_{\star,0} \cdot \mathsf{N}(0, n\|\mathbf{v}\|_2^2) \to_P 0$. The second term is $c_{\star,1}$ times a sample mean of i.i.d. random variables of the form $\mathbf{v}^\top(\boldsymbol{x}_i\boldsymbol{x}_i^\top - 1)\boldsymbol{\beta}_\star$, which have zero mean by the Gaussianity of $\boldsymbol{x}_i$, and for which all moments are finite. Hence, by the weak law of large numbers, this term converges to zero in probability.

Similarly, the third term is a sample mean of i.i.d. random variables of the form $\mathbf{v}^\top \boldsymbol{x}_i (P_{>1}\sigma_\star)(\boldsymbol{x}_i^\top \boldsymbol{\beta}_\star)$, which have zero mean by the Gaussianity of $\boldsymbol{x}_i$ and Lemma C.1, and whose second moments are finite since $\sigma_\star$ is Lipschitz. Hence, by the weak law of large numbers, this term also converges to zero in probability. This finishes the proof of the first claim.

Next, the second statement follows from (Ba et al., 2022, Lemma 18). While that work has slightly different assumptions on the teacher function $f_\star$, it is straightforward to check that their proof goes through unchanged under our assumptions. Specifically, their proof requires that $\boldsymbol{x} \mapsto f_\star(\boldsymbol{x}) = \sigma_\star(\boldsymbol{x}^\top \boldsymbol{\beta}_\star)$ is $O(1)$-Lipschitz, which holds in our case because $\sigma_\star$ is $O(1)$-Lipschitz, and $\|\boldsymbol{\beta}_\star\|_2 = O_\mathbb{P}(1)$.

### N.3 PROOF OF LEMMA K.2

By plugging in $\hat{\boldsymbol{a}}$ into the training loss, we find

$$
\begin{aligned}
\mathcal{L}_{\mathrm{tr}}(\mathbf{F}) &= \frac{1}{n}\|\tilde{\boldsymbol{y}} - \mathbf{F}\hat{\boldsymbol{a}}\|_2^2 + \lambda\|\hat{\boldsymbol{a}}\|_2^2 = \frac{1}{n}\|\tilde{\boldsymbol{y}}\|_2^2 - \frac{2}{n}\tilde{\boldsymbol{y}}^\top \mathbf{F}\hat{\boldsymbol{a}} + \frac{1}{n}\hat{\boldsymbol{a}}^\top (\mathbf{F}^\top \mathbf{F} + \lambda n \mathbf{I}_N)\hat{\boldsymbol{a}} \\
&= \frac{1}{n}\|\tilde{\boldsymbol{y}}\|_2^2 - \frac{1}{n}\tilde{\boldsymbol{y}}^\top \mathbf{F}\hat{\boldsymbol{a}} = \frac{1}{n}\|\tilde{\boldsymbol{y}}\|_2^2 - \frac{1}{n}\tilde{\boldsymbol{y}}^\top \mathbf{F}(\mathbf{F}^\top \mathbf{F} + \lambda n \mathbf{I}_N)^{-1}\mathbf{F}^\top \tilde{\boldsymbol{y}} \\
&= \frac{1}{n}\|\tilde{\boldsymbol{y}}\|_2^2 - \frac{1}{n}\tilde{\boldsymbol{y}}^\top \mathbf{F}\mathbf{F}^\top (\mathbf{F}\mathbf{F}^\top + \lambda n \mathbf{I}_n)^{-1}\tilde{\boldsymbol{y}} \\
&= \frac{1}{n}\|\tilde{\boldsymbol{y}}\|_2^2 - \frac{1}{n}\tilde{\boldsymbol{y}}^\top (\mathbf{F}\mathbf{F}^\top + \lambda n \mathbf{I}_n)(\mathbf{F}\mathbf{F}^\top + \lambda n \mathbf{I}_n)^{-1}\tilde{\boldsymbol{y}} + \lambda \tilde{\boldsymbol{y}}^\top (\mathbf{F}\mathbf{F}^\top + \lambda n \mathbf{I}_n)^{-1}\tilde{\boldsymbol{y}} \\
&= \lambda \tilde{\boldsymbol{y}}^\top (\mathbf{F}\mathbf{F}^\top + \lambda n \mathbf{I}_n)^{-1}\tilde{\boldsymbol{y}},
\end{aligned}
$$

which proves the lemma.

### N.4 PROOF OF LEMMA K.3

To prove the concentration of this term around its mean, we will use the generalized Hanson-Wright inequality (Sambale, 2023, Theorem 2.1) for $\alpha$-subexponential random variables. Note that, by definition, if $Z$ is a Gaussian random variable, $H_p(Z)$ is $2/p$-subexponential (see the definition in equation (1.1) of Sambale (2023)) and for these variables the Orlicz norm of order $2/p$ is bounded (see equation (1.3) of Sambale (2023)). Also note that $\|\mathbf{D}\|_{\mathrm{Fr}} \leq \sqrt{n}\|\mathbf{D}\|_{\mathrm{op}} = O_\mathbb{P}(1/\sqrt{n})$. Thus, using (Sambale, 2023, Theorem 2.1) and setting $t = \frac{\log(n)}{\sqrt{n}}$, we find

$$
\mathbb{P}\left(\left|g(\mathbf{Z})^\top \mathbf{D}\, g(\mathbf{Z}) - \mathbb{E}[g(\mathbf{Z})^\top \mathbf{D}\, g(\mathbf{Z})]\right| \geq \frac{\log n}{\sqrt{n}}\right) \leq 2\exp\left(-C\min\left\{\log^2(n), (\sqrt{n}\log n)^{1/p}\right\}\right),
$$

where $C > 0$ is some constant. This concludes the proof.

### N.5 PROOF OF LEMMA K.4

First, we show that switching from $\boldsymbol{w}_{0,i} \overset{i.i.d.}{\sim} \mathrm{Unif}(\mathbb{S}^{d-1})$ to $\hat{\boldsymbol{w}}_{0,i} \overset{i.i.d.}{\sim} \mathsf{N}(0, \frac{1}{d}\mathbf{I}_d)$ will not change the limit of the terms $\frac{1}{d}\mathbb{E}\mathrm{tr}(\tilde{\mathbf{X}}^\top \bar{\mathbf{R}}_0 \tilde{\mathbf{X}})$ and $\mathbb{E}\mathrm{tr}(\bar{\mathbf{R}}_0)$ which will appear later in the proof. First, we define $\hat{\mathbf{W}}_0 = [\hat{\boldsymbol{w}}_{0,1}, \cdots, \hat{\boldsymbol{w}}_{0,N}]^\top$,

$$
\mathbf{D} = \mathrm{diag}\left(\frac{1}{\|\hat{\boldsymbol{w}}_{0,1}\|_2}, \cdots, \frac{1}{\|\hat{\boldsymbol{w}}_{0,N}\|_2}\right), \ \mathbf{W}_0 =^d \mathbf{D}\hat{\mathbf{W}}_0, \ \hat{\mathbf{F}}_0 = \sigma(\tilde{\mathbf{X}}\hat{\mathbf{W}}_0^\top),
$$

$$
\text{and } \hat{\bar{\mathbf{R}}}_0 = (\hat{\mathbf{F}}_0\hat{\mathbf{F}}_0^\top + \lambda n \mathbf{I}_n)^{-1}.
$$

Then,

$$
\begin{aligned}
\left|\mathrm{tr}\left[\bar{\mathbf{R}}_0 - \hat{\bar{\mathbf{R}}}_0\right]\right| &= \left|\mathrm{tr}\left[(\mathbf{F}_0\mathbf{F}_0^\top + \lambda n \mathbf{I}_n)^{-1} - (\hat{\mathbf{F}}_0\hat{\mathbf{F}}_0^\top + \lambda n \mathbf{I}_n)^{-1}\right]\right| \\
&= \left|\mathrm{tr}\left[(\mathbf{F}_0\mathbf{F}_0^\top + \lambda n \mathbf{I}_n)^{-1}(\mathbf{F}_0\mathbf{F}_0^\top - \hat{\mathbf{F}}_0\hat{\mathbf{F}}_0^\top)(\hat{\mathbf{F}}_0\hat{\mathbf{F}}_0^\top + \lambda n \mathbf{I}_n)^{-1}\right]\right| \\
&\leq \mathrm{tr}(\mathbf{F}_0\mathbf{F}_0^\top + \lambda n \mathbf{I}_n)^{-1}\|(\hat{\mathbf{F}}_0\hat{\mathbf{F}}_0^\top + \lambda n \mathbf{I}_n)^{-1}\|_{\mathrm{op}}\|\mathbf{F}_0\mathbf{F}_0 - \hat{\mathbf{F}}_0\hat{\mathbf{F}}_0\|_{\mathrm{op}} \\
&\leq \frac{C}{n}\|\mathbf{F}_0\mathbf{F}_0 - \hat{\mathbf{F}}_0\hat{\mathbf{F}}_0\|_{\mathrm{op}}.
\end{aligned}
$$

Now, using Conjecture 4.3, we can replace $\mathbf{F}_0$ and $\hat{\mathbf{F}}_0$ with $\mathbf{F}_0 = c_1\tilde{\mathbf{X}}\mathbf{W}_0^\top + c_{>1}\mathbf{Z}$ and $\hat{\mathbf{F}}_0 = c_1\tilde{\mathbf{X}}\hat{\mathbf{W}}_0^\top + c_{>1}\mathbf{Z}$, respectively, without changing the limit. With this, we have

$$\mathbf{F}_0\mathbf{F}_0^\top - \hat{\mathbf{F}}_0\hat{\mathbf{F}}_0^\top$$
$$= c_1^2\tilde{\mathbf{X}}(\mathbf{W}_0\mathbf{W}_0^\top - \hat{\mathbf{W}}_0\hat{\mathbf{W}}_0^\top)\tilde{\mathbf{X}}^\top + c_1c_{>1}\tilde{\mathbf{X}}(\mathbf{W}_0 - \hat{\mathbf{W}}_0)^\top\mathbf{Z}^\top + c_1c_{>1}\mathbf{Z}(\mathbf{W}_0 - \hat{\mathbf{W}}_0)\tilde{\mathbf{X}}^\top.$$

Now,

$$\|\mathbf{W}_0\mathbf{W}_0^\top - \hat{\mathbf{W}}_0\hat{\mathbf{W}}_0^\top\|_{\mathrm{op}} \le \|\mathbf{I}_N - \mathbf{D}\|_{\mathrm{op}}\|\mathbf{W}_0\mathbf{W}_0^\top\|_{\mathrm{op}}(\|\mathbf{D}\|_{\mathrm{op}} + 1).$$

Note that $\|\mathbf{W}_0\mathbf{W}_0^\top\|_{\mathrm{op}} = O_\mathbb{P}(1)$, $\|\mathbf{D}\|_{\mathrm{op}} = O_\mathbb{P}(1)$, and $\|\mathbf{I}_N - \mathbf{D}\|_{\mathrm{op}} = o_\mathbb{P}(1)$. Thus $\|\mathbf{W}_0\mathbf{W}_0^\top - \hat{\mathbf{W}}_0\hat{\mathbf{W}}_0^\top\|_{\mathrm{op}} = o_\mathbb{P}(1)$. Also, similarly, $\|\mathbf{W}_0 - \hat{\mathbf{W}}_0\|_{\mathrm{op}} = o_\mathbb{P}(1)$. Hence, noting that $\|\tilde{\mathbf{X}}\|_{\mathrm{op}}$ and $\|\mathbf{Z}\|_{\mathrm{op}}$ are both $O_\mathbb{P}(\sqrt{N})$, we have $\frac{1}{n}\|\mathbf{F}_0\mathbf{F}_0^\top - \hat{\mathbf{F}}_0\hat{\mathbf{F}}_0^\top\|_{\mathrm{op}} \to_P 0$. This implies that $|\mathrm{tr}[\bar{\mathbf{R}}_0 - \hat{\bar{\mathbf{R}}}_0]| = o_\mathbb{P}(1)$. Also,

$$\left|\frac{1}{d}\mathrm{tr}\left[\tilde{\mathbf{X}}^\top\bar{\mathbf{R}}_0\tilde{\mathbf{X}}\right] - \frac{1}{d}\mathrm{tr}\left[\tilde{\mathbf{X}}^\top\hat{\bar{\mathbf{R}}}_0\tilde{\mathbf{X}}\right]\right| \le |\mathrm{tr}[\bar{\mathbf{R}}_0 - \hat{\bar{\mathbf{R}}}_0]|\frac{\|\tilde{\mathbf{X}}\tilde{\mathbf{X}}^\top\|_{\mathrm{op}}}{d} \to_P 0.$$

Finally, we can prove the required claims as follows:

(a) Since $\boldsymbol{\beta}_\star \sim \mathsf{N}(0, \frac{1}{d}\mathbf{I}_d)$, we have

$$\boldsymbol{\beta}_\star^\top\tilde{\mathbf{X}}^\top\bar{\mathbf{R}}_0\tilde{\mathbf{X}}\boldsymbol{\beta}_\star = \frac{1}{d}\mathbb{E}\mathrm{tr}(\tilde{\mathbf{X}}^\top\bar{\mathbf{R}}_0\tilde{\mathbf{X}}) + o_\mathbb{P}(1),$$

by the Hanson-Wright inequality. Note that by the argument above, we can assume that $\hat{\boldsymbol{w}}_{0,i} \overset{i.i.d.}{\sim} \mathsf{N}(0, \frac{1}{d}\mathbf{I}_d)$ without changing the limiting trace. Further, from (Adlam & Pennington, 2020a, Proposition 1), see also Adlam et al. (2022), we have $\frac{1}{d}\mathbb{E}\mathrm{tr}(\tilde{\mathbf{X}}^\top\bar{\mathbf{R}}_0\tilde{\mathbf{X}}) \to \frac{\psi}{\phi}m_2$; see the discussion at the end of this proof for the detailed explanation. Now, we arrive at the conclusion by applying Lemma K.1.

(b) Since $\boldsymbol{a} \sim \mathsf{N}(0, \frac{1}{N}\mathbf{I}_N)$, we have

$$\boldsymbol{a}^\top\mathbf{F}_0^\top\bar{\mathbf{R}}_0\mathbf{F}_0\boldsymbol{a} - \|\boldsymbol{a}\|_2^2 = \frac{1}{N}\mathrm{tr}\left(\mathbf{F}_0^\top\bar{\mathbf{R}}_0\mathbf{F}_0\right) - 1 + o_\mathbb{P}(1)$$

by the Hanson-Wright inequality. Moreover,

$$\mathbf{F}_0^\top\bar{\mathbf{R}}_0\mathbf{F}_0 = \mathbf{F}_0^\top\mathbf{F}_0(\mathbf{F}_0^\top\mathbf{F}_0 + \lambda n\mathbf{I}_N)^{-1}$$
$$= (\mathbf{F}_0^\top\mathbf{F}_0 + \lambda n\mathbf{I}_N - \lambda n\mathbf{I}_N)(\mathbf{F}_0^\top\mathbf{F}_0 + \lambda n\mathbf{I}_N)^{-1} = \mathbf{I}_N - \lambda n(\mathbf{F}_0^\top\mathbf{F}_0 + \lambda n\mathbf{I}_N)^{-1}.$$

Hence, $\frac{1}{N}\mathrm{tr}\left(\mathbf{F}_0^\top\bar{\mathbf{R}}_0\mathbf{F}_0\right) - 1 = -\frac{\lambda n}{N}\mathrm{tr}(\mathbf{F}_0^\top\mathbf{F}_0 + \lambda n\mathbf{I}_N)^{-1}$. From the argument above, we can assume that $\hat{\boldsymbol{w}}_{0,i} \overset{i.i.d.}{\sim} \mathsf{N}(0, \frac{1}{d}\mathbf{I}_d)$ without changing the limiting trace. It follows from (Adlam & Pennington, 2020a, Proposition 1) that $\mathbb{E}\mathrm{tr}\,\bar{\mathbf{R}}_0 \to \frac{\psi}{\phi}m_1$; again see the discussion at the end of this proof for the detailed explanation. Note that $\lim\mathbb{E}\mathrm{tr}\,\bar{\mathbf{R}}_0$ is the limiting Stieltjes transform of $\mathbf{F}_0\mathbf{F}_0^\top$. Hence, $\bar{m}_1 = \lim\mathbb{E}\mathrm{tr}(\mathbf{F}_0^\top\mathbf{F}_0 + \lambda n\mathbf{I}_N)^{-1}$ is the limiting companion Stieltjes transform of $m_1$ which is given by

$$\bar{m}_1 = \frac{\psi}{\phi}m_1 - \left(1 - \frac{\phi}{\psi}\right)\frac{1}{\lambda}. \tag{34}$$

This concludes the proof.

For the reader's convenience, we provide the following diagram that shows how the notations of Adlam & Pennington (2020a) (left) match ($\Leftrightarrow$) ours (right):

$$n_0 \Leftrightarrow d, \qquad n_1 \Leftrightarrow N, \qquad m \Leftrightarrow n, \qquad \phi, \psi \Leftrightarrow \phi, \psi,$$

$$\mathbf{X}^\top \in \mathbb{R}^{m \times n_0} \Leftrightarrow \tilde{\mathbf{X}} \in \mathbb{R}^{n \times d}, \qquad \mathbf{F}^\top \in \mathbb{R}^{m \times n_1} \Leftrightarrow \mathbf{F}_0 \in \mathbb{R}^{n \times N}, \qquad \sigma_{W_2} = 0,$$

$$\frac{1}{n_1}\mathbf{K}(\lambda m/n_1)^{-1} = \frac{1}{n_1}\mathbf{F}^\top\mathbf{F} + \lambda\mathbf{I}_m \Leftrightarrow \bar{\mathbf{R}}_0^{-1} = \mathbf{F}_0\mathbf{F}_0^\top + \lambda n\mathbf{I}_n, \quad \zeta \Leftrightarrow c_1^2, \qquad \eta \Leftrightarrow c_1^2 + c_{>1}^2,$$

$$\tau_1 = \frac{1}{m}\mathbb{E}\mathrm{tr}\,\mathbf{K}^{-1} \Leftrightarrow m_1 = \frac{N}{n}\mathbb{E}\mathrm{tr}\,\bar{\mathbf{R}}_0, \quad \tau_2 = \frac{1}{mn_0}\mathbb{E}\mathrm{tr}\,\mathbf{X}^\top\mathbf{X}\mathbf{K}^{-1} \Leftrightarrow m_2 = \frac{N}{nd}\mathbb{E}\mathrm{tr}\,\tilde{\mathbf{X}}\tilde{\mathbf{X}}^\top\bar{\mathbf{R}}_0.$$

## N.6  PROOF OF LEMMA K.5

Define $\hat{\mathbf{X}} = \tilde{\mathbf{X}} - \tilde{\mathbf{X}}\boldsymbol{u}\boldsymbol{u}^\top$, which implies $\hat{\mathbf{X}} \perp\!\!\!\perp \tilde{\mathbf{X}}\boldsymbol{u}$ due to the Gaussianity of $\mathbf{X}$. Based on Conjecture 4.3, we can replace $\mathbf{F}_0$ with $c_1\tilde{\mathbf{X}}\mathbf{W}_0^\top + c_{>1}\mathbf{Z}$, where $\mathbf{Z} \in \mathbb{R}^{n\times d}$ is an independent random matrix with $\mathsf{N}(0, 1)$ entries, without changing the conclusion. Hence, from now on, we write $\mathbf{F}_0 = c_1\tilde{\mathbf{X}}\mathbf{W}_0^\top + c_{>1}\mathbf{Z}$. Further, we define

$$\hat{\mathbf{F}}_0 = c_1\hat{\mathbf{X}}\mathbf{W}_0^\top + c_{>1}\mathbf{Z}. \tag{35}$$

Thus, by the definition of $\hat{\mathbf{X}}$, $\hat{\mathbf{F}}_0 = \mathbf{F}_0 - c_1\tilde{\mathbf{X}}\boldsymbol{u}(\mathbf{W}_0\boldsymbol{u})^\top$. As a consequence, we also have $\mathbf{F}_0\mathbf{F}_0^\top = \hat{\mathbf{F}}_0\hat{\mathbf{F}}_0^\top + \mathbf{V}\mathbf{D}\mathbf{V}^\top$, where $\mathbf{V} = \begin{bmatrix} \hat{\mathbf{F}}_0\mathbf{W}_0\boldsymbol{u} & \tilde{\mathbf{X}}\boldsymbol{u} \end{bmatrix} \in \mathbb{R}^{n\times 2}$ and

$$\mathbf{D} = \begin{bmatrix} 0 & c_1 \\ c_1 & c_1^2\|\mathbf{W}_0\boldsymbol{u}\|_2^2 \end{bmatrix}.$$

Noting that $D$ is invertible, and using the Woodbury formula, with $\hat{\mathbf{R}}_0 = (\hat{\mathbf{F}}_0\hat{\mathbf{F}}_0^\top + \lambda n\mathbf{I}_n)^{-1}$, we find

$$\bar{\mathbf{R}}_0 = \hat{\mathbf{R}}_0 - \hat{\mathbf{R}}_0\mathbf{V}(\mathbf{D}^{-1} + \mathbf{V}^\top\hat{\mathbf{R}}_0\mathbf{V})^{-1}\mathbf{V}^\top\hat{\mathbf{R}}_0. \tag{36}$$

Now, we can write

$$H_q(\tilde{\mathbf{X}}\boldsymbol{u})^\top\bar{\mathbf{R}}_0 H_p(\tilde{\mathbf{X}}\boldsymbol{u})$$
$$= H_q(\tilde{\mathbf{X}}\boldsymbol{u})^\top\hat{\mathbf{R}}_0 H_p(\tilde{\mathbf{X}}\boldsymbol{u}) - H_q(\tilde{\mathbf{X}}\boldsymbol{u})^\top\hat{\mathbf{R}}_0\mathbf{V}(\mathbf{D}^{-1} + \mathbf{V}^\top\hat{\mathbf{R}}_0\mathbf{V})^{-1}\mathbf{V}^\top\hat{\mathbf{R}}_0 H_p(\tilde{\mathbf{X}}\boldsymbol{u}).$$

Next, we can analyze each term in the above sum separately.

The first term on the right hand side converges to zero by using Lemma K.3 to prove the concentration of this term around its mean and noting that the mean is zero using the orthogonality property of Hermite polynomials (Lemma C.1).

To analyze the second term, we first study the matrix $\mathbf{K} = (\mathbf{D}^{-1} + \mathbf{V}^\top\hat{\mathbf{R}}_0\mathbf{V})^{-1}$, writing

$$\mathbf{K}^{-1} = (\mathbf{D}^{-1} + \mathbf{V}^\top\hat{\mathbf{R}}_0\mathbf{V}) = \begin{bmatrix} \boldsymbol{u}^\top\mathbf{W}_0^\top\hat{\mathbf{F}}_0^\top\hat{\mathbf{R}}_0\hat{\mathbf{F}}_0\mathbf{W}_0\boldsymbol{u} - \|\mathbf{W}_0\boldsymbol{u}\|_2^2 & \boldsymbol{u}^\top\mathbf{W}_0^\top\hat{\mathbf{F}}_0^\top\hat{\mathbf{R}}_0\tilde{\mathbf{X}}\boldsymbol{u} - \frac{1}{c_1} \\ \boldsymbol{u}^\top\tilde{\mathbf{X}}^\top\hat{\mathbf{R}}_0\hat{\mathbf{F}}_0\mathbf{W}_0\boldsymbol{u} - \frac{1}{c_1} & \boldsymbol{u}^\top\tilde{\mathbf{X}}^\top\hat{\mathbf{R}}_0\tilde{\mathbf{X}}\boldsymbol{u} \end{bmatrix}.$$

It can readily verified that all elements in this matrix are $O_\mathbb{P}(1)$ by checking the order of the operator and Euclidean norms. Next, we analyze the terms in the expression

$$H_q(\tilde{\mathbf{X}}\boldsymbol{u})^\top\hat{\mathbf{R}}_0\mathbf{V}\mathbf{K}\mathbf{V}^\top\hat{\mathbf{R}}_0 H_p(\tilde{\mathbf{X}}\boldsymbol{u}) = [\mathbf{K}]_{1,1} H_q(\tilde{\mathbf{X}}\boldsymbol{u})^\top\hat{\mathbf{R}}_0(\hat{\mathbf{F}}_0\mathbf{W}_0\boldsymbol{u})(\hat{\mathbf{F}}_0\mathbf{W}_0\boldsymbol{u})^\top\hat{\mathbf{R}}_0 H_p(\tilde{\mathbf{X}}\boldsymbol{u})$$
$$+ [\mathbf{K}]_{1,2} H_q(\tilde{\mathbf{X}}\boldsymbol{u})^\top\hat{\mathbf{R}}_0(\hat{\mathbf{F}}_0\mathbf{W}_0\boldsymbol{u})(\tilde{\mathbf{X}}\boldsymbol{u})^\top\hat{\mathbf{R}}_0 H_p(\tilde{\mathbf{X}}\boldsymbol{u})$$
$$+ [\mathbf{K}]_{2,1} H_q(\tilde{\mathbf{X}}\boldsymbol{u})^\top\hat{\mathbf{R}}_0(\tilde{\mathbf{X}}\boldsymbol{u})(\hat{\mathbf{F}}_0\mathbf{W}_0\boldsymbol{u})^\top\hat{\mathbf{R}}_0 H_p(\tilde{\mathbf{X}}\boldsymbol{u})$$
$$+ [\mathbf{K}]_{2,2} H_q(\tilde{\mathbf{X}}\boldsymbol{u})^\top\hat{\mathbf{R}}_0(\tilde{\mathbf{X}}\boldsymbol{u})(\tilde{\mathbf{X}}\boldsymbol{u})^\top\hat{\mathbf{R}}_0 H_p(\tilde{\mathbf{X}}\boldsymbol{u}).$$

Without loss of generality, we can assume that $p \neq 1$.

- **First Term.** Note that $H_q(\tilde{\mathbf{X}}\boldsymbol{u})^\top$ and $H_p(\tilde{\mathbf{X}}\boldsymbol{u})$ are orthogonal in $L^2$ by the properties of the Hermite polynomials, and conditional on $\boldsymbol{u}$, they are independent of $\hat{\mathbf{R}}_0(\hat{\mathbf{F}}_0\mathbf{W}_0\boldsymbol{u})(\hat{\mathbf{F}}_0\mathbf{W}_0\boldsymbol{u})^\top\hat{\mathbf{R}}_0$. Moreover,

  $$\|\hat{\mathbf{R}}_0(\hat{\mathbf{F}}_0\mathbf{W}_0\boldsymbol{u})(\hat{\mathbf{F}}_0\mathbf{W}_0\boldsymbol{u})^\top\hat{\mathbf{R}}_0\|_{\mathrm{op}} = O_\mathbb{P}(1/n).$$

  Thus, by using Lemma K.3, this term converges to zero.

- **Second Term.** Similar to the argument above, we can show that $(\tilde{\mathbf{X}}\boldsymbol{u})^\top\hat{\mathbf{R}}_0 H_p(\tilde{\mathbf{X}}\boldsymbol{u})$ converges to zero. Also, by analyzing the operator norms, we have $H_q(\tilde{\mathbf{X}}\boldsymbol{u})^\top\hat{\mathbf{R}}_0(\hat{\mathbf{F}}_0\mathbf{W}_0\boldsymbol{u}) = O(1)$. This implies that the second term converges to zero.

- **Third Term.** First, note that by a simple order-wise analysis, $H_q(\tilde{\mathbf{X}}\boldsymbol{u})^\top\hat{\mathbf{R}}_0(\tilde{\mathbf{X}}\boldsymbol{u}) = O_\mathbb{P}(1)$. Now, we have $H_p(\tilde{\mathbf{X}}\boldsymbol{u})$ is independent of $(\hat{\mathbf{F}}_0\mathbf{W}_0\boldsymbol{u})^\top\hat{\mathbf{R}}_0$ and $\|(\hat{\mathbf{F}}_0\mathbf{W}_0\boldsymbol{u})^\top\hat{\mathbf{R}}_0\|_2 = O_\mathbb{P}(1/\sqrt{n})$. The term $(\hat{\mathbf{F}}_0\mathbf{W}_0\boldsymbol{u})^\top\hat{\mathbf{R}}_0 H_p(\tilde{\mathbf{X}}\boldsymbol{u})$ converges to zero in probability by noting that $H_p(\tilde{\mathbf{X}}\boldsymbol{u})$ is mean zero for $p \neq 0$. For the $p = 0$ case, we can use an orthogonality invariance argument identical to the one used to analyze equation 27.

- **Fourth Term.** This term also converges to zero because $(\tilde{\mathbf{X}}\boldsymbol{u})^\top \hat{\mathbf{R}}_0 H_p(\tilde{\mathbf{X}}\boldsymbol{u})$ converges to zero, as argued above.

Putting everything together, the proof is completed.

### N.7  PROOF OF LEMMA K.6

We will prove part (a) first. To do this, we will first handle the cases where $p = 0$ and $p = 1$.

For $p = 0$, we have

$$\sqrt{N} H_0(\tilde{\boldsymbol{\theta}}_\star) \bar{\mathbf{R}}_0 \mathbf{F}_0 \boldsymbol{a}^{\circ 2} = \sqrt{N} \mathbf{1}_n^\top \bar{\mathbf{R}}_0 \mathbf{F}_0 \boldsymbol{a}^{\circ 2}.$$

This is identical to the second term in equation 27 and it is shown to be $o_{\mathbb{P}}(1)$

For $p = 1$, we need to analyze

$$\sqrt{N} H_1(\tilde{\boldsymbol{\theta}}_\star) \bar{\mathbf{R}}_0 \mathbf{F}_0 \boldsymbol{a}^{\circ 2} = \sqrt{N} \boldsymbol{\beta}_\star^\top \tilde{\mathbf{X}}^\top \bar{\mathbf{R}}_0 \mathbf{F}_0 \boldsymbol{a}^{\circ 2}.$$

Note that $\boldsymbol{\beta}_\star \sim \mathsf{N}(0, \frac{1}{d}\mathbf{I}_d)$ is independent of $\sqrt{N}\tilde{\mathbf{X}}^\top \bar{\mathbf{R}}_0 \mathbf{F}_0 \boldsymbol{a}^{\circ 2}$ and

$$\|\sqrt{N}\tilde{\mathbf{X}}^\top \bar{\mathbf{R}}_0 \mathbf{F}_0 \boldsymbol{a}^{\circ 2}\|_2 \leq \sqrt{N}\|\tilde{\mathbf{X}}\|_{\mathrm{op}} \cdot \|\bar{\mathbf{R}}_0\|_{\mathrm{op}} \cdot \|\mathbf{F}_0\|_{\mathrm{op}} \cdot \|\boldsymbol{a}^{\circ 2}\|_2 = O_{\mathbb{P}}(1).$$

Thus, we can conclude that $\sqrt{N} H_1(\tilde{\boldsymbol{\theta}}_\star) \bar{\mathbf{R}}_0 \mathbf{F}_0 \boldsymbol{a}^{\circ 2} \to 0$ in probability.

To analyze the case where $p > 1$, we first define $\hat{\mathbf{X}} = \tilde{\mathbf{X}} - \tilde{\boldsymbol{\theta}}_\star \boldsymbol{\beta}_\star^\top$. By construction, we have $\hat{\mathbf{X}} \perp\!\!\!\perp \tilde{\boldsymbol{\theta}}_\star$. As in the proof of Lemma K.5, Based on Conjecture 4.3, we can replace $\mathbf{F}_0$ with $c_1 \tilde{\mathbf{X}} \mathbf{W}_0^\top + c_{>1}\mathbf{Z}$ in our computations without changing the limiting result, where $\mathbf{Z} \in \mathbb{R}^{n \times d}$ is an independent random matrix with $\mathsf{N}(0,1)$ entries. Thus, from now on, we denote $\mathbf{F}_0 = c_1 \tilde{\mathbf{X}} \mathbf{W}_0^\top + c_{>1}\mathbf{Z}$. We define $\hat{\mathbf{F}}_0$ as in equation 35. Thus, $\hat{\mathbf{F}}_0 = \mathbf{F}_0 - c_1 \tilde{\boldsymbol{\theta}}_\star (\mathbf{W}_0 \boldsymbol{\beta}_\star)^\top$. As a consequence, we can write $\mathbf{F}_0 \mathbf{F}_0^\top = \hat{\mathbf{F}}_0 \hat{\mathbf{F}}_0^\top + \mathbf{V}\mathbf{D}\mathbf{V}^\top$, where $\mathbf{V} = \begin{bmatrix} \hat{\mathbf{F}}_0 \mathbf{W}_0 \boldsymbol{\beta}_\star & \tilde{\boldsymbol{\theta}}_\star \end{bmatrix} \in \mathbb{R}^{n \times 2}$ and

$$\mathbf{D} = \begin{bmatrix} 0 & c_1 \\ c_1 & c_1^2 \|\mathbf{W}_0 \boldsymbol{\beta}_\star\|_2^2 \end{bmatrix}.$$

Using the Woodbury formula, we find that equation 36 still holds. Now, we can write

$$\sqrt{N} H_p(\tilde{\boldsymbol{\theta}}_\star)^\top \bar{\mathbf{R}}_0 \mathbf{F}_0 \boldsymbol{a}^{\circ 2} \tag{37}$$
$$= \sqrt{N} H_p(\tilde{\boldsymbol{\theta}}_\star)^\top \hat{\mathbf{R}}_0 \mathbf{F}_0 \boldsymbol{a}^{\circ 2} - \sqrt{N} H_p(\tilde{\boldsymbol{\theta}}_\star)^\top \hat{\mathbf{R}}_0 \mathbf{V}(\mathbf{D}^{-1} + \mathbf{V}^\top \hat{\mathbf{R}}_0 \mathbf{V})^{-1} \mathbf{V}^\top \hat{\mathbf{R}}_0 \mathbf{F}_0 \boldsymbol{a}^{\circ 2}$$
$$= \sqrt{N} H_p(\tilde{\boldsymbol{\theta}}_\star)^\top \hat{\mathbf{R}}_0 (\hat{\mathbf{F}}_0 + c_1 \tilde{\boldsymbol{\theta}}_\star (\mathbf{W}_0 \boldsymbol{\beta}_\star)^\top) \boldsymbol{a}^{\circ 2}$$
$$\quad - \sqrt{N} H_p(\tilde{\boldsymbol{\theta}}_\star)^\top \hat{\mathbf{R}}_0 \mathbf{V}(\mathbf{D}^{-1} + \mathbf{V}^\top \hat{\mathbf{R}}_0 \mathbf{V})^{-1} \mathbf{V}^\top \hat{\mathbf{R}}_0 (\hat{\mathbf{F}}_0 + c_1 \tilde{\boldsymbol{\theta}}_\star (\mathbf{W}_0 \boldsymbol{\beta}_\star)^\top) \boldsymbol{a}^{\circ 2}.$$

Now, we can analyze each term in the above sum separately.

**Term 1.**  Note that by a simple orderwise analysis,

$$\|\sqrt{N} \hat{\mathbf{R}}_0 \hat{\mathbf{F}}_0 \boldsymbol{a}^{\circ 2}\|_{\mathrm{op}} \leq \sqrt{N}\|\hat{\mathbf{R}}_0\|_{\mathrm{op}}\|\hat{\mathbf{F}}_0\|_{\mathrm{op}}\|\boldsymbol{a}^{\circ 2}\|_2 = O(1/\sqrt{N}).$$

We have $\|H_p(\tilde{\boldsymbol{\theta}}_\star)\|_2 = O_{\mathbb{P}}(\sqrt{N})$, $\mathbb{E}[H_p(\tilde{\boldsymbol{\theta}}_\star)] = 0$, and $H_p(\tilde{\boldsymbol{\theta}}_\star)$ has independent entries. Also $H_p(\tilde{\boldsymbol{\theta}}_\star) \perp\!\!\!\perp \hat{\mathbf{R}}_0 \hat{\mathbf{F}}_0 \boldsymbol{a}^{\circ 2}$. Thus, $\sqrt{N} H_p(\tilde{\boldsymbol{\theta}}_\star)^\top \hat{\mathbf{R}}_0 \hat{\mathbf{F}}_0 \boldsymbol{a}^{\circ 2} \to_P 0$.

We now need to analyze $\sqrt{N} H_p(\tilde{\boldsymbol{\theta}}_\star)^\top \hat{\mathbf{R}}_0 \tilde{\boldsymbol{\theta}}_\star \boldsymbol{\beta}_\star^\top \mathbf{W}_0^\top \boldsymbol{a}^{\circ 2}$. Note that $H_p(\tilde{\boldsymbol{\theta}}_\star)^\top \hat{\mathbf{R}}_0 \tilde{\boldsymbol{\theta}}_\star = O_{\mathbb{P}}(1)$ by a simple order analysis of the norms. We also have $\sqrt{N}\boldsymbol{\beta}_\star^\top \mathbf{W}_0^\top \boldsymbol{a}^{\circ 2} \to_P 0$, because $\boldsymbol{\beta}_\star \sim \mathsf{N}(0, \frac{1}{d}\mathbf{I}_d)$ is independent of the norm bounded vector $\sqrt{N}\mathbf{W}_0^\top \boldsymbol{a}^{\circ 2}$.

**Term 2.**  To analyze the second term, we first study the matrix $\mathbf{K} = (\mathbf{D}^{-1} + \mathbf{V}^\top \hat{\mathbf{R}}_0 \mathbf{V})^{-1}$:

$$\mathbf{K}^{-1} = (\mathbf{D}^{-1} + \mathbf{V}^\top \hat{\mathbf{R}}_0 \mathbf{V}) = \begin{bmatrix} \boldsymbol{\beta}_\star^\top \mathbf{W}_0^\top \hat{\mathbf{F}}_0^\top \hat{\mathbf{R}}_0 \hat{\mathbf{F}}_0 \mathbf{W}_0 \boldsymbol{\beta}_\star - \|\mathbf{W}_0 \boldsymbol{\beta}_\star\|_2^2 & \boldsymbol{\beta}_\star^\top \mathbf{W}_0^\top \hat{\mathbf{F}}_0^\top \hat{\mathbf{R}}_0 \tilde{\boldsymbol{\theta}}_\star - \frac{1}{c_1} \\ \boldsymbol{\beta}_\star^\top \tilde{\mathbf{X}}^\top \hat{\mathbf{R}}_0 \hat{\mathbf{F}}_0 \mathbf{W}_0 \boldsymbol{\beta}_\star - \frac{1}{c_1} & \boldsymbol{\beta}_\star^\top \tilde{\mathbf{X}}^\top \hat{\mathbf{R}}_0 \tilde{\boldsymbol{\theta}}_\star \end{bmatrix}.$$

By orderwise analysis, all elements in this matrix converge to deterministic $O_{\mathbb{P}}(1)$ values in probability. We write the second term in equation 37 as follows:

$$\sqrt{N} H_p(\tilde{\boldsymbol{\theta}}_\star)^\top \hat{\mathbf{R}}_0 \mathbf{V} \mathbf{K} \mathbf{V}^\top \hat{\mathbf{R}}_0 \mathbf{F}_0 \boldsymbol{a}^{\circ 2}$$

$$= [\mathbf{K}]_{1,1} H_p(\tilde{\boldsymbol{\theta}}_\star)^\top \hat{\mathbf{R}}_0 (\hat{\mathbf{F}}_0 \mathbf{W}_0 \boldsymbol{\beta}_\star)(\hat{\mathbf{F}}_0 \mathbf{W}_0 \boldsymbol{\beta}_\star)^\top \hat{\mathbf{R}}_0 \mathbf{F}_0 (\sqrt{N} \boldsymbol{a}^{\circ 2})$$

$$+ [\mathbf{K}]_{1,2} H_p(\tilde{\boldsymbol{\theta}}_\star)^\top \hat{\mathbf{R}}_0 (\hat{\mathbf{F}}_0 \mathbf{W}_0 \boldsymbol{\beta}_\star) \tilde{\boldsymbol{\theta}}_\star^\top \hat{\mathbf{R}}_0 \mathbf{F}_0 (\sqrt{N} \boldsymbol{a}^{\circ 2})$$

$$+ [\mathbf{K}]_{2,1} H_p(\tilde{\boldsymbol{\theta}}_\star)^\top \hat{\mathbf{R}}_0 (\tilde{\boldsymbol{\theta}}_\star)(\hat{\mathbf{F}}_0 \mathbf{W}_0 \boldsymbol{\beta}_\star)^\top \hat{\mathbf{R}}_0 \mathbf{F}_0 (\sqrt{N} \boldsymbol{a}^{\circ 2})$$

$$+ [\mathbf{K}]_{2,2} H_p(\tilde{\boldsymbol{\theta}}_\star)^\top \hat{\mathbf{R}}_0 (\tilde{\boldsymbol{\theta}}_\star) \tilde{\boldsymbol{\theta}}_\star^\top \hat{\mathbf{R}}_0 \mathbf{F}_0 (\sqrt{N} \boldsymbol{a}^{\circ 2}).$$

In the sum above, we will show that each term converges to zero.

- First term: By orderwise analysis, we have $\|\hat{\mathbf{R}}_0 (\hat{\mathbf{F}}_0 \mathbf{W}_0 \boldsymbol{\beta}_\star)\|_{\mathrm{op}} = O_{\mathbb{P}}(1/\sqrt{N})$. Further, $H_p(\tilde{\boldsymbol{\theta}}_\star)$ is independent of it (only considering the randomness in $\tilde{\mathbf{X}}$) with mean zero and $\|H_p(\tilde{\boldsymbol{\theta}}_\star)\|_2 = O_{\mathbb{P}}(\sqrt{N})$. This implies that

$$H_p(\tilde{\boldsymbol{\theta}}_\star)^\top \hat{\mathbf{R}}_0 (\hat{\mathbf{F}}_0 \mathbf{W}_0 \boldsymbol{\beta}_\star) \to_P 0. \tag{38}$$

  We can use a simple order argument to show that $\sqrt{N} (\hat{\mathbf{F}}_0 \mathbf{W}_0 \boldsymbol{\beta}_\star)^\top \hat{\mathbf{R}}_0 \mathbf{F}_0 \boldsymbol{a}^{\circ 2} = O_{\mathbb{P}}(1)$. Thus, the first term converges to zero.

- Second term: For this term, we use the fact that $H_p(\tilde{\boldsymbol{\theta}}_\star)^\top \hat{\mathbf{R}}_0 (\hat{\mathbf{F}}_0 \mathbf{W}_0 \boldsymbol{\beta}_\star) \to_P 0$. We can also use an orderwise analysis to prove that $\sqrt{N} (\tilde{\boldsymbol{\theta}}_\star)^\top \hat{\mathbf{R}}_0 \mathbf{F}_0 \boldsymbol{a}^{\circ 2} = O_{\mathbb{P}}(1)$. This proves that the second term also converges to zero.

- Third term: By a simple orderwise analysis, we have $\sqrt{N} (\hat{\mathbf{F}}_0 \mathbf{W}_0 \boldsymbol{\beta}_\star)^\top \hat{\mathbf{R}}_0 \mathbf{F}_0 \boldsymbol{a}^{\circ 2} = O_{\mathbb{P}}(1)$. To show that the third term converges to zero, it is enough to show that $H_p(\tilde{\boldsymbol{\theta}}_\star)^\top \hat{\mathbf{R}}_0 (\tilde{\boldsymbol{\theta}}_\star) \to_P 0$, which is true for $p \neq 1$ by using Lemma K.3 and the orthogonality property of Hermite polynomials (Lemma C.1).

- Fourth term: By a simple orderwise analysis, we have $\sqrt{N} \tilde{\boldsymbol{\theta}}_\star^\top \hat{\mathbf{R}}_0 \mathbf{F}_0 \boldsymbol{a}^{\circ 2} = O_{\mathbb{P}}(1)$. Again, to show that the fourth term converges to zero, it is enough to show that $H_p(\tilde{\boldsymbol{\theta}}_\star)^\top \hat{\mathbf{R}}_0 (\tilde{\boldsymbol{\theta}}_\star) \to_P 0$, which is true for $p \neq 1$ as argued above.

Putting everything together, part (a) follows. The proof for part (b) is identical and omitted.

## N.8 Proof of Lemma K.7

We will study the cases where $s = 1$ and $s = 2$ separately. For $s = 1$, we can use Lemma K.1 to show that $H_p(\tilde{\boldsymbol{\theta}}_\star) \bar{\mathbf{R}}_0 \tilde{\boldsymbol{\theta}} = c_{\star,1} H_p(\tilde{\boldsymbol{\theta}}_\star) \bar{\mathbf{R}}_0 \tilde{\boldsymbol{\theta}}_\star + o_{\mathbb{P}}(1)$. Also, by Lemma K.5, we have $H_p(\tilde{\boldsymbol{\theta}}_\star) \bar{\mathbf{R}}_0 (\tilde{\boldsymbol{\theta}}_\star) = o(1)$ in probability if $p \neq 1$, which proves the lemma.

For the case $s = 2$, we define $\tilde{\boldsymbol{\beta}} = \boldsymbol{\beta} / \|\boldsymbol{\beta}\|_2$ and write

$$H_p(\tilde{\boldsymbol{\theta}}_\star) \bar{\mathbf{R}}_0 (\tilde{\boldsymbol{\theta}})^{\circ 2} = \|\boldsymbol{\beta}\|_2^2 \, H_p(\tilde{\boldsymbol{\theta}}_\star) \bar{\mathbf{R}}_0 (\tilde{\mathbf{X}} \tilde{\boldsymbol{\beta}})^{\circ 2} = \|\boldsymbol{\beta}\|_2^2 \, H_p(\tilde{\boldsymbol{\theta}}_\star) \bar{\mathbf{R}}_0 H_2 (\tilde{\mathbf{X}} \tilde{\boldsymbol{\beta}}) + o_{\mathbb{P}}(1).$$

Now, we define $\boldsymbol{\beta}_\perp = \frac{\boldsymbol{\beta}_\star - \langle \boldsymbol{\beta}_\star, \tilde{\boldsymbol{\beta}} \rangle \tilde{\boldsymbol{\beta}}}{\|\boldsymbol{\beta}_\star - \langle \boldsymbol{\beta}_\star, \tilde{\boldsymbol{\beta}} \rangle \tilde{\boldsymbol{\beta}}\|_2}$, and set

$$\hat{\mathbf{X}} = \tilde{\mathbf{X}} - \tilde{\mathbf{X}} \tilde{\boldsymbol{\beta}} \tilde{\boldsymbol{\beta}}^\top - \tilde{\mathbf{X}} \boldsymbol{\beta}_\perp \boldsymbol{\beta}_\perp^\top.$$

By construction, we have $\hat{\mathbf{X}} \perp \!\!\! \perp \tilde{\mathbf{X}} \tilde{\boldsymbol{\beta}}, \tilde{\boldsymbol{\theta}}_\star$. Based on Conjecture 4.3, we can again replace $\mathbf{F}_0$ with $\mathbf{F}_0 = c_1 \tilde{\mathbf{X}} \mathbf{W}_0^\top + c_{>1} \mathbf{Z}$, where $\mathbf{Z} \in \mathbb{R}^{n \times d}$ is an independent random matrix with $N(0, 1)$ entries. Again, we define $\hat{\mathbf{F}}_0$ as in equation 35. Thus, $\hat{\mathbf{F}}_0 = \mathbf{F}_0 - c_1 \tilde{\mathbf{X}} \tilde{\boldsymbol{\beta}} (\mathbf{W}_0 \tilde{\boldsymbol{\beta}})^\top - c_1 \tilde{\mathbf{X}} \boldsymbol{\beta}_\perp (\mathbf{W}_0 \boldsymbol{\beta}_\perp)^\top$. As a consequence, we also have $\mathbf{F}_0 \mathbf{F}_0^\top = \hat{\mathbf{F}}_0 \hat{\mathbf{F}}_0^\top + \mathbf{V} \mathbf{D} \mathbf{V}^\top$, where $\mathbf{V} = \begin{bmatrix} \tilde{\mathbf{X}} \tilde{\boldsymbol{\beta}} & \tilde{\mathbf{X}} \boldsymbol{\beta}_\perp & \hat{\mathbf{F}}_0 \mathbf{W}_0 \tilde{\boldsymbol{\beta}} & \hat{\mathbf{F}}_0 \mathbf{W}_0 \boldsymbol{\beta}_\perp \end{bmatrix} \in \mathbb{R}^{n \times 4}$ and

$$\mathbf{D} = \begin{bmatrix} c_1^2 \langle \mathbf{W}_0 \tilde{\boldsymbol{\beta}}, \mathbf{W}_0 \tilde{\boldsymbol{\beta}} \rangle & c_1^2 \langle \mathbf{W}_0 \tilde{\boldsymbol{\beta}}, \mathbf{W}_0 \boldsymbol{\beta}_\perp \rangle & c_1 & 0 \\ c_1^2 \langle \mathbf{W}_0 \tilde{\boldsymbol{\beta}}, \mathbf{W}_0 \boldsymbol{\beta}_\perp \rangle & c_1^2 \langle \mathbf{W}_0 \boldsymbol{\beta}_\perp, \mathbf{W}_0 \boldsymbol{\beta}_\perp \rangle & 0 & c_1 \\ c_1 & 0 & 0 & 0 \\ 0 & c_1 & 0 & 0 \end{bmatrix}.$$

Using the Woodbury formula, we find that equation 36 still holds. We can write

$$
\begin{aligned}
H_p(\tilde{\boldsymbol{\theta}}_\star)^\top \bar{\mathbf{R}}_0 H_2(\tilde{\mathbf{X}}\tilde{\boldsymbol{\beta}}) =& H_p(\tilde{\boldsymbol{\theta}}_\star)^\top \hat{\mathbf{R}}_0 H_2(\tilde{\mathbf{X}}\tilde{\boldsymbol{\beta}}) \\
& - H_p(\tilde{\boldsymbol{\theta}}_\star)^\top \hat{\mathbf{R}}_0 \mathbf{V}(\mathbf{D}^{-1} + \mathbf{V}^\top \hat{\mathbf{R}}_0 \mathbf{V})^{-1}\mathbf{V}^\top \hat{\mathbf{R}}_0 H_2(\tilde{\mathbf{X}}\tilde{\boldsymbol{\beta}}).
\end{aligned}
\tag{39}
$$

The first term converges to zero for any $p \neq 2$, analogously to the argument in Section K.2.1 for the term (1,2).

To prove that the second term will also converge to zero, we first observe that the elements of $\mathbf{K} = (\mathbf{D}^{-1} + \mathbf{V}^\top \hat{\mathbf{R}}_0 \mathbf{V})^{-1}$ are all $O_\mathbb{P}(1)$. The second term will involve quantities of the form

$$
[\mathbf{K}]_{i,j} H_p(\tilde{\boldsymbol{\theta}}_\star)^\top \hat{\mathbf{R}}_0 \boldsymbol{v}_i \boldsymbol{v}_j^\top \hat{\mathbf{R}}_0 H_2(\tilde{\mathbf{X}}\tilde{\boldsymbol{\beta}}),
$$

where $\boldsymbol{v}_i$, for $i \in \{1,2,3,4\}$, is the $i$-th column of the matrix $\mathbf{V} = \begin{bmatrix} \tilde{\mathbf{X}}\tilde{\boldsymbol{\beta}} & \tilde{\mathbf{X}}\boldsymbol{\beta}_\perp & \hat{\mathbf{F}}_0\mathbf{W}_0\tilde{\boldsymbol{\beta}} & \hat{\mathbf{F}}_0\mathbf{W}_0\boldsymbol{\beta}_\perp \end{bmatrix}$. We can argue that all these terms converge to zero, as follows:

- The terms where $j = 1$ converge to zero because $(\tilde{\mathbf{X}}\tilde{\boldsymbol{\beta}})^\top \hat{\mathbf{R}}_0 H_2(\tilde{\mathbf{X}}\tilde{\boldsymbol{\beta}})$ converges to zero analogously to the argument in Section K.2.1 for the term (1,2). The same argument applies to the terms where $j = 2$, via the convergence of $(\tilde{\mathbf{X}}\boldsymbol{\beta}_\perp)^\top \hat{\mathbf{R}}_0 H_2(\tilde{\mathbf{X}}\tilde{\boldsymbol{\beta}})$ to zero.

- For $j = 3, 4$, since $H_2(\tilde{\mathbf{X}}\tilde{\boldsymbol{\beta}})$ is independent of $\hat{\mathbf{R}}_0[\hat{\mathbf{F}}_0\mathbf{W}_0\tilde{\boldsymbol{\beta}} \quad \hat{\mathbf{F}}_0\mathbf{W}_0\boldsymbol{\beta}_\perp]$, and has zero-mean i.i.d. entries, it also follows that these entries converge to zero in probability.

Finally we study $H_2(\tilde{\boldsymbol{\theta}}_\star)^\top \bar{\mathbf{R}}_0 \tilde{\boldsymbol{\theta}}^{\circ 2}$, by analyzing the terms in equation 39 for $p = 2$.

For $H_2(\tilde{\boldsymbol{\theta}}_\star)^\top \hat{\mathbf{R}}_0 H_2(\tilde{\mathbf{X}}\tilde{\boldsymbol{\beta}})$, since $H_2(\tilde{\boldsymbol{\theta}}_\star), H_2(\tilde{\mathbf{X}}\tilde{\boldsymbol{\beta}})$ are independent of $\hat{\mathbf{R}}_0$, it follows from Lemma K.3, as in the analysis of term $(1,2)$ in Section K.2, that $H_2(\tilde{\boldsymbol{\theta}}_\star)^\top \hat{\mathbf{R}}_0 H_2(\tilde{\mathbf{X}}\tilde{\boldsymbol{\beta}}) - \mathbb{E}\hat{\mathbf{R}}_0 \cdot \mathbb{E}H_2(\tilde{\boldsymbol{\theta}}_\star)^\top H_2(\tilde{\mathbf{X}}\tilde{\boldsymbol{\beta}}) \to_P 0$. Now notice that $\hat{\mathbf{F}}_0$ is left-orthogonally invariant in distribution, and thus $\hat{\mathbf{R}}_0 =_d \mathbf{O}\hat{\mathbf{R}}_0\mathbf{O}^\top$, where $\mathbf{O}$ is uniformly distributed over the Haar measure of $n$-dimensional orthogonal matrices, independently of all other randomness. Hence, $\mathbb{E}\hat{\mathbf{R}}_0 = \mathbb{E}\operatorname{tr}\hat{\mathbf{R}}_0\mathbf{I}_n/n$. Moreover, from the Woodbury formula in equation 21,

$$
\begin{aligned}
|\operatorname{tr}\bar{\mathbf{R}}_0 - \operatorname{tr}\hat{\mathbf{R}}_0| &\leq |\operatorname{tr}\hat{\mathbf{R}}_0\mathbf{V}(\mathbf{D}^{-1} + \mathbf{V}^\top\hat{\mathbf{R}}_0\mathbf{V})^{-1}\mathbf{V}^\top\hat{\mathbf{R}}_0| \\
&\leq |\operatorname{tr}(\mathbf{D}^{-1} + \mathbf{V}^\top\hat{\mathbf{R}}_0\mathbf{V})^{-1}\mathbf{V}^\top\mathbf{V}| \cdot \|\hat{\mathbf{R}}_0\|_{\text{op}}^2.
\end{aligned}
$$

From our previous analysis and as the entries of $\mathbf{V}^\top\mathbf{V}$ are $O_\mathbb{P}(n)$, it follows that the first term is $O_\mathbb{P}(n)$; whereas $\|\hat{\mathbf{R}}_0\|_{\text{op}}^2 = O(1/n^2)$. Hence, $|\operatorname{tr}\bar{\mathbf{R}}_0 - \operatorname{tr}\hat{\mathbf{R}}_0| \to_P 0$, and thus by the bounded convergence theorem $|\mathbb{E}\operatorname{tr}\bar{\mathbf{R}}_0 - \mathbb{E}\operatorname{tr}\hat{\mathbf{R}}_0| \to_P 0$. Moreover, we have already argued in the proof of Lemma K.4 that $\mathbb{E}\operatorname{tr}\bar{\mathbf{R}}_0 \to \psi m_1/\phi$.

Further, by Lemmas C.1 and K.1,

$$
\begin{aligned}
\mathbb{E}H_2(\tilde{\boldsymbol{\theta}}_\star)^\top H_2(\tilde{\mathbf{X}}\tilde{\boldsymbol{\beta}}) &= n \cdot \mathbb{E}H_2(\tilde{\boldsymbol{x}}_1^\top\boldsymbol{\beta}_\star)H_2(\tilde{\boldsymbol{x}}_1^\top\tilde{\boldsymbol{\beta}}) \\
&= 2n\mathbb{E}(\boldsymbol{\beta}_\star^\top\tilde{\boldsymbol{\beta}})^2 = 2n\mathbb{E}\frac{(\boldsymbol{\beta}_\star^\top\boldsymbol{\beta})^2}{\|\boldsymbol{\beta}\|^2} = 2n\frac{c_{\star,1}^2}{\phi(c_\star^2 + \sigma_\varepsilon^2) + c_{\star,1}^2} + o_\mathbb{P}(1).
\end{aligned}
$$

This shows that

$$
H_2(\tilde{\boldsymbol{\theta}}_\star)^\top \hat{\mathbf{R}}_0 H_2(\tilde{\mathbf{X}}\tilde{\boldsymbol{\beta}}) \to_P 2\frac{\psi m_1}{\phi}\frac{c_{\star,1}^2}{\phi(c_\star^2 + \sigma_\varepsilon^2) + c_{\star,1}^2}.
$$

Next, we consider $H_2(\tilde{\boldsymbol{\theta}}_\star)^\top \hat{\mathbf{R}}_0 \mathbf{V}$ with $\mathbf{V} = \begin{bmatrix} \tilde{\mathbf{X}}\tilde{\boldsymbol{\beta}} & \tilde{\mathbf{X}}\boldsymbol{\beta}_\perp & \hat{\mathbf{F}}_0\mathbf{W}_0\tilde{\boldsymbol{\beta}} & \hat{\mathbf{F}}_0\mathbf{W}_0\boldsymbol{\beta}_\perp \end{bmatrix}$. For the first two entries of the vector $H_2(\tilde{\boldsymbol{\theta}}_\star)^\top \hat{\mathbf{R}}_0 \mathbf{V}$, an analysis very similar to the one above for $H_2(\tilde{\boldsymbol{\theta}}_\star)^\top \hat{\mathbf{R}}_0 H_2(\tilde{\mathbf{X}}\tilde{\boldsymbol{\beta}})$ shows that they converge to zero in probability. For the last two entries, since $H_2(\tilde{\boldsymbol{\theta}}_\star)$ is independent of $\hat{\mathbf{R}}_0[\hat{\mathbf{F}}_0\mathbf{W}_0\tilde{\boldsymbol{\beta}} \quad \hat{\mathbf{F}}_0\mathbf{W}_0\boldsymbol{\beta}_\perp]$, and has zero-mean i.i.d. entries, it also follows that these entries converge to zero in probability. Moreover, the limiting entries of $(\mathbf{D}^{-1} + \mathbf{V}^\top\hat{\mathbf{R}}_0\mathbf{V})^{-1}$ have been shown to be bounded in our above analysis. Hence, the second term converges to zero in probability.

Now, note that $\tilde{\beta} = \beta/\|\beta\|_2$. From Lemma K.1, $\|\beta\|^2 \to_P \phi(c_\star^2 + \sigma_\varepsilon^2) + c_{\star,1}^2$. Hence,

$$H_2(\tilde{\theta}_\star)^\top \hat{\mathbf{R}}_0 H_2(\tilde{\mathbf{X}}\beta) = 2\frac{\psi m_1}{\phi} \frac{c_{\star,1}^2 \|\beta\|_2^2}{\phi(c_\star^2 + \sigma_\varepsilon^2) + c_{\star,1}^2} + o_\mathbb{P}(1) \to_P \frac{2c_{\star,1}^2 \psi m_1}{\phi},$$

which concludes the proof.

### N.9 PROOF OF LEMMA K.8

As in the proof of Lemma K.6, we define $\hat{\mathbf{X}} = \tilde{\mathbf{X}} - \tilde{\theta}\beta^\top$. By construction, we have $\hat{\mathbf{X}} \perp \tilde{\theta}$. As in the proof of Lemma K.5, based on Conjecture 4.3, we can replace $\mathbf{F}_0$ with $c_1 \mathbf{X}\mathbf{W}_0^\top + c_{>1}\mathbf{Z}$ in our computations without changing the limiting result, where $\mathbf{Z} \in \mathbb{R}^{n \times d}$ is an independent random matrix with $\mathsf{N}(0,1)$ entries. Thus, from now on, we denote $\mathbf{F}_0 = c_1 \mathbf{X}\mathbf{W}_0^\top + c_{>1}\mathbf{Z}$. We define $\hat{\mathbf{F}}_0$ as in equation 35; thus, $\hat{\mathbf{F}}_0 = \mathbf{F}_0 - c_1 \tilde{\theta}(\mathbf{W}_0\beta)^\top$. As a consequence, we can write $\mathbf{F}_0 \mathbf{F}_0^\top = \hat{\mathbf{F}}_0 \hat{\mathbf{F}}_0^\top + \mathbf{V}\mathbf{D}\mathbf{V}^\top$, where $\mathbf{V} = \begin{bmatrix} \hat{\mathbf{F}}_0 \mathbf{W}_0\beta & \tilde{\theta} \end{bmatrix} \in \mathbb{R}^{n \times 2}$ and

$$\mathbf{D} = \begin{bmatrix} 0 & c_1 \\ c_1 & c_1^2 \|\mathbf{W}_0\beta\|_2^2 \end{bmatrix}.$$

Using the Woodbury formula, we find that equation 36 still holds. Now, we can write

$$\tilde{\theta}^{\circ 2 \top} \bar{\mathbf{R}}_0 \tilde{\theta}^{\circ 2} = \tilde{\theta}^{\circ 2 \top} \hat{\mathbf{R}}_0 \tilde{\theta}^{\circ 2} - \tilde{\theta}^{\circ 2 \top} \hat{\mathbf{R}}_0 \mathbf{V}(\mathbf{D}^{-1} + \mathbf{V}^\top \hat{\mathbf{R}}_0 \mathbf{V})^{-1} \mathbf{V}^\top \hat{\mathbf{R}}_0 \tilde{\theta}^{\circ 2}. \tag{40}$$

We can analyze each term in the above sum separately.

By Lemma K.3, $\tilde{\theta}^{\circ 2 \top} \hat{\mathbf{R}}_0 \tilde{\theta}^{\circ 2} - \mathbb{E}\tilde{\theta}^{\circ 2 \top} \hat{\mathbf{R}}_0 \tilde{\theta}^{\circ 2} \to_P 0$. Further, conditional on $\beta$, $\mathbb{E}\tilde{\theta}^{\circ 2 \top} \hat{\mathbf{R}}_0 \tilde{\theta}^{\circ 2} = 3\|\beta\|_2^4 \mathbb{E} \operatorname{tr} \hat{\mathbf{R}}_0$; and as in the proof of Lemma K.7, $\mathbb{E} \operatorname{tr} \hat{\mathbf{R}}_0 - \mathbb{E} \operatorname{tr} \bar{\mathbf{R}}_0 \to 0$. Moreover, we have already argued in the proof of Lemma K.4 that $\mathbb{E} \operatorname{tr} \bar{\mathbf{R}}_0 \to \psi m_1/\phi$. In addition, from Lemma K.1, $\|\beta\|^2 \to_P \phi(c_\star^2 + \sigma_\varepsilon^2) + c_{\star,1}^2$. Hence,

$$\tilde{\theta}^{\circ 2 \top} \hat{\mathbf{R}}_0 \tilde{\theta}^{\circ 2} \to_P 3\psi m_1 [\phi(c_\star^2 + \sigma_\varepsilon^2) + c_{\star,1}^2]^2/\phi.$$

To analyze the second term in equation 40, we first study $\tilde{\theta}^{\circ 2 \top} \hat{\mathbf{R}}_0 \hat{\mathbf{F}}_0 \mathbf{W}_0\beta$. By an argument similar to the ones above, we can show that it concentrates around $\mathbf{1}_n^\top \hat{\mathbf{R}}_0 \hat{\mathbf{F}}_0 \mathbf{W}_0\beta = \mathbf{1}_n^\top \hat{\mathbf{F}}_0 \hat{\mathbf{R}}_0 \mathbf{W}_0\beta$. Since $\hat{\mathbf{F}}_0$ is left-orthogonally invariant, $\mathbf{1}_n^\top \hat{\mathbf{F}}_0 \hat{\mathbf{R}}_0 \mathbf{W}_0\beta =_d \mathbf{1}_n^\top \mathbf{O}\hat{\mathbf{F}}_0 \hat{\mathbf{R}}_0 \mathbf{W}_0\beta$, where $\mathbf{O}$ is uniformly distributed over the Haar measure of $n$-dimensional orthogonal matrices, independently of all other randomness. Then, it follows as in the analysis of term (1,2) from Section K.2 that $\mathbf{1}_n^\top \mathbf{O}\hat{\mathbf{F}}_0 \hat{\mathbf{R}}_0 \mathbf{W}_0\beta \to_P 0$; and hence $\tilde{\theta}^{\circ 2 \top} \hat{\mathbf{R}}_0 \hat{\mathbf{F}}_0 \mathbf{W}_0\beta \to_P 0$.

Moreover, the limiting entries of $(\mathbf{D}^{-1} + \mathbf{V}^\top \hat{\mathbf{R}}_0 \mathbf{V})^{-1}$ can be shown to be bounded by a simple orderwise analysis. Hence, the second term in equation 40 is $o_\mathbb{P}(1)$.

### N.10 PROOF OF LEMMA L.2

Denoting $\tilde{\beta} = \beta/\|\beta\|_2$, we have

$$(\tilde{\mathbf{X}}\beta)^{\circ i \top} \bar{\mathbf{R}}_0 (\tilde{\mathbf{X}}\beta)^{\circ j} = \|\beta\|_2^{i+j} (\tilde{\mathbf{X}}\tilde{\beta})^{\circ i \top} \bar{\mathbf{R}}_0 (\tilde{\mathbf{X}}\tilde{\beta})^{\circ j}$$

$$= \|\beta\|_2^{i+j} \sum_{k_1=0}^{i} \sum_{k_2=0}^{j} \xi_{i,k_1} \xi_{j,k_2} H_{k_1}(\tilde{\mathbf{X}}\tilde{\beta})^\top \bar{\mathbf{R}}_0 H_{k_2}(\tilde{\mathbf{X}}\tilde{\beta})$$

$$= \|\beta\|_2^{i+j} \sum_{k=0}^{\min(i,j)} \xi_{j,k} \xi_{i,k} H_k(\tilde{\mathbf{X}}\tilde{\beta})^\top \bar{\mathbf{R}}_0 H_k(\tilde{\mathbf{X}}\tilde{\beta}) + o_\mathbb{P}(1)$$

$$= \|\beta\|_2^{i+j} \left[ \xi_{i,1}\xi_{j,1}(\tilde{\mathbf{X}}\tilde{\beta})^\top \bar{\mathbf{R}}_0 (\tilde{\mathbf{X}}\tilde{\beta}) + \sum_{k=0,\ k\neq 1}^{\min(i,j)} \xi_{i,k}\xi_{j,k} H_k(\tilde{\mathbf{X}}\tilde{\beta})^\top \bar{\mathbf{R}}_0 H_k(\tilde{\mathbf{X}}\tilde{\beta}) \right] + o_\mathbb{P}(1).$$

The third line follows from Lemma K.5. Now, we claim that for any $p \in \{0, 2, 3, \dots\}$, we have $H_p(\tilde{\mathbf{X}}\beta/\|\beta\|_2)^\top \bar{\mathbf{R}}_0 H_p(\tilde{\mathbf{X}}\beta/\|\beta\|_2) \to_P p! \ \psi m_1/\phi$. Using this claim, the facts that $\|\beta\|_2^2 \to_P$

$c_{\star,1}^2 + \phi(c_\star^2 + \sigma_\varepsilon^2)$, and $\mathrm{tr}(\tilde{\mathbf{X}}^\top(\mathbf{F}_0\mathbf{F}_0^\top + \lambda n\mathbf{I}_n)^{-1}\tilde{\mathbf{X}})/d \to_P \psi m_2/\phi$, we can conclude

$$(\tilde{\mathbf{X}}\boldsymbol{\beta})^{\circ i\top}\bar{\mathbf{R}}_0(\tilde{\mathbf{X}}\boldsymbol{\beta})^{\circ j} \to_P \left(c_{\star,1}^2 + \phi(c_\star^2 + \sigma_\varepsilon^2)\right)^{(i+j)/2}\left[\xi_{i,1}\xi_{j,1}\frac{\psi m_2}{\phi} + \frac{\psi m_1}{\phi}\sum_{k=0,\ k\neq 1}^{\min(i,j)} k!\,\xi_{i,k}\xi_{j,k}\right].$$

Now, it remains to prove the claim that for any $p \in \{0, 2, 3, \dots\}$, we have

$$H_p(\tilde{\mathbf{X}}\boldsymbol{\beta}/\|\boldsymbol{\beta}\|_2)^\top\bar{\mathbf{R}}_0 H_p(\tilde{\mathbf{X}}\boldsymbol{\beta}/\|\boldsymbol{\beta}\|_2) \to_P p!\,\psi m_1/\phi.$$

As in the proof of Lemma K.8, we define $\hat{\mathbf{X}} = \tilde{\mathbf{X}} - \tilde{\mathbf{X}}\tilde{\boldsymbol{\beta}}\tilde{\boldsymbol{\beta}}^\top$. By construction, we have $\hat{\mathbf{X}} \perp\!\!\!\perp \tilde{\mathbf{X}}\tilde{\boldsymbol{\beta}}$. As in the proof of Lemma K.5, based on Conjecture 4.3, we can replace $\mathbf{F}_0$ with $c_1\tilde{\mathbf{X}}\mathbf{W}_0^\top + c_{>1}\mathbf{Z}$ in our computations without changing the limiting result, where $\mathbf{Z} \in \mathbb{R}^{n\times d}$ is an independent random matrix with $\mathsf{N}(0,1)$ entries. Thus, from now on, we denote $\mathbf{F}_0 = c_1\tilde{\mathbf{X}}\mathbf{W}_0^\top + c_{>1}\mathbf{Z}$. We define $\hat{\mathbf{F}}_0$ as in equation 35; thus, $\hat{\mathbf{F}}_0 = \mathbf{F}_0 - c_1\tilde{\mathbf{X}}\tilde{\boldsymbol{\beta}}(\mathbf{W}_0\tilde{\boldsymbol{\beta}})^\top$. As a consequence, we can write $\mathbf{F}_0\mathbf{F}_0^\top = \hat{\mathbf{F}}_0\hat{\mathbf{F}}_0^\top + \mathbf{V}\mathbf{D}\mathbf{V}^\top$, where $\mathbf{V} = \begin{bmatrix}\hat{\mathbf{F}}_0\mathbf{W}_0\tilde{\boldsymbol{\beta}} & \tilde{\mathbf{X}}\tilde{\boldsymbol{\beta}}\end{bmatrix} \in \mathbb{R}^{n\times 2}$ and

$$\mathbf{D} = \begin{bmatrix} 0 & c_1 \\ c_1 & c_1^2\|\mathbf{W}_0\tilde{\boldsymbol{\beta}}\|_2^2 \end{bmatrix}.$$

Using the Woodbury formula, we find that equation 36 still holds. Now, we can write

$$H_p(\tilde{\mathbf{X}}\tilde{\boldsymbol{\beta}})^\top\bar{\mathbf{R}}_0 H_p(\tilde{\mathbf{X}}\tilde{\boldsymbol{\beta}})$$
$$= H_p(\tilde{\mathbf{X}}\tilde{\boldsymbol{\beta}})^\top\hat{\mathbf{R}}_0 H_p(\tilde{\mathbf{X}}\tilde{\boldsymbol{\beta}}) - H_p(\tilde{\mathbf{X}}\tilde{\boldsymbol{\beta}})^\top\hat{\mathbf{R}}_0\mathbf{V}(\mathbf{D}^{-1} + \mathbf{V}^\top\hat{\mathbf{R}}_0\mathbf{V})^{-1}\mathbf{V}^\top\hat{\mathbf{R}}_0 H_p(\tilde{\mathbf{X}}\tilde{\boldsymbol{\beta}}). \quad (41)$$

We can analyze each term in the above sum separately.

By Lemma K.3, $H_p(\tilde{\mathbf{X}}\tilde{\boldsymbol{\beta}})^\top\hat{\mathbf{R}}_0 H_p(\tilde{\mathbf{X}}\tilde{\boldsymbol{\beta}}) - \mathbb{E}H_p(\tilde{\mathbf{X}}\tilde{\boldsymbol{\beta}})^\top\hat{\mathbf{R}}_0 H_p(\tilde{\mathbf{X}}\tilde{\boldsymbol{\beta}}) \to_P 0$. Further, conditional on $\tilde{\boldsymbol{\beta}}$, and using C.1, we have

$$\mathbb{E}H_p(\tilde{\mathbf{X}}\tilde{\boldsymbol{\beta}})^\top\hat{\mathbf{R}}_0 H_p(\tilde{\mathbf{X}}\tilde{\boldsymbol{\beta}}) = \mathbb{E}\,\mathrm{tr}\left[\hat{\mathbf{R}}_0 H_p(\tilde{\mathbf{X}}\tilde{\boldsymbol{\beta}})H_p(\tilde{\mathbf{X}}\tilde{\boldsymbol{\beta}})^\top\right] = p!\,\mathbb{E}\,\mathrm{tr}\left[\hat{\mathbf{R}}_0\right],$$

and as in the proof of Lemma K.7, $\mathbb{E}\,\mathrm{tr}\,\hat{\mathbf{R}}_0 - \mathbb{E}\,\mathrm{tr}\,\bar{\mathbf{R}}_0 \to 0$. Moreover, we have already argued in the proof of Lemma K.4 that $\mathbb{E}\,\mathrm{tr}\,\bar{\mathbf{R}}_0 \to \psi m_1/\phi$. Hence,

$$H_p(\tilde{\mathbf{X}}\tilde{\boldsymbol{\beta}})^\top\hat{\mathbf{R}}_0 H_p(\tilde{\mathbf{X}}\tilde{\boldsymbol{\beta}}) \to_P p!\,\psi m_1/\phi.$$

To analyze the second term in equation 41, we first study $H_p(\tilde{\mathbf{X}}\tilde{\boldsymbol{\beta}})^\top\hat{\mathbf{R}}_0\hat{\mathbf{F}}_0\mathbf{W}_0\boldsymbol{\beta}$. Conditional on $\tilde{\boldsymbol{\beta}}$, $H_p(\tilde{\mathbf{X}}\tilde{\boldsymbol{\beta}})$ is a vector with independent mean-zero, bounded variance entries, independent of the vector $\hat{\mathbf{R}}_0\hat{\mathbf{F}}_0\mathbf{W}_0\tilde{\boldsymbol{\beta}}$ that has norm $O(1/\sqrt{n})$. Hence, we conclude that this term goes to zero. Next, note that $H_p(\tilde{\mathbf{X}}\tilde{\boldsymbol{\beta}})^\top\hat{\mathbf{R}}_0(\tilde{\mathbf{X}}\tilde{\boldsymbol{\beta}}) \to_P 0$ using Lemma K.3 and Lemma C.1. Moreover, the limiting entries of $(\mathbf{D}^{-1} + \mathbf{V}^\top\hat{\mathbf{R}}_0\mathbf{V})^{-1}$ can be shown to be bounded by a simple orderwise analysis. Hence, the second term in equation 41 is $o_{\mathbb{P}}(1)$. This concludes the proof.

## N.11 PROOF OF LEMMA L.3

We define $\boldsymbol{\beta}_\perp = \frac{\boldsymbol{\beta}_\star - \langle\boldsymbol{\beta}_\star,\tilde{\boldsymbol{\beta}}\rangle\tilde{\boldsymbol{\beta}}}{\|\boldsymbol{\beta}_\star - \langle\boldsymbol{\beta}_\star,\tilde{\boldsymbol{\beta}}\rangle\tilde{\boldsymbol{\beta}}\|_2}$, and set

$$\hat{\mathbf{X}} = \tilde{\mathbf{X}} - \tilde{\mathbf{X}}\tilde{\boldsymbol{\beta}}\tilde{\boldsymbol{\beta}}^\top - \tilde{\mathbf{X}}\boldsymbol{\beta}_\perp\boldsymbol{\beta}_\perp^\top.$$

By construction, we have $\hat{\mathbf{X}} \perp\!\!\!\perp \tilde{\mathbf{X}}\tilde{\boldsymbol{\beta}}, \boldsymbol{\theta}_\star$. Based on Conjecture 4.3, we can again replace $\mathbf{F}_0$ with $\mathbf{F}_0 = c_1\tilde{\mathbf{X}}\mathbf{W}_0^\top + c_{>1}\mathbf{Z}$, where $\mathbf{Z} \in \mathbb{R}^{n\times d}$ is an independent random matrix with $\mathsf{N}(0,1)$ entries. Again, we define $\hat{\mathbf{F}}_0$ as in equation 35. Thus, $\hat{\mathbf{F}}_0 = \mathbf{F}_0 - c_1\tilde{\mathbf{X}}\tilde{\boldsymbol{\beta}}(\mathbf{W}_0\tilde{\boldsymbol{\beta}})^\top - c_1\tilde{\mathbf{X}}\boldsymbol{\beta}_\perp(\mathbf{W}_0\boldsymbol{\beta}_\perp)^\top$. As a consequence, we also have $\mathbf{F}_0\mathbf{F}_0^\top = \hat{\mathbf{F}}_0\hat{\mathbf{F}}_0^\top + \mathbf{V}\mathbf{D}\mathbf{V}^\top$, where $\mathbf{V} = \begin{bmatrix}\tilde{\mathbf{X}}\tilde{\boldsymbol{\beta}} & \tilde{\mathbf{X}}\boldsymbol{\beta}_\perp & \hat{\mathbf{F}}_0\mathbf{W}_0\tilde{\boldsymbol{\beta}} & \hat{\mathbf{F}}_0\mathbf{W}_0\boldsymbol{\beta}_\perp\end{bmatrix} \in \mathbb{R}^{n\times 4}$ and

$$\mathbf{D} = \begin{bmatrix} c_1^2\langle\mathbf{W}_0\tilde{\boldsymbol{\beta}},\mathbf{W}_0\tilde{\boldsymbol{\beta}}\rangle & c_1^2\langle\mathbf{W}_0\tilde{\boldsymbol{\beta}},\mathbf{W}_0\boldsymbol{\beta}_\perp\rangle & c_1 & 0 \\ c_1^2\langle\mathbf{W}_0\tilde{\boldsymbol{\beta}},\mathbf{W}_0\boldsymbol{\beta}_\perp\rangle & c_1^2\langle\mathbf{W}_0\boldsymbol{\beta}_\perp,\mathbf{W}_0\boldsymbol{\beta}_\perp\rangle & 0 & c_1 \\ c_1 & 0 & 0 & 0 \\ 0 & c_1 & 0 & 0 \end{bmatrix}.$$

Using the Woodbury formula, we find that equation 36 still holds. We can write

$$
\begin{aligned}
H_p(\tilde{\boldsymbol{\theta}}_\star)^\top \bar{\mathbf{R}}_0 H_q(\tilde{\mathbf{X}}\tilde{\boldsymbol{\beta}}) =& H_p(\tilde{\boldsymbol{\theta}}_\star)^\top \hat{\mathbf{R}}_0 H_q(\tilde{\mathbf{X}}\tilde{\boldsymbol{\beta}}) \\
& - H_p(\tilde{\boldsymbol{\theta}}_\star)^\top \hat{\mathbf{R}}_0 \mathbf{V}(\mathbf{D}^{-1} + \mathbf{V}^\top \hat{\mathbf{R}}_0 \mathbf{V})^{-1}\mathbf{V}^\top \hat{\mathbf{R}}_0 H_q(\tilde{\mathbf{X}}\tilde{\boldsymbol{\beta}}).
\end{aligned}
\tag{42}
$$

$p \neq q$ **case:** The first term converges to zero for any $p \neq q$, analogously to the argument in Section K.2.1 for the terms (1,2) and (2,4). In particular, for $p = 0$, we can use orthogonal invariance as in the analysis of the term (2,4). To prove that the second term will also converge to zero when $p \neq q$, we first observe that the elements of $\mathbf{K} = (\mathbf{D}^{-1} + \mathbf{V}^\top \hat{\mathbf{R}}_0 \mathbf{V})^{-1}$ are all $O(1)$. The second term will involve quantities of the form

$$
[\mathbf{K}]_{i,j} H_p(\tilde{\boldsymbol{\theta}}_\star)^\top \hat{\mathbf{R}}_0 \boldsymbol{v}_i \boldsymbol{v}_j^\top \hat{\mathbf{R}}_0 H_q(\tilde{\mathbf{X}}\tilde{\boldsymbol{\beta}}),
$$

where $\boldsymbol{v}_i$, for $i \in \{1,2,3,4\}$, is the $i$-th column of the matrix $\mathbf{V} = \begin{bmatrix} \tilde{\mathbf{X}}\tilde{\boldsymbol{\beta}} & \tilde{\mathbf{X}}\boldsymbol{\beta}_\perp & \hat{\mathbf{F}}_0 \mathbf{W}_0 \tilde{\boldsymbol{\beta}} & \hat{\mathbf{F}}_0 \mathbf{W}_0 \boldsymbol{\beta}_\perp \end{bmatrix}$. We can argue that all these terms converge to zero, as follows. Because $p \neq q$, without loss of generality, assume that $q \neq 1$.

- The terms where $j = 1$ converge to zero because $(\tilde{\mathbf{X}}\tilde{\boldsymbol{\beta}})^\top \hat{\mathbf{R}}_0 H_q(\tilde{\mathbf{X}}\tilde{\boldsymbol{\beta}})$ converges to zero using the concentration argument from Lemma K.3 and the orthogonality of Hermite polynomials from Lemma C.1. The same argument applies to the terms where $j = 2$, via the convergence of $(\tilde{\mathbf{X}}\boldsymbol{\beta}_\perp)^\top \hat{\mathbf{R}}_0 H_q(\tilde{\mathbf{X}}\tilde{\boldsymbol{\beta}})$ to zero.

- For $j = 3,4$, and for $q > 0$, since $H_q(\tilde{\mathbf{X}}\tilde{\boldsymbol{\beta}})$ is independent of $\hat{\mathbf{R}}_0[\hat{\mathbf{F}}_0 \mathbf{W}_0 \tilde{\boldsymbol{\beta}} \quad \hat{\mathbf{F}}_0 \mathbf{W}_0 \boldsymbol{\beta}_\perp]$, and has zero-mean i.i.d. entries, it also follows that these entries converge to zero in probability. For $q = 0$, we can again use orthogonal invariance as in the analysis of the term (2,4).

**The case when $p = q \neq 1$:** Finally we study $H_p(\tilde{\boldsymbol{\theta}}_\star)^\top \bar{\mathbf{R}}_0 H_p(\tilde{\mathbf{X}}\tilde{\boldsymbol{\beta}})$, by analyzing the terms in equation 39.

For $H_p(\tilde{\boldsymbol{\theta}}_\star)^\top \hat{\mathbf{R}}_0 H_p(\tilde{\mathbf{X}}\tilde{\boldsymbol{\beta}})$, since $H_p(\tilde{\boldsymbol{\theta}}_\star), H_p(\tilde{\mathbf{X}}\tilde{\boldsymbol{\beta}})$ are independent of $\hat{\mathbf{R}}_0$, it follows from Lemma K.3, as in the analysis of term $(1,2)$ in the Section K.2, that $H_p(\tilde{\boldsymbol{\theta}}_\star)^\top \hat{\mathbf{R}}_0 H_p(\tilde{\mathbf{X}}\tilde{\boldsymbol{\beta}}) - \mathbb{E}\hat{\mathbf{R}}_0 \cdot \mathbb{E}H_p(\tilde{\boldsymbol{\theta}}_\star)^\top H_p(\tilde{\mathbf{X}}\tilde{\boldsymbol{\beta}}) \to_P 0$. Now notice that $\hat{\mathbf{F}}_0$ is left-orthogonally invariant in distribution, and thus $\hat{\mathbf{R}}_0 =_d \mathbf{O}\hat{\mathbf{R}}_0 \mathbf{O}^\top$, where $\mathbf{O}$ is uniformly distributed over the Haar measure of $n$-dimensional orthogonal matrices, independently of all other randomness. Hence, $\mathbb{E}\hat{\mathbf{R}}_0 = \mathbb{E}\operatorname{tr}\hat{\mathbf{R}}_0 \mathbf{I}_n/n$. Also, similar to the proof of Lemma K.7, we have $|\operatorname{tr}\bar{\mathbf{R}}_0 - \operatorname{tr}\hat{\mathbf{R}}_0| = o_\mathbb{P}(1)$. Moreover, we have already argued in the proof of Lemma K.4 that $\mathbb{E}\operatorname{tr}\bar{\mathbf{R}}_0 \to \psi m_1/\phi$. Further, by Lemmas C.1 and K.1,

$$
H_p(\tilde{\boldsymbol{\theta}}_\star)^\top \hat{\mathbf{R}}_0 H_p(\tilde{\mathbf{X}}\tilde{\boldsymbol{\beta}}) \to_P p! \frac{\psi m_1}{\phi} \left( \frac{c_{\star,1}}{\sqrt{\phi(c_\star^2 + \sigma_\varepsilon^2) + c_{\star,1}^2}} \right)^p.
$$

Next, we consider $H_2(\tilde{\boldsymbol{\theta}}_\star)^\top \hat{\mathbf{R}}_0 \mathbf{V}$ with $\mathbf{V} = \begin{bmatrix} \tilde{\mathbf{X}}\tilde{\boldsymbol{\beta}} & \tilde{\mathbf{X}}\boldsymbol{\beta}_\perp & \hat{\mathbf{F}}_0 \mathbf{W}_0 \tilde{\boldsymbol{\beta}} & \hat{\mathbf{F}}_0 \mathbf{W}_0 \boldsymbol{\beta}_\perp \end{bmatrix}$. For the first two entries of the vector $H_p(\tilde{\boldsymbol{\theta}}_\star)^\top \hat{\mathbf{R}}_0 \mathbf{V}$, an analysis very similar to the one above for $H_p(\tilde{\boldsymbol{\theta}}_\star)^\top \hat{\mathbf{R}}_0 H_p(\tilde{\mathbf{X}}\tilde{\boldsymbol{\beta}})$ shows that they converge to zero in probability. For the last two entries, since $H_p(\tilde{\boldsymbol{\theta}}_\star)$ is independent of $\hat{\mathbf{R}}_0[\hat{\mathbf{F}}_0 \mathbf{W}_0 \tilde{\boldsymbol{\beta}} \quad \hat{\mathbf{F}}_0 \mathbf{W}_0 \boldsymbol{\beta}_\perp]$, and has zero-mean i.i.d. entries, it also follows that these entries converge to zero in probability. Moreover, the limiting entries of $(\mathbf{D}^{-1} + \mathbf{V}^\top \hat{\mathbf{R}}_0 \mathbf{V})^{-1}$ have been shown to be bounded in our above analysis. Hence, the second term converges to zero in probability.

**The case when $p = q = 1$:** In this case, we have

$$
(\tilde{\mathbf{X}}\boldsymbol{\beta}_\star)^\top \bar{\mathbf{R}}_0 (\tilde{\mathbf{X}}\tilde{\boldsymbol{\beta}}) = \frac{(\tilde{\mathbf{X}}\boldsymbol{\beta}_\star)^\top \bar{\mathbf{R}}_0(\tilde{\mathbf{X}}\boldsymbol{\beta})}{\|\boldsymbol{\beta}\|_2} = \frac{c_{\star,1}\frac{\psi m_2}{\phi}}{\sqrt{\phi(c_\star^2 + \sigma_\varepsilon^2) + c_{\star,1}^2}} + o_\mathbb{P}(1),
$$

using Lemma K.1 and by arguments similar to the ones in the proof of Lemma K.4.

Putting everything together concludes the proof.

## O    ADDITIONAL EXPERIMENTS

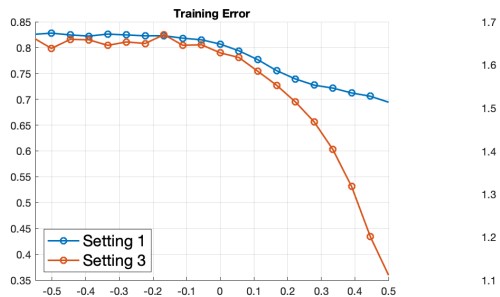 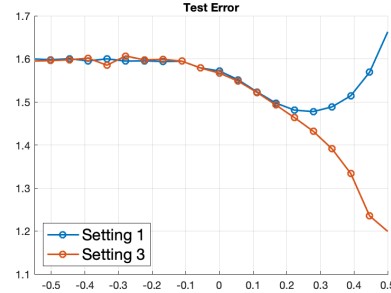

Figure 4: We repeat the experiments in Figure 3 (**Left, Middle**) with $y = H_1(\boldsymbol{\beta}_\star^\top \boldsymbol{x}) + \frac{1}{\sqrt{6}} H_3(\boldsymbol{\beta}_\star^\top \boldsymbol{x})$ as setting 3. Here we use the activation $\sigma(x) = \frac{e^x - e^{-x}}{e^x + e^{-x}}$ so that $c_3 \neq 0$.

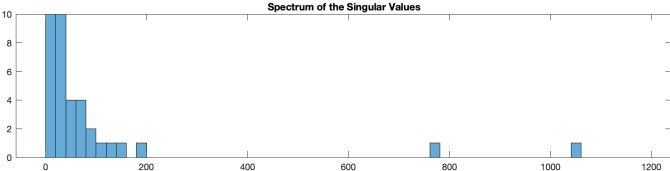

Figure 5: We repeat the experiment in Figure 2 with the MNIST dataset. Although the MNIST dataset does not satisfy our theoretical conditions (Gaussian input, single-index model, etc.), we empirically observe similar phenomena such as emergence of spikes after one-step gradient update.

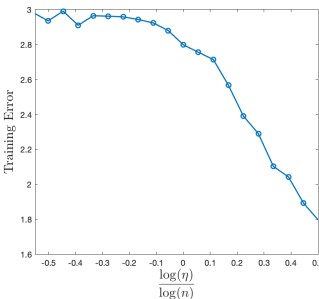 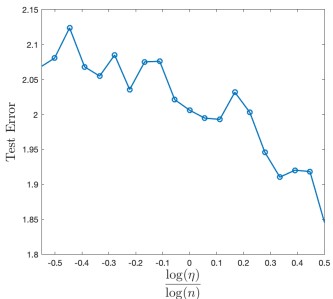

Figure 6: We plot the training and test error of a two-layer neural network ($N = 1000$) trained on the MNIST dataset with one step of gradient descent of varying step size. In order to make the experiments compatible with our theoretical setup, the model is trained using the MSE loss. We demonstrate that huge step size can still be beneficial in this more realistic problem.

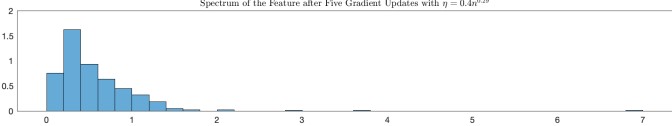

Figure 7: Singular value spectrum of the feature matrix after 5 gradient updates. We use the same experimental setting as Figure 2.