# OpenReview forum: "A Theory of Non-Linear Feature Learning with One Gradient Step in Two-Layer Neural Networks"
_ICLR.cc/2024/Conference — Submitted to ICLR 2024_

### Official Review · Reviewer_nmYk · 2023-10-30

**Soundness:** 3 good
**Presentation:** 2 fair
**Contribution:** 2 fair
**Rating:** 5
**Confidence:** 3

**Summary:**

This paper studies how neural networks learn the features with different learning rates. Specifically, this paper considers a setup of two-layer neural network under one-step GD with the step size $\eta=n^\alpha$, where $n$ is the number of training samples.
This paper provides a theoretical characterization of how a specific choice of $\alpha$ influences the neural network's ability to learn various types of features.

**Strengths:**

The paper goes beyond the NTK region and characterizes the performance of the neural networks with a relatively large learning rate. Instead of doing lazy training, this paper shows the ability of neural networks to learn features and characterize the relationship between the learning rate and the learned features by the weights.

**Weaknesses:**

1. Assuming that the data follows a zero-mean Gaussian distribution is strong, and whether real-world data satisfies this assumption can vary. Specifically, when considering Gaussian data, it is assumed to exhibit symmetric properties, which are necessary for the proofs. However, it's important to recognize that not all real-world data inherently possesses such symmetrical properties.

2. The assumptions for the activation function are unclear. I had a hard time understanding Condition 2.3 and Condition 2.4. Could the authors directly tell us which activation functions satisfy these conditions?

3. More discussions are needed for the magnitude of $c_\star$. For example, for some common activation functions, like ReLU and Sigmoid,  where $M$ can go to infinity, I would like to learn about the dependence of the sample complexity.

4. I noticed that the theoretical results regarding the training loss are only provided for the cases where $\ell$ equals 1 and 2. While the authors mentioned that results for the general case of $\ell$ can be found in the Appendix, I believe it is essential to include and discuss these results in the main content.

5. Show the decrease in training loss may not be surprising.  Instead, the focus should be on generalization and test error.

6. Emphasizing the technical challenges involved in deriving the proofs would make it easier to appreciate the technical novelty. Currently, it's challenging to discern these technical aspects when looking at the LONG proofs in the Appendix.

7. We need some experiments of higher-order feature learning to justify the theoretical findings in Theorem 4.1.

8. We need some numerical experiments on real data and deep neural networks to justify the theoretical findings. Currently, due to the strong assumptions on the input data and activation functions, it's challenging to envision practical applications.

**Questions:**

1. The setup of this paper and Zhenmei et al.2022 seems significantly different from my POV, but both are counted as feature learnings. It would be better to clarify the field of feature learning and how they are connected.

2. The values of training error and test error are quite confusing. It would be helpful if the authors included some baseline measures to assess the algorithm's performance. Specifically, it's challenging to see the significance of studying an algorithm with a large test error.  While it may be caused by the scaling issue, the authors should consider making adjustments to avoid any misunderstanding.

3. How does Figure 2 change as the number of iterations increases?

---

> ### Author Response · Authors · 2023-11-17
> **Part 1/2**
>
> We thank the reviewer for their valuable comments.
> We tried to answer each point in detail below.
> Please let us know if the reviewer has other concerns/questions regarding our paper.
> We are happy to discuss this further.
>
> We have uploaded a revised version of the manuscript on OpenReview. Please note that the numbering of theorems and sections has changed in the revised version. We are using the new numbering in our rebuttal.
>
> **Weaknesses**
>
> 1- Isotropic Gaussian input is a standard assumption in many theoretical studies. See e.g. (Mei \& Montanari (2022); Hu \& Lu (2023); Adlam et al. (2022); Damian et al. (2022)).
> Some of previous work adds more structure to the input by allowing general covariance structure to study specific problems in learning such as covariate shift (see e.g., Mel \& Pennington (2023); Tripuraneni et al. (2022)). However, this often requires very technical tools such as operator-valued free probability, and is an interesting direction for future work.
> Mean zero assumption can be also relaxed by techniques similar to Tripuraneni et al. (2022); Lee et al. (2023). We have mentioned this point in the limitations section in the revised manuscript.
>
>
> 2- Since the Hermite polynomials form an orthonormal basis of $L^2(\mu)$, where $\mu$ is the standard Gaussian measure on $\mathbb{R}$, any activation in $L^2(\mu)$ admits the Hermite expansion.
> In Condition 2.3, we additionally require that $c_1 \neq 0$ and a certain decay rate.
> In the limitation section, we discuss why this assumption is necessary.
> Many popular activation functions including (centered) ReLU, tanh, sigmoid satisfy Condition 2.3.
> We added this discussion in the revision.
>
> 3- In the revision, we changed Condition 2.4 to infinite sum.
> As mentioned above, as long as $\sigma_\star$ is square-integrable, it will admit a Hermite expansion.
> At the moment, we are not sure how this is related to sample complexity.
> We would be happy to provide answers if you could clarify your question on the sample complexity.
>
> 4- We moved the general $\ell$ result to the main text in the revision.
> We note that due to space limitations, it was not possible to include the full formula for the limiting error.
>
> 5- We added a limitations section to the end of this paper discussing this point. In Section 4 of the paper, we wanted to demonstrate that the emergence of spikes in the spectrum of the feature matrix will indeed translate to going beyond linear models for large enough step size through the analysis of training loss. Unlike the training loss, the test error does not allow a simple expression such as Lemma K.2. Specifically, the bias term in the test error will contain a term involving two resolvent matrices with the covariance matrix of the learned features multiplied in the middle (see equation C.10 in Ba et al. (2022)). The strategy that we use to simplify such expressions is to use Woodbury formula to take out the spike terms from the resolvent matrices and study terms appearing in the expansion. However, for the case of this complicated formula, one needs to study various forms of interactions between the spike terms. We find that this would require a much more laborious calculation, and since our paper is already quite long (cca 50 pages total), we believe it is better suited for follow-up work.
>
> 6- We have added *proof idea* sections after the theorems to briefly emphasize the techniques and challenges involved in proving each result.
>
> 7- We repeated the experiment in Section 5 with a target function with cubic term.
> See Figure 4 in the revised manuscript.
> We observe that the cubic part is learned with a large enough step size.
>
>
> 8- In the supplementary material, we included additional experiments on the MNIST dataset (see Figure 5 and 6 in the revised manuscript).
> We plot the histogram of the singular values of the feature matrix after one gradient step with $\eta = n^{0.29}$ in Figure 5.
>
> Although the MNIST dataset does not satisfy our theoretical conditions (Gaussian input, single-index model, etc.), we empirically observe similar phenomena such as emergence of spikes after one-step gradient update.
> We would like to emphasize that the main purpose of our work is to gain a deeper understanding of feature learning through analysis of a simple toy model.
> Modern deep learning operates based on vague intuitions and works for mysterious reasons.  It is a grand challenge understand its behavior, and any progress in the area is valuable in our view.
> Thus, we believe fundamental research is necessary for future advances in the field. The use of simple theoretical models to demystify various phenomenon in deep learning (such as double descent, etc.) has been a very active and successful area of research (see e.g., Mei \& Montanari, (2022), etc.).

---

> > ### Author Response · Authors · 2023-11-17
> > **Part 2/2**
> >
> > **Questions**
> >
> > 1- In Zhenmei et al. (2022), the input is defined as a linear combination of $D$ patterns, and feature learning in this setting means identifying the patterns and their relation to the label from training data.
> > In our problem, the feature means the direction $\boldsymbol{\beta}_{\star}$ in the single-index model.
> > We believe both perspectives are valid and provide independent insights to feature learning in neural networks.
> > We added details to the paper,
> > please see Section 1.1 and Appendix A in the revision.
> >
> > 2- Setting 1 serves as a baseline in our experiment.  We see that for small step size, in setting 2, the higher order component is treated as noise and the resulting loss is equal to the loss in setting 1. When we increase the step size further, higher order components starts to be learned in setting 2 and we see an improvement compared to the baseline (setting 1).
> >
> > 3- We conducted the same experiment with five gradient steps (see Figure 7 in the revised manuscript). We see that multiple spikes emerge in the spectrum. However, this goes beyond the scope of this paper and our theory for one step of gradient update. We leave the study of multiple steps for future work.

---

> > ### Comment · Reviewer_nmYk · 2023-11-22
> >
> > Thanks for the author's reply.
> >
> > For point 3, I am wondering how the sample complexity (or performance) changes as $c^\star$ changes.

---

> ### Author Response · Authors · 2023-11-22
>
> We thank the reviewer for the comments and the clarification.
>
> For example, when looking at the results in Corollary 4.5 (in particular, the expressions for $\Delta_1$ and $\Delta_2$), we can see the effect of changing $c_\star$ on the benefit we gain from doing the one-step gradient update. In particular, we can see how increasing $c_\star$ while keeping $c_{\star, 1}$ (for the $\ell = 1$ case) and $c_{\star, 1}, c_{\star, 2}$ (for the $\ell = 2$ case) constant, will reduce the benefit of the gradient update. Please also refer to the discussion below Corollary 4.5.
>
> In general, Theorem 4.4 and Corollary 4.5 fully characterize the limiting value of the loss (in the high-dimensional proportional regime of Condition 2.1) as a function of $c_\star$ (and $\phi = d/n$, etc.).
>
> Should you have any further questions, we are more than happy to answer.

---

### Official Review · Reviewer_boim · 2023-11-01

**Soundness:** 3 good
**Presentation:** 3 good
**Contribution:** 3 good
**Rating:** 6
**Confidence:** 2

**Summary:**

The paper delves into the feature learning capabilities of two-layer fully-connected neural networks. It builds on the understanding that with a constant gradient descent step size, only linear components of the target function can be learned. The research introduces a varying learning rate that grows with sample size, which results in the emergence of multiple rank-one components in the feature matrix, each representing a specific polynomial feature. Through spectral analysis, it's shown that the feature matrix's spectrum undergoes phase transitions based on the learning rate, leading to the addition of separated singular values or "spikes". These spikes, as the paper demonstrates, are aligned with polynomial features of varying degrees, influencing the neural network's ability to learn non-linear components. The study establishes that the training and test errors of the updated networks are determined by the initial feature matrix and these spikes, with specific cases illustrating the network's capacity to learn quadratic components of the target function.

**Strengths:**

Strength:
1. Paper is well organized
2. I think this paper has a good contribution to understanding the learning dynamics of non-linear features by networks, with concrete improvements over Ba et al. 2022.

**Weaknesses:**

Weakness:
1. Based on my understanding, the core advantage of the proposed analysis is from the Hermite expansion of the activation layer, which can characterize higher-order nonlinearity and explain more non-linear behaviors than the orthogonal decomposition used in Ba et al. 2022. Please clarify this.
2. The required condition on the learning rate (scaling with the number of samples) is not scalable. I never see a step size grows with the sample size in practice, which will lead to unreasonably large learning rate when learning on large-scale dataset. I understand the authors need a way to precisely characterize the benefit of large learning rates, but this condition is not realistic itself.

**Questions:**

Question:
In Figure 3 left/middle: is the total number of training steps fixed? I.e. more iterations for small LR, and fewer iterations for large LR? This is important for a fair comparison between small and large learning rates.

---

> ### Author Response · Authors · 2023-11-17
>
> We thank the reviewer for their valuable comments.
> We tried to answer each point in detail below.
> Please let us know if the reviewer has other concerns/questions regarding our paper.
> We are happy to discuss this further.
>
> We have uploaded a revised version of the manuscript on OpenReview. Please note that the numbering of theorems and sections has changed in the revised version. We are using the new numbering in our rebuttal.
>
> **Weaknesses**
>
> 1- The core contribution of this paper is that we consider a step size $\eta \asymp n^\alpha, \alpha \in (0, \frac{1}{2})$ that grows with the sample size $n$ and examine how the learned features change with $\alpha$. We show that large enough $\alpha$ can enable learning nonlinear features that is impossible when the step size is constant. Ba et al. (2022) only studied the learned features when the step size is constant.
>
> 2- Giant step sizes have recently gained a lot of attention in the deep learning theory community. See e.g., (Dandi et al. 2023). One can heuristically think of one giant step as an approximation to taking multiple small steps.
>
> We would like to emphasize that the main purpose of our work is to gain a deeper understanding of feature learning through analysis of a simple toy model.
> Modern deep learning operates based on vague intuitions and works for mysterious reasons.  It is a grand challenge understand its behavior, and any progress in the area is valuable in our view.
> Thus, we believe fundamental research is necessary for future advances in the field. The use of simple theoretical models to demystify various phenomenon in deep learning (such as double descent, etc.) has been a very active and successful area of research (see e.g., Mei \& Montanari, (2022), etc.).
>
>
> **Questions**
>
> 1- In all the experiments in the paper, we stick to the one gradient step update and only change the step size.

---

### Official Review · Reviewer_w1FP · 2023-11-08

**Soundness:** 3 good
**Presentation:** 3 good
**Contribution:** 3 good
**Rating:** 6
**Confidence:** 3

**Summary:**

The authors studied the effect of feature learning in a two-layer neural network, where the first-layer weight matrix receives one gradient update with large learning rate, and the target function is a single-index model. The main contribution is a spike decomposition of the feature matrix, where the corresponding singular vectors contain polynomial features of different degrees
depending on the scaling of step size $\eta$. This allows the authors to compute the asymptotic training error under a Gaussian equivalence conjecture and quantify the improvement in the loss due to feature learning.

**Strengths:**

This submission generalizes the result in (Ba et al. 2022) to step sizes that scales with $\eta\asymp n^\alpha$ for $\alpha\in (0,1/2)$, and provides a precise description of the nonlinear feature learning after one gradient update. Moreover, the authors identified a sequence of phase transitions with respect to the learning rate scaling, where a degree-$\ell$ spike appears when the exponent of the step size exceeds $\alpha^2>1-\frac{1}{\ell}$. This finding may motivate random matrix theory research on similar nonlinear spiked matrix models.

**Weaknesses:**

My main concern is that unlike (Ba et al. 2022), the theoretical results in the current submission does not translate to learning guarantees for the studied single-index teacher. As a result, it is unclear if a larger learning rate provides any statistical benefits, so the claim that *"for large enough step sizes, the model can learn non-linear components of the teacher function"* is not supported.
In Figure 3 the authors plotted the test error which exhibits improvement due to the learning of the quadratic component, but such improvement is not proved, and the experimental setting is only for target function with information exponent $s=1$. In fact, by inspecting the formulae for $\Delta$, it appears that the test error cannot improve for $s>1$.
This limitation needs to be explicitly mentioned in the main text.

**Questions:**

I have the following questions regarding the figures in the main text.

1. In Figure 2, do the crosses represent the theoretical predictions of the spike location? If so, how are these values obtained?

2. In Figure 3, do the solid lines correspond to the analytic predictions based on Theorems 4.4 and 4.5? If so, why do we observe fluctuations in the curve?

---

> ### Author Response · Authors · 2023-11-17
>
> We thank the reviewer for their valuable comments.
> We tried to answer each point in detail below.
> Please let us know if the reviewer has other concerns/questions regarding our paper.
> We are happy to discuss this further.
>
> We have uploaded a revised version of the manuscript on OpenReview. Please note that the numbering of theorems and sections has changed in the revised version. We are using the new numbering in our rebuttal.
>
> **Weaknesses**
>
>
> 1-  Unlike the training loss, the test error does not allow a simple expression such as Lemma K.2. Specifically, the bias term in the test error will contain a term involving two resolvent matrices with the covariance matrix of the learned features multiplied in the middle (see equation C.10 in Ba et al. (2022)). The strategy that we use to simplify such expressions is to use Woodbury formula to take out the spike terms from the resolvent matrices and study terms appearing in the expansion. However, for the case of this complicated formula, one needs to study various forms of interactions between the spike terms. We find that this would require a much more laborious calculation, and since our paper is already quite long (cca 50 pages total), we believe it is better suited for follow-up work.
>
> In the nonlinear random matrix theory community, it is common to only derive the limiting training loss  after the study of the spectrum of the feature matrix (see e.g., Pennington \& Worah (2017); Louart et al. (2017), etc.). Deriving the precise asymptotics for the test loss is a very interesting problem and we leave it as a future work.
>
>
> We have added this to a limitation section at the end of the paper in the revised manuscript.
> Also, we changed out statement to "Moreover, for large enough step sizes,
> the model can fit non-linear components of the teacher function."
>
>
> 2- We thank the reviewer for pointing this out. It is true that when learning single index functions, with one gradient step, in the proportional regime, the information exponent needs to be one in order for the gradient step to be beneficial. This is also true for the results of Ba et al. (2022).  Recently, Dandi et al. (2023) showed that for single index models with information exponent $\kappa$, there are hard directions such that learning them will require $\Theta(d^\kappa)$ samples.
> They also show that with one gradient step, and a sample size $\Theta(d)$, only a single direction of a multi-index target function can be learned. In the present work, we study the problem of learning nonlinear components of a single-index target function with $\kappa = 1$. We clarified these points in the related works, and the limitations section that can be found in the revised manuscript.
>
> **Questions**
>
> 1- In this plot, the crosses are showing the location of the singular values, based on simulations.
>
> 2- In this plot, the the curves are drawn using simulation results. Note that the convergence can be very slow in this problem. For example note that when $\ell = 2$, we need to have a large enough $n$, $N$, $d$ such that terms like $n^{1/2}/\eta^2$ to go to zero with $\eta = n^{\alpha}$ with $\alpha \in (1/4, 1/3)$  (Section K.2.1, analysis of terms (4,2) and (2,4)).
> For example, if we choose $\alpha = 0.3$, making $n^{1/2}/\eta^2 = 0.01$ requires $n = 10^{20}$.
> In other words, with small $n, N, d \approx 10^{3} \sim 10^4$ that can be handled with a personal computer and academic cluster that we have access to, we cannot expect the simulated result to closely match the asymptotics.
> We believe some prior works in the polynomial scaling settings might have experienced the same scalability issue, and therefore they could only provide cartoons demonstrating the staircase plot (see e.g.,  Ghorbani et al. (2020), Misiakiewicz (2022), etc.).

---

### Official Review · Reviewer_EiLn · 2023-11-10

**Soundness:** 3 good
**Presentation:** 3 good
**Contribution:** 3 good
**Rating:** 5
**Confidence:** 4

**Summary:**

This paper studied the spikes in the feature map matrix of a two-layer neural network with large step gradient descent (GD) on mean square loss. When learning rate $\eta=n^\alpha$ with $\frac{\ell-1}{2\ell}<\alpha<\frac{\ell}{2\ell+\ell}$, there will be $\ell$ large spikes in the feature map matrix (or the Conjugate Kernel matrix) and these spikes are correlated to the degree $\ell$ Hermite components of the target function. The asymptotic training errors for ridge regression of this trained feature map matrix has been presented in the proportional limit. This paper fills the gap in the learning rates, when $1\ll\eta\ll\sqrt{n}$, in Ba et al. (2022).

**Strengths:**

Despite its technical nature, the paper is very well written. The particular setting the authors study is novel and interesting for both the random matrix theory community and deep learning theory. The detailed analysis of the scaling of learning provides us with a more comprehensive understanding of the features learned in GD training processes, although the authors only consider one step of GD. The result is precise and clean, showing the asymptotic improvement in the loss for potential feature learning.

**Weaknesses:**

1. A limitations section is missing. In the conclusion section, the authors should state the limitations of the assumptions and the results. The authors only proved the improvement of training loss in this two-stage training process for neural networks (NNs). There is a lack of analysis for generalization errors, although I understand there may be some difficulty with this kind of theoretical result. There should be some remark or discussion on this, or providing some conjectures related to the generalization error.

2. Additional simulations are needed. In Section 5, there are only cases for linear and quadratic target functions. It would be better to provide more simulations for training and testing errors with more complicated target functions to show the feature learning when $\eta$ is sufficiently large. There is no empirical simulation for the staircase phenomenon in Figure 3 (Right).

3. Theorems 4.4 and 4.5 rely on Conjecture 4.3. However, this conjecture is not well stated in the main text. It would be also better to explain the difficulty of the proof and why this conjecture cannot be proved by previous results like Hu&Lu, (2023) and Ba et al. (2022).

**Questions:**

1. In Section 2.1, the scaling of the neural network is different from Ba et al. (2022). In Ba et al. (2022), they used a mean-field regime with learning rate $\eta\sqrt{N}$ and there is an extra $1/\sqrt{N}$ for the second layer $\mathbf{a}$. Is this regime the same as the setting of this paper?

2. The initialization of $\mathbf{W}_0$ is sampled from a uniform distribution on the unit sphere, which is different from the Gaussian initialization of Ba et al. (2022). I guess this initialization will make the analysis simpler, e.g. Lemma B.1 can be applied directly. This should be mentioned somewhere in the paper and explain why you use this initialization.

3. Condition 2.3 assumes that $\sigma$ has bounded first three derivatives but this won't be true if you consider the general polynomial activation function, which is set in Theorem 3.3. For Theorems 3.4, 4.1-4.2, and 4.4-4.5, do you only consider $\sigma$ as a polynomial or center ReLU function? I am confused why Theorem 3.3 needs polynomial activation functions which may contradict with Condition 2.3.

4. For Figure 2, are the locations of the spikes and the alignments empirically simulated or can you predict them from your theory? From Theorem 3.4, these alignments should converge to one, right?

5. In Theorem 3.3, how about the case when $\alpha=(\ell-1)/(2\ell)$? Any observations in this critical regime?

6. [1] and [2] also studied the initial feature matrix $\\mathbf{F}_{0}$. And [1] also presented the limit of training error for random feature ridge regression but with a slightly different definition than yours.

7. Above Theorem 3.4, vector $\mathbf{w}_i$ is not defined.

8. Below (4), why does $c_{>1}$ also include $c_1$?

9. For Theorems 4.4 and 4.3, can you say something about some extreme cases? For instance, $n\gg N,d$ or $N\gg n,d$.

10. In Section 4.2, why not present the theory of training loss for general $\ell$ like Appendix L? Can Appendix L directly cover Theorems 4.4 and 4.3? Besides, using Appendix L, can you plot Figure 3 (Right) and show that the training loss is always decreasing?

11. I cannot see how Figure 3 (Left and Middle) matches Theorems 4.4 and 4.3 for training loss with $\log\eta/\log n<1/4$ or $1/4<\log\eta/\log n<1/3$. You may need to point out the threshold in the figures. Besides, in the middle figure, why is the testing error increasing for setting 1 with a large learning rate? Can you explain this phenomenon here? It seems like in this case, we do not have improvement for feature learning. Besides, there should be a benchmark, the prediction risk for the best linear model in this figure to compare with the feature learning.

12. In Appendix A, a typo for the definition of $\mathbf{R}_{0}$.

13. Lemma B.1, do you need to require $\\|a\\|=\\|b\\|=1$?

14. In the proof of proposition 3.1, how do you use Lemma J.1 to derive the limit of $\boldsymbol{\beta}^\top\boldsymbol{\beta}_{\*}$?

15. In the proof of Theorem 3.4, how do you show the rank of the sum of the spikes is exactly $\ell$? Is it easy to see that $(\tilde{\mathbf{X}}\boldsymbol{\beta})^{\odot k}$ are linearly independent for different $k$?

16. In the final result of Appendix L, why is $c_{\*,0}$ also included in the asymptotic difference of the training errors? In $\ell=1,2$, there is no $c_{\*,0}$; see (5) and (6). And What is $M$ in the summation? There should be some discussion about this result.



==================================================================================================

[1] Louart, et al. "A random matrix approach to neural networks."

[2] Fan and Wang. "Spectra of the conjugate kernel and neural tangent kernel for linear-width neural networks."

---

> ### Author Response · Authors · 2023-11-17
> **Part 1/3**
>
> We thank the reviewer for their valuable comments. We tried to answer each point in detail below. Please let us know if the reviewer has other concerns/questions regarding our paper. We are happy to discuss this further.
>
> We have uploaded a revised version of the manuscript on OpenReview. Please note that the numbering of theorems and sections has changed in the revised version. We are using the new numbering in our rebuttal.
>
> **Weaknesses**
>
> 1- We thank the reviewer for the suggestion to write a limitations section. We have written a limitation section at the end of the paper in the revised manuscript.
>
> In Section 4 of the paper, we wanted to demonstrate that the emergence of spikes in the spectrum of the feature matrix will indeed translate to going beyond linear models for large enough step size through the analysis of training loss. Unlike the training loss, the test error does not allow a simple expression such as Lemma K.2. Specifically, the bias term in the test error will contain a term involving two resolvent matrices with the covariance matrix of the learned features multiplied in the middle (see equation C.10 in Ba et al. (2022)). The strategy that we use to simplify such expressions is to use Woodbury formula to take out the spike terms from the resolvent matrices and study terms appearing in the expansion. However, for the case of this complicated formula, one needs to study various forms of interactions between the spike terms. We find that this would require a much more laborious calculation, and since our paper is already quite long (cca 50 pages total), we believe it is better suited for follow-up work.
>
> In the nonlinear random matrix theory community, it is common to only derive the limiting training loss  after the study of the spectrum of the feature matrix (see e.g., Pennington \& Worah (2017); Louart et al. (2017), etc.). Deriving the precise asymptotics for the test loss is an interesting problem and we leave it as a future work.
>
> 2- Regarding your point on conducting experiments for the staircase plot in Figure~3 (Right), for example note that when $\ell = 2$, we need to have a large enough $n$, $N$, $d$ such that terms like $n^{1/2}/\eta^2$ to go to zero with $\eta = n^{\alpha}$ with $\alpha \in (1/4, 1/3)$  (Section K.2.1, analysis of terms (4,2) and (2,4)).
> For example, if we choose $\alpha = 0.3$, making $n^{1/2}/\eta^2 = 0.01$ requires $n = 10^{20}$.
> In other words, with small $n, N, d \approx 10^{3} \sim 10^4$ that can be handled with our personal computers and academic clusters at the moment, we cannot expect the simulated result to closely match the asymptotics.
> We believe some prior works in the polynomial scaling settings might have experienced the same scalability issue, and therefore they could only provide cartoons demonstrating the staircase plot (see e.g.,  Ghorbani et al. (2020), Misiakiewicz (2022), etc.).
>
> We repeated the experiment in Section 5 with a target function with cubic term.
> See Figure 4 in the revised manuscript.
> We observe that the cubic part is learned with a large enough step size.
>
> 3- Our results in Section 4.2 rely on a Gaussian equivalence
> conjecture for the untrained features $\mathbf{F}_0$ (and not the trained features $\mathbf{F}$). In the proofs of Section 4.2, by the application of Woodbury's formula, and taking out the spike terms in $\mathbf{F}$ from the resolvent matrix $\mathbf{R}$ to reduce the problem to a problem involving  $\mathbf{F}_0$ and spikes, and we only need a Gaussian equivalence theorem to linearize $\mathbf{F}_0$; as it is common in the literature of random features models. The Gaussian equivalence conjecture we use, despite being very similar to the results discussed in Section J, does not directly follow from prior work. One can try to use the Lindeberg exchange idea used by Hu \& Lu (2023) and Bosch et al. (2023) to try to prove this specific conjecture. Although this approach might work, due to the complication form of the terms under consideration (listed in Section I),  we have not yet been able to prove this specific form of Gaussian equivalence conjecture, despite some serious attempts.
> We have also discussed this in the limitations section.

---

> > ### Author Response · Authors · 2023-11-17
> > **Part 2/3**
> >
> > **Questions**
> >
> > 1- Yes. The two settings are exactly identical. The scaling difference of the second layer is the reason behind the extra $\sqrt{N}$ term in the steps sizes in Ba et al. (2022) and this paper.
> >
> > 2- Yes, as pointed out by the reviewer, this choice of initialization will make the analysis simpler; e.g., it will enable us to use Lemma C.1 without the need for a normalization. However, many arguments can be shown to hold if we switch from the uniform distribution over sphere to Gaussian. For example, please see Section M.5. The choice of uniform over sphere distribution for weights is commonly used in the papers that study random features regression (see e.g. Mei \& Montanari (2022)). We mentioned this in the revision in Section 2.1.
> >
> > 3- Thanks for pointing out this error. In the revision, we added a decay condition to Condition 2.3 and modified the theorem statements and proof in Appendix F accordingly.
> > This condition is satisfied by activation functions such as (shifted) ReLU, hyperbolic tangent, and sigmoid.
> >
> > 4- Locations of the spikes in Figure 2 are not from the theoretical predictions but are empirically generated, see Section 5 for the experimental detail.
> > For the $\ell = 1$ case, the alignment between $\tilde{\mathbf{X}} \boldsymbol{\beta}$ and the first left singular vector converges to one by Theorem 3.4. However, for $\ell > 1$, since $[(\tilde{\mathbf{X}} \boldsymbol{\beta})^{\circ k}]_{k = 1}^\ell$ is not necessarily orthogonal, we can only argue about the subspace distance between their span and the span of top-$\ell$ left singular vectors.
> >
> >
> > 5- The critical regime where $\alpha = \frac{\ell - 1}{2\ell}$ requires a more fine-grained analysis and is left for future work. For example, in the proof of Theorem 3.3 (page 18), the log terms in the bound of  $\boldsymbol{\Delta_{1}}, \dots, \boldsymbol{\Delta}_{4}$  can no longer be neglected. Also, in the computation of limiting training loss, there will be new asymptotically non-vanishing entries. For example, when $\eta = N^{1/4}$, the terms (2,4) and (4,2) in the matrix $\mathbf{T}^{-1}$ defined below Eq. (26) will no longer be $o(1)$ as $\sqrt{N}/\eta^2$ will go to a non-zero constant. We added a discussion about this in the limitations and future work section.
> >
> >
> > In a very recent work (Guionnet et al. (2023)), that appeared on Arxiv after this paper was submitted to ICLR, the authors study study a problem with a setting very similar to setting in Section 3 of the current paper, with $\alpha = \frac{\ell-1}{2\ell}$ (exactly on the boundary), and derive BBP-like phase transitions for the leading singular value of the feature matrix; assuming the first $\ell-1$ Hermite coefficients of the activation function are zero.
> >
> > =============
> >
> > Alice Guionnet, Justin Ko, Florent Krzakala, Pierre Mergny, Lenka Zdeborová. Spectral Phase Transitions in Non-Linear Wigner Spiked Models, 2023. (https://arxiv.org/abs/2310.14055)
> >
> > =============
> >
> > 6- We thank the reviewer for pointing out these two papers. We have added them (alongside other nonlinear random matrix theory papers) to the related works section.
> >
> > 7- Thanks for pointing this out. We defined them in the revised manuscript when defining the updated weight matrix $\mathbf{W}$.
> >
> >
> > 9- In the extreme case, the expressions for $m_1, m_2$ will further simplify. For example, if $n \gg N, d$, then $m_1, m_2  \to \phi / \lambda \psi$. Note that in this limit  $L_{tr}(\mathbf{F_0}) = \sigma_\epsilon^2 + c_\star^2 = \sigma_\epsilon^2 + c_{\star,1}^2 + 2 c_{\star,2}^2 + c_{\star,>2}^2$. Plugging the expressions in Corollary 4.5, we see that for example when $\ell = 2$, we have $\mathcal{L}(\mathbf{F}) \to \sigma_\epsilon^2 + \frac{2 c_{\star,2}^2}{3} + c_{\star,>2}^2$. Note that the term corresponding to the linear component of the teacher function in $\mathcal{L}(\mathbf{F_0})$ completely canceled out with the corresponding term in $\Delta_2$.
> >
> > Note that the training loss improvement becomes independent of $\sigma_\varepsilon$. This is intuitive since large sample size can make the gradient more accurate regardless of the noise level.
> >
> > We have added this to appendix $M$ of the revised manuscript.

---

> > > ### Author Response · Authors · 2023-11-17
> > > **Part 3/3**
> > >
> > > 10, 16- We thank the reviewer for this suggestion. As the reviewer suggested, we now present the theorem for the limiting training loss for general $\ell$ first and state the special cases $\ell = 1$ and $\ell = 2$ as special cases.
> > >
> > > As the reviewer points out, there was a minor mistake in the final expression of limiting training loss with general $\ell$ in the previous version. This is now fixed. Please see section L in the revised manuscript. The result for general $\ell$ does indeed cover the results for $\ell = 1$ and $\ell = 2$ and gives the expressions in Corollary 4.5. We thank the reviewer for pointing this out.
> > >
> > >
> > > According to the expression of $\Delta_\ell$ in Theorem L.1, we can see that for any $\ell$, there is always an improvement due to the gradient update compared to random features regression. However, it is not guaranteed that the loss will decrease at every threshold. Looking at the special cases of $\ell$ in Corollary 4.5, we see that when going from
> > > $\ell = 1$ to $\ell = 2$, the loss will indeed decrease.
> > >
> > >
> > > 11- Since the simulation is done with finite $n = 1,000$, the curves do not perfectly match the asymptotics derived in the theorems.
> > > However, we can still see that the training loss curves branch near $\log \eta / \log n = \frac{1}{4}$, where the model starts to learn the quadratic part of the target function in Setting 2.
> > > As we discuss in the limitation section, we leave the complete analysis of test error to future work.
> > > Intuitively, we believe that the test error cannot enjoy improvement from the quadratic feature learning when $\log \eta / \log n \in (1/4, 1/3)$ since the quadratic part does not exist in Setting 1, and having less diverse features (due to the large step) leads to increase in the test error.
> > > Finally, setting 1 serves as a baseline in our experiment.  We see that for small step size, in setting 2, the higher order component is treated as noise and the resulting loss is equal to the loss in setting 1. When we increase the step size further, the higher order component starts to be learned in setting 2.
> > >
> > > 8, 12, 13- Thanks for pointing these out. These typos are now fixed in the revised manuscript.
> > >
> > >
> > > 14- We have clarified how we used Lemma K.1 in the proof of Proposition 3.1 in the revised manuscript.
> > >
> > > 15- The vectors are linearly independent almost surely. Conditioned on $\boldsymbol{\beta}$, the vector $(\boldsymbol{\beta}^\top \tilde{\mathbf{x}}_i, \dots, (\boldsymbol{\beta}^\top \tilde{\mathbf{x}}_i)^{\ell})$ has the distribution $(Z, \dots, Z^\ell)$ with $Z \sim \mathrm{N}(0, \Vert \boldsymbol{\beta} \Vert_2^2)$.
> > > It can be seen that the distribution is supported on $\mathbb{R}^\ell$, and therefore $\ell$ independent copies from this distribution will be almost surely linearly independent.

---

> > > > ### Author Response · Authors · 2023-11-22
> > > >
> > > > Dear Reviewer EiLn,
> > > >
> > > > We thank you for your valuable comments on our manuscript.
> > > >
> > > > We have provided detailed responses in the rebuttal, including responses to the points you raised regarding the analysis of the test error, simulations, gaussian equivalence, analysis in the boundary cases, etc. New experiments are also conducted on more complicated target functions.
> > > >
> > > > Also, we have revised the manuscript according to the suggestions from the reviewers. For example, we have  added a new limitations and future works section, we have rearranged the paper so that the general $\ell$ result is presented first, and we have added new citations and expanded the related works.
> > > > Since the discussion period is ending today, we are wondering if you could take some time to read our responses and let us know if we have adequately addressed your questions. Should you have any further questions, we are more than happy to answer.
> > > >
> > > > Best regards,
> > > >
> > > > Authors

---

### Author Response · Authors · 2023-11-21

Dear Reviewers,

We thank you for your valuable comments on our manuscript. We have provided detailed responses in the rebuttal. We have also revised the manuscript according to the suggestions made in the reviews. Since the discussion period is ending on Nov. 22nd, we are wondering if you could take some time to read our responses and let us know if we have adequately addressed your questions. Should you have any further questions, we are more than happy to answer.

Best regards,

Authors

---

### Meta-Review · Area_Chair_bQSA · 2023-12-14

**Metareview:**

This paper shows that one gradient descent update with a large step size can find not only the first order component of the target function but also the second order information depending on the scale of the step size. This is a direct extension of previous work Ba et al. (2022). The theoretical result is verified through some synthetic data experiments.

The extension to the higher order exponent is an interesting extension from Ba et al. (2022). On the other hand, it is pointed out by reviewers that the analysis is only for the training loss but there is no theory that ensures benefit of large step size feature learning in terms of the test loss. This would be difficult, but it is indeed desired from the practical point of view. Another concern is its experiments. Although the authors manages to extended experiments during the rebuttal period, it is better that the experiments would be done in a more organized manner.

For these reasons, I unfortunately cannot recommend acceptance for the current version. It would be much more polished by carefully placing the position of this paper and showing results in more understandable and convincing way.

**Justification For Why Not Higher Score:**

Although the theoretical analysis shows an interesting fact, it is not completely satisfactory because it does not give any advantage in terms of test loss. Moreover, the writing of the paper could be improved.

**Justification For Why Not Lower Score:**

N/A

---

### Decision · Program_Chairs · 2024-01-16

Reject